# PROVABLE ANYTIME ENSEMBLE SAMPLING ALGORITHMS IN NONLINEAR CONTEXTUAL BANDITS

## ABSTRACT

We provide a unified algorithmic framework for ensemble sampling in nonlinear contextual bandits and develop corresponding regret bounds for two most common nonlinear contextual bandit settings: Generalized Linear Ensemble Sampling (`GLM-ES`) for generalized linear bandits and Neural Ensemble Sampling (`Neural-ES`) for neural contextual bandits. Both methods maintain multiple estimators for the reward model parameters via maximum likelihood estimation on randomly perturbed data. We prove high-probability frequentist regret bounds of $\mathcal{O}(d^{3/2}\sqrt{T} + d^{9/2})$ for `GLM-ES` and $\mathcal{O}(\widetilde{d}\sqrt{T})$ for `Neural-ES`, where $d$ is the dimension of feature vectors, $\widetilde{d}$ is the effective dimension of a neural tangent kernel matrix and $T$ is the number of rounds. These regret bounds match the state-of-the-art results of randomized exploration algorithms in nonlinear contextual bandit settings. In the theoretical analysis, we introduce techniques that address challenges specific to nonlinear models. Practically, we remove fixed-time horizon assumptions by developing anytime versions of our algorithms, suitable when $T$ is unknown. Finally, we empirically evaluate `GLM-ES`, `Neural-ES` and their anytime variants, demonstrating strong performance. Overall, our results establish ensemble sampling as a provable and practical randomized exploration approach for nonlinear contextual bandits.

## 1 INTRODUCTION

The contextual bandit is an online learning problem where an agent interacts with an environment by pulling arms, each associated with a feature vector. After each pull, the agent receives a stochastic reward whose expected value depends on the chosen arm's feature vector. The agent's goal is to maximize the accumulated reward. Contextual bandits provide a natural abstraction for real-world sequential decision-making problems such as content recommendation (Zhu & Van Roy, 2023) and clinical trials (Varatharajah & Berry, 2022). To maximize rewards, the agent must learn the mapping from an arm's feature vector to its expected reward. Most prior work has focused on the linear contextual bandit setting (Abbasi-Yadkori et al., 2011; Abeille & Lazaric, 2017; Kveton et al., 2020b), where the expected reward is assumed to be a linear function of the feature vector. While this assumption facilitates theoretical analysis and efficient implementations, it fails to capture complex relationships between features and rewards. This has motivated the study of nonlinear contextual bandits, where the expected reward is modeled as a nonlinear function of the arm's features, e.g., through generalized linear models (GLMs) or neural networks. In the GLM setting (Filippi et al., 2010; Kveton et al., 2020b), the reward is generated by applying a nonlinear function $\mu(\cdot)$ to the inner product of the feature vector and an unknown parameter vector, a setting referred to as generalized linear bandits. In more general cases where the reward cannot be expressed within the GLM structure, neural contextual bandits approximate the reward function with deep neural networks, without assuming any particular functional form (Jacot et al., 2018; Zhou et al., 2020; Zhang et al., 2021; Xu et al., 2022a; Jia et al., 2022). These nonlinear approaches have substantially improved the empirical performance of linear bandit algorithms, especially in complex environments (Xu et al., 2022b; Jia et al., 2022).

While nonlinear models enhance expressivity, they also complicate the design of effective exploration strategies. Existing exploration methods such as Upper Confidence Bound (UCB) (Abbasi-Yadkori et al., 2011) and Thompson Sampling (TS) (Agrawal & Goyal, 2013) heavily depend on

reward structure and distributional assumptions. Extending them to nonlinear settings requires significant approximations (Zhou et al., 2020; Zhang et al., 2021; Xu et al., 2022a;b), often making the methods impractical in real-world applications. This motivates the search for exploration strategies that combine strong empirical performance with theoretical guarantees in nonlinear bandits. Ensemble sampling (Lu & Van Roy, 2017) has emerged as a promising class of algorithms for online decision-making problems, including bandits (Lee & Oh, 2024), deep reinforcement learning (Osband et al., 2016), and recommendation systems (Zhu & Van Roy, 2023). Ensemble sampling maintains an ensemble of $m$ models, each trained on randomly perturbed historical data consisting of arm features and rewards. At each round, one model is sampled to estimate expected rewards, and the arm with the highest estimate is selected. After receiving the reward, the new data point—with random perturbation—is added to the dataset, and all models are updated.

In recent years, ensemble sampling has gained popularity due to its strong empirical performance and moderate computational cost. However, theoretical understanding has lagged behind. Existing analyses provide regret guarantees only in the linear contextual bandit setting. For example, Lee & Oh (2024) proved a high-probability $T$-round regret bound of $\widetilde{\mathcal{O}}(d^{3/2}\sqrt{T})$ with ensemble size $m = \Omega(K \log T)$, where $d$ is the feature dimension and $K$ is the number of arms. More recently, Janz et al. (2024a) established a regret bound of $\widetilde{\mathcal{O}}(d^{5/2}\sqrt{T})$ for infinitely many arms with ensemble size $\Theta(d \log T)$. While these theoretical works provide valuable insights, they are far from fully elucidating the empirical success of ensemble sampling in complex decision-making applications, where reward models are typically nonlinear in the arm features.

In this work, we extend ensemble sampling to nonlinear bandit settings with finitely many arms. Specifically, we study the two most widely used models: generalized linear bandits and neural contextual bandits. We show that ensemble sampling in these settings achieves high-probability regret bounds matching the state-of-the-art for randomized exploration algorithms, with ensemble size logarithmic in $T$. In addition, we develop anytime versions of ensemble sampling using the doubling trick, addressing the limitation that ensemble size and other hyperparameters traditionally depend on the horizon $T$. These anytime variants significantly broaden the applicability of ensemble sampling. Finally, we complement our theoretical results with experiments on `Lin-ES`, `GLM-ES`, and `Neural-ES` against baselines. Our experiments highlight the practicality of ensemble sampling in nonlinear bandits, balancing strong performance with computational efficiency.

Our contributions are summarized as follows.

- We propose a general framework for ensemble sampling in bandit problems and introduce `GLM-ES` and `Neural-ES` as its realizations in nonlinear settings.
- We provide theoretical analyses of `GLM-ES` and `Neural-ES`, proving regret bounds of $\widetilde{\mathcal{O}}(d^{3/2}\sqrt{T} + d^{9/2})$ and $\widetilde{\mathcal{O}}(\widetilde{d}\sqrt{T})$, respectively, both matching the state-of-the-art for randomized exploration algorithms. To the best of our knowledge, these are the first high-probability regret bounds for ensemble sampling in nonlinear bandit settings.
- For generalized linear bandits, we optimize the warm-up procedure in existing literature, reducing regret from $d^9$ to $d^{9/2}$. This improvement also applies to perturbed-history type of exploration strategies. We further remove the need for adaptive reward perturbations, simplifying the design and improving the efficiency of our algorithm.
- We develop anytime versions of ensemble sampling using the doubling trick and show that their asymptotic cumulative regret guarantees are preserved.
- We conduct empirical evaluations comparing cumulative regret and computational cost with baselines, demonstrating the practicality of ensemble sampling.

## 2 RELATED WORK

**Randomized Exploration** Randomized exploration strategies add controlled randomness to promote exploration of actions with high uncertainty. In sequential decision making problems, randomized exploration strategies often outperforms deterministic strategies such as Upper Confidence Bound (UCB) (Chu et al., 2011; Lattimore & Szepesvári, 2020) by preventing early convergence to suboptimal actions (Jin et al., 2021; 2023). Among such methods, Thompson Sampling (TS) (Thompson, 1933) is a key approach for multi-armed bandits (Agrawal & Goyal, 2017), contextual bandits (Agrawal & Goyal, 2013), and RL (Osband et al., 2013). TS maintains a posterior over

model parameters, updated each round from a prior (e.g., Gaussian) and observed rewards (Agrawal & Goyal, 2013), and samples a parameter from this posterior for arm selection. Despite its simplicity, many TS variants rely on exact posteriors or accurate Laplace approximations, which can be costly. To address this, approximate sampling methods such as Langevin Monte Carlo (LMC) (Xu et al., 2022b; Hsu et al., 2024), Stochastic gradient Langevin dynamics (SGLD) variants (Mazumdar et al., 2020; Zheng et al., 2024) and variational inference (Clavier et al., 2024) have been developed and applied to various problem settings, including multi-armed bandits with non-conjugate or highly nonlinear rewards, nonlinear contextual bandits and RL (Ishfaq et al., 2024a;b; Hsu et al., 2024). Another important method is perturb-history exploration (PHE) method, which involves introducing random perturbations in the historical data to approximate posterior sampling, making it applicable to complex reward distributions (Kveton et al., 2020b; Ishfaq et al., 2021). Ensemble sampling maintains a small set of independently perturbed model replicas and selects arms using a randomly chosen replica (Lu & Van Roy, 2017). Follow-up work provided theory for the linear contextual bandit setting. Qin et al. (2022) gave the first regret bound. Janz et al. (2024a) tightened guarantees with an ensemble of size $\Theta(d \log T)$ for linear bandits with infinitely many arms. LinES (Lee & Oh, 2024) further improved the regret to $\widetilde{O}(d^{3/2}\sqrt{T})$ and clarified its connection to LinPHE.

**Generalized Linear Bandits** Generalized linear contextual bandits model rewards via a link function of a linear predictor, extending linear bandits to a more general setting. Early work introduced GLM-UCB and proved regret guarantees under standard regularity conditions (Filippi et al., 2010). Subsequent advances focused on optimality and efficiency. Li et al. (2017) gave provably optimal algorithms with refined confidence sets. Ding et al. (2021) combined online stochastic gradient updates with Thompson Sampling for scalable inference. Kveton et al. (2020b) developed randomized exploration for GLMs with sharper analyses. Perturbation-based methods provide practical alternatives: linearly perturbed loss minimization yields simple, sampling-free exploration with strong guarantees (Janz et al., 2024b), and PHE adapts perturb-history exploration to sub-Gaussian GLMs (Liu, 2023). Sawarni et al. (2024) analyze GLMs under limited adaptivity (batched policies) with communication and deployment constraints. Anytime-valid confidence sequences for GLMs enable principled UCB/TS decisions and valid sequential inference (Lee et al., 2024). At large horizons, One-pass update methods achieve near-optimal regret with single-pass, low-memory updates (Zhang et al., 2025).

**Neural Bandits** Neural bandits combine deep neural networks (DNNs) with contextual bandit algorithms. This setting leverages the representation power of DNNs and insights from neural tangent kernel (NTK) theory (Jacot et al., 2018). NeuralUCB (Zhou et al., 2020) builds confidence sets using DNN-derived random features to enable UCB-style exploration. NeuralTS (Zhang et al., 2021) extends Thompson Sampling to this setting by using a neural estimator to approximate the posterior over rewards. NeuralLCB (Nguyen-Tang et al., 2022) studies offline neural bandits and uses a neural lower confidence bound to take pessimistic decisions under uncertainty. To reduce the compute burden of explicit exploration, NPR (Jia et al., 2022) learns a neural bandit model with perturbed rewards, avoiding separate exploration updates. Subsequent work explores added networks for exploitation (Ban et al., 2022), provable guarantees with smooth activations (Salgia, 2023), and extensions to combinatorial selection (Hwang et al., 2023; Atalar & Joe-Wong, 2025; Wang et al., 2025), where the learner selects a subset (e.g., multiple arms under constraints) each round and receives a corresponding reward.

## 3 PROBLEM SETTING

Contextual bandits form a broad class of sequential decision making problems where the player chooses an action from an observed action set based on interaction history. Each action is associated with a feature vector (context). At round $t$, the player observes an action set $\mathcal{X}_t \subseteq \mathbb{R}^d$, where we assume for any $X \in \mathcal{X}_t$, $\|X\|_2 \leq 1$. The agent then selects an arm (action) $X_t \in \mathcal{X}_t$ and the environment immediately reveals a reward $Y_t$. We consider the setting that the action set is finite with $K$ arms and fixed across different rounds. For simplicity, we use $\mathcal{X}$ to denote the fixed arm set. We assume that the mean reward for a feature $X \in \mathbb{R}^d$ is generated by an unknown function $h(X) : \mathbb{R}^d \to \mathbb{R}$, and the observed reward satisfies $Y = h(X) + \eta$, where $\eta$ is observation noise and is assumed to be $\sigma$-sub-gaussian. In general, the form of $h(X)$ is unknown. One special case is that we set reward model to be a linear function $h(X) = X^\top \theta^*$, then $h(X)$ is parameterized

using a vector $\theta^* \in \mathbb{R}^d$ and we have the standard linear contextual bandit setting (Chu et al., 2011; Abbasi-Yadkori et al., 2011; Agrawal & Goyal, 2013).

In this work, we consider nonlinear contextual bandits where the reward model $h(X)$ is a nonlinear function of feature vector $X$. We focus on two most common nonlinear settings: (1) generalized linear bandits with $h(X) = \mu(X^\top \theta^*_{\text{GLM}})$, where $\theta^*_{\text{GLM}} \in \mathbb{R}^d$ is the true parameter with $\|\theta^*_{\text{GLM}}\| \leq S$ and $\mu(\cdot)$ is a strictly increasing link function (Li et al., 2017; Kveton et al., 2020b); (2) neural contextual bandits where no assumptions are made about $h(X)$ other than that it is bounded, we use a neural network $f(X; \theta_{\text{Neural}})$ to approximate $h(X)$, where $\theta_{\text{Neural}}$ is the concatenation of all weights and its dimension is determined by the structure of the neural network (Zhou et al., 2020; Zhang et al., 2021; Jia et al., 2022).

The goal of a bandit algorithm is to maximize the cumulative reward over a horizon $T$, equivalently to minimize the pseudo-regret (Lattimore & Szepesvári, 2020)

$$R(T) = \sum_{t=1}^{T} \big(h(X^*) - h(X_t)\big), \tag{3.1}$$

where $X_t \in \mathcal{X}$ is the arm played at round $t$, and $X^* = \text{argmax}_{X \in \mathcal{X}} h(X)$ is the arm with the highest expected reward. To minimize the cumulative regret $R(T)$, the agent needs to collect information and learn the true reward model $h(X)$ from interactions with the environment.

**Notations**    We adopt the following standard notations throughout this paper. The set $\{1, 2, ...n\}$ is denoted by $[n]$. For any positive semi-definite matrix $M$, we use $\lambda_{\max}(M)$ and $\lambda_{\min}(M) \geq 0$ to denote maximum and minimum eigenvalues of $M$. The 2-norm of a symmetric matrix $M$ is defined as $\|M\|_2 = |\lambda_{\max}(M)|$. For any positive semi-definite matrices $M_1$ and $M_2$, $M_1 \preceq M_2$ if and only if $x^\top M_1 x \leq x^\top M_2 x$ for all $x \in \mathbb{R}^d$. All vectors are column vectors. For any vector $x$, we use the following vector norms: $\|x\|_2 = \sqrt{x^\top x}$, $\|x\|_M = \sqrt{x^\top M x}$. The indicator function that event $\mathcal{E}$ occurs is $\mathbb{1}\{\mathcal{E}\}$. We use $\widetilde{\mathcal{O}}$ for the big-O notation up to logarithmic factors. We define the filtration $\mathcal{F}'_t = \sigma(X_1, ..., X_t, Y_1, ..., Y_t)$ as the $\sigma$-algebra generated by the pulled arms and observed rewards by the end of round $t \in [T]$, and filtration $\mathcal{F}_t = \sigma(X_1, ..., X_t, Y_1, ..., Y_t, j_1, ..., j_t, \{Z_l^j\})$ as the $\sigma$-algebra generated by the pulled arms, observed rewards, chosen model and perturbations by the end of round $t \in [T]$.

## 4    Ensemble Sampling for Nonlinear Contextual Bandits

We apply the design principle of ensemble sampling to nonlinear contextual bandits, extending previous works on linear ensemble sampling (`Lin-ES`) to broader applications. In this section, we first present a unified algorithm framework for ensemble sampling in nonlinear contextual bandit, then we focus on two common nonlinear cases: 1) generalized linear contextual bandits (GLM); 2) neural contextual bandits, and respectively provide algorithms `GLM-ES` and `Neural-ES`.

### 4.1    Unified Algorithm Framework

Ensemble sampling follows the randomized exploration principle and exploration is realized through adding perturbations to observed rewards. Therefore, the perturbed history $\mathcal{D}_t = \{(X_l, Y_l + Z_l)\}_{l=1}^{t}$ is utilized to estimate the true reward model $h(X)$ and choose the best arm, where $\{X_l\}_{l=1}^{t}$ are pulled arms, $\{Y_l\}_{l=1}^{t}$ are observed rewards and $\{Z_l\}_{l=1}^{t}$ are perturbations.

In particular, we maintain an ensemble of perturbed models, each with different perturbed history $\mathcal{D}_t^j$, where $j \in [m]$ is the model index. At each round, we randomly select one model to estimate the mean reward of each arm in the arm set, then select the arm that maximizes the estimated mean reward. We use $f_t(X)$ to denote the estimated mean reward of arm $X$ from the chosen model in round $t$. After observing the reward $Y_t$ from the environment, each model in the ensemble is updated incrementally based on history $\mathcal{D}_{t+1}^j$. At each round, we only sample one perturbation $Z_t^j$ for each model $j \in [m]$, the previous perturbations $\{Z_l^j\}_{l=1}^{t-1}$ do not need to be resampled.

**Remark 4.1.** The design of ensemble sampling is similar to that of perturb-history exploration (PHE)-based algorithms. In PHE-based algorithms (Kveton et al., 2020a;b), we only keep one

model, but the entire perturbation sequence $\{Z_l\}_{l=1}^t$ is freshly sampled in each round. As a result, the per-round computational cost due to sampling increases linearly in $t$, the algorithm becomes impractical for large $T$. Ensemble sampling can significantly reduce the computational cost by keeping previous perturbations, the per-round sampling cost remains constant for any $t$.

A unified algorithmic framework is given in Algorithm 1. For the generalized linear and neural bandit settings, we specify (i) a warm-up/initialization step (Line 2) and (ii) a parameter-estimation loss $L_{\text{nonlin}}$ (Line 9). The following sections detail these two algorithmic instantiations.

---

**Algorithm 1** Ensemble Sampling for Nonlinear Contextual Bandits

1: **Input:** ensemble size $m$, regularization parameter $\lambda$, reward-perturbation distribution $\mathcal{P}_R$ on $\mathbb{R}$, number of warm-up exploration rounds $\tau$
2: Warm-up for the first $\tau$ rounds and initialization          ◁ GLM-ES or Neural-ES
3: **for** $t = \tau + 1, ..., T$ **do**
4:     Sample $j_t$ uniformly from $[m]$
5:     Pull arm $X_t \leftarrow \arg\max_{X \in \mathcal{X}_t} f_t(X)$ and receive reward $Y_t$
6:     **for** $j = 1, 2, ..., m$ **do**
7:        Sample $Z_t^j \sim \mathcal{P}_R$
8:        Update $\mathcal{D}_t \leftarrow \mathcal{D}_{t-1} \cup \{(X_t, Y_t + Z_t^j)\}$
9:        $\theta_t^j \leftarrow \arg\min_\theta L_{\text{nonlin}}(\theta; \mathcal{D}_t)$          ◁ GLM-ES or Neural-ES
10:    **end for**
11: **end for**

---

### 4.2 ENSEMBLE SAMPLING FOR GENERALIZED LINEAR MODEL (GLM-ES)

We first consider the generalized linear model (GLM) and provide the algorithm design of `GLM-ES`.

**Generalized Linear Model with Regularization:** We assume that the reward $Y$ given feature vector $X$ has an exponential-family distribution with mean $\mu(X^\top \theta_{\text{GLM}}^*)$, where $\mu(\cdot)$ is the link function and $\theta_{\text{GLM}}^* \in \mathbb{R}^d$ is the (true) model parameter. Specifically, at each round $t$, the observed reward is generated by $Y_t = \mu(X_t^\top \theta_{\text{GLM}}^*) + \eta_t$, where $\eta_t$ is the $\sigma$-sub-Gaussian random noise. Detailed discussions on the exponential-family assumption are included in Appendix A. Given observed data set $\mathcal{D}_t = \{(X_l, Y_l)\}_{l=1}^t$, the $\lambda$-regularized negative log-likelihood of $\mathcal{D}_t$ under parameter $\theta$ is

$$L_{\text{GLM}}(\theta; \mathcal{D}_t) = \frac{\lambda}{2}\|\theta\|^2 - \sum_{l=1}^t \left(Y_l \cdot X_l^\top \theta - b(X_l^\top \theta)\right), \qquad \dot{b}(\cdot) = \mu(\cdot). \tag{4.1}$$

**Algorithm Design:** In `GLM-ES`, we maintain an ensemble of $m$ models (estimators), each model is parametrized by parameter $\theta_t^j, j \in [m]$, which is an estimation of $\theta_{\text{GLM}}^*$ based on perturbed history. We use the $\lambda$-regularized negative log-likelihood to obtain the parameter estimation,

$$\theta_t^j = \arg\min_{\theta \in \mathbb{R}^d} L_{\text{GLM}}\left(\theta; \{X_l, Y_l + Z_l^j\}_{l=1}^t\right), \tag{4.2}$$

where $Z_l^j \in \mathbb{R}$ are perturbations sampled from distribution $\mathcal{P}_R$. In `GLM-ES`, we also use a warm-up procedure that approximates a G-optimal design, that is,

$$\zeta = \arg\min_{\zeta \in \Delta_{\mathcal{X}}} \max_{X \in \mathcal{X}} \|X\|_{V(\zeta)^{-1}}^2, \quad \text{where } V(\zeta) = \sum_{X \in \mathcal{X}} \zeta(x) X X^\top.$$

We then sample $X_1, \ldots, X_\tau$ based on G-optimal design $\zeta$ by following the rounding procedure Algorithm 3 given in Pukelsheim (2006, Chapter 12) and detailed in Fiez et al. (2019). The complete warm-up procedure Algorithm 2 is provided in Appendix B.6.

**Remark 4.2.** The algorithm `GLM-ES` is an application of ensemble sampling in GLM settings. Compared to `Lin-ES`, a warm-up procedure Algorithm 2 is required to guarantee that optimism is satisfied with constant probability. In the generalized linear bandit literature, the number of rounds in warm-up procedure is typically chosen by enforcing a lower bound on the minimum eigenvalue of the empirical feature covariance matrix (Kveton et al., 2020b; Li et al., 2017; Liu, 2023). However,

most works do not specify the required order for the number of warm-up rounds — effectively treating it as an assumption. We propose a practical warm-up scheme that directly controls the uncertainty level (see Lemma B.5 for details). The same procedure is also used in Liu (2023). Additionally, compared to the algorithm design in Liu (2023), we removed the requirement for adapted perturbation on rewards and reduced requirements on the number of rounds in the warm-up procedure, making the algorithm simpler and more efficient.

### 4.3 ENSEMBLE SAMPLING FOR NEURAL CONTEXTUAL BANDIT (NEURAL-ES)

We now introduce neural contextual bandit setting and introduce the algorithm `Neural-ES`.

**Neural Contextual Bandit:** In the GLM setting, we assume the mean reward is given by $\mu(X^\top \theta^*)$, where the link function $\mu(\cdot)$ is known to the agent. In neural contextual bandits, we only assume that the reward model $h(X)$ is bounded as $0 \le h(\cdot) \le 1$. Since we add no assumptions to $h(\cdot)$ except that it is bounded, we need to use a deep neural network (DNN) to approximate the true relation. We adopt the following fully connected neural network $f(X; \theta)$ to approximate $h(X)$:

$$f(X; \theta) = \sqrt{N}\, W_L \phi\big(W_{L-1}\phi\big(\cdots\phi(W_1 X)\big)\big),$$

where $N$ and $L$ are the width and depth of the neural network, $\phi(x) = \text{ReLU}(x)$, $W_l$ are learnable parameter matrices, and $\theta = [\text{vec}(W_1), \cdots, \text{vec}(W_L)] \in \mathbb{R}^{d'}$ is the concatenation of all learnable parameters. The dimension of $\theta$ is $d' = N + Nd + N^2(L-1)$. For simplicity, we design the neural network such that each layer has the same width $N$.

To learn the parameters in the neural network, we define the following $\lambda$-regularized loss function:

$$L_{\text{Neural}}(\theta; \mathcal{D}_t) = \frac{1}{2}\sum_{l=1}^{t}\big(f(X_l; \theta) - Y_l\big)^2 + \frac{1}{2}\lambda N \,||\theta - \theta_0||_2^2,$$

where parameter $\theta_0$ is randomly sampled at initialization. The parameter $\theta$ is estimated using gradient descent to minimize the loss function. We use learning rate $\eta$ and number of steps $J$ in the gradient descent in neural network learning.

**Algorithm Design:** We follow the unified algorithm framework (Algorithm 1) to design `Neural-ES`. We maintain $m$ different models to approximate the true mapping $h(\cdot)$, each model is a deep neural network with the same structure. We first initialize the neural network $\theta_0 = [\text{vec}(W_1), \cdots, \text{vec}(W_L)] \in \mathbb{R}^{d'}$ using random parameters sampled from Gaussian distribution: for $1 \le l \le L-1$, $W_l = (W, 0; 0, W)$ with each entry sampled from $\mathcal{N}(0, 4/N)$; $W_L = (\mathbf{w}^\top, -\mathbf{w}^\top)$ with each entry sampled from $\mathcal{N}(0, 2/N)$. Note that the initialization $\theta_0$ is shared across all models in the ensemble.

We then perform a simple warm-up by pulling each arm once. At each round, we uniformly randomly choose one model $j_t$ from the ensemble and choose arm $X_t$ which maximizes the learned function $f(X; \theta_{t-1}^{j_t})$ and receive reward $Y_t$. We use gradient descent on the $\lambda$-regularized loss function to update the parameter estimation for each model:

$$\theta_t^j = \text{argmin}_{\theta \in \mathbb{R}^{d'}} L_{\text{Neural}}(\theta; \{X_l, Y_l + Z_l^j\}_{l=1}^{t}). \tag{4.3}$$

**Remark 4.3.** The most relevant design is Neural Bandit with Perturbed Reward (`NPR`) proposed by (Jia et al., 2022), except that our `Neural-ES` keeps an ensemble of models and updates the perturbations incrementally instead of resampling all perturbations at each round. While this design could consume more memory, different models can be updated in parallel if we distribute the ensemble in $m$ different machines. Since the computational cost of updating one model is reduced compared to `NPR`, our algorithm can still accelerate the overall training process in this setting.

## 5 THEORETICAL ANALYSIS

In this section, we provide theoretical analysis to the proposed algorithms.

## 5.1 REGRET BOUND OF GLM-ES

To analyze `GLM-ES`, we first lay down the following assumptions commonly used in the literature or generalized linear bandits. The assumption on the derivative of link function $\mu(\cdot)$ is standard in the GLM setting (Li et al., 2017; Kveton et al., 2020b), while the $M$-self-concordant assumption is recently proposed and applies to a broad class of functions (Liu et al., 2024).

**Assumption 5.1** (Link function in GLM). The reward model $Y$ follows exponential family distribution with mean $\mu(\cdot)$, and $\mu(\cdot)$ is known to the agent. The link function $\mu(\cdot)$ is *strictly increasing* and the derivative of $\mu(\cdot)$ is *bounded* as follows: $\dot{\mu}(s) > 0, \forall s \in \mathbb{R}$ and $0 < \dot{\mu}_{\min} \leq \dot{\mu}(s) \leq \dot{\mu}_{\max}$.

**Assumption 5.2** ($M$-Self-concordant). The link function $\mu(\cdot)$ is $M$-self-concordant with a constant $M > 0$ known to the agent: $|\ddot{\mu}(u)| \leq M\dot{\mu}(u), \forall u \in \mathbb{R}$.

**Remark 5.3.** We adopt the $M$-self-concordant assumption as a mild, notation-simplifying condition for exponential family models. In fact, for common reward distributions (Gaussian, exponential, Poisson, and Beta), it have been shown that Assumption 5.2 automatically holds (Liu, 2023, Table 3.1). This assumption is used to get an upper bound on $H(\theta; \mathcal{D}_t)$ in terms of $Q(\theta, \theta'; \mathcal{D}_t)$ (Lemmas A.1 and A.2). This assumption has also been used in Janz et al. (2024b, Theorem 2).

**Remark 5.4** (Removal of the regularity assumption). We use $\lambda$-regularized negative log-likelihood in (4.1). This is used to remove the additional regularity assumptions in existing papers (Li et al., 2017; Kveton et al., 2020b): (i) there exists a constant $\sigma_0 > 0$ such that $\lambda_{\min}\big(\mathbb{E}[\frac{1}{K}\sum_{a\in[K]} X_{t,a}X'_{t,a}]\big) \geq \sigma_0^2$ for all $t$, and (ii) arm context vectors $\{X_{t,a} | a \in [K]\} \subset \mathbb{R}^d$ are i.i.d. drawn. These two assumptions are usually used to guarantee $V_t$ is invertible. By deploying the $\lambda$-regularized negative log-likelihood in (4.1), our analysis does not require these assumptions.

We present the frequentist regret bound of `GLM-ES` as follows. The complete proof and expression of the regret bound is presented in Appendix A.

**Theorem 5.5** (Regret Bound for GLM-ES). Fix $\delta \in (0, 1]$. Assume $|\mathcal{X}| = K < \infty$ and run `GLM-ES` with regularization parameter $\lambda = 1 \vee (2dM/S)\log\big(e\sqrt{1 + T\dot{\mu}_{\max}/d} \vee 1/\delta\big)$, ensemble size $m = \Omega(K\log T)$, perturbation distribution $\mathcal{P}_R = \mathcal{N}(0, \sigma_R^2)$, where $\sigma_R = \Theta(d\log T)$. Then, with probability at least $1 - 4\delta$, the cumulative regret of `GLM-ES` is bounded by

$$R(T) = \widetilde{\mathcal{O}}\big(d^{\frac{3}{2}}\sqrt{T} + d^{\frac{9}{2}}\big). \tag{5.1}$$

**Remark 5.6.** When we choose $T = \mathcal{O}(d^6)$, the regret bound of `GLM-ES` becomes $\widetilde{\mathcal{O}}(d^{\frac{3}{2}}\sqrt{T})$, which matches the result of `GLM-TSL`, `GLM-FPL` (Kveton et al., 2020b), `EVILL` (Janz et al., 2024b) and `Lin-ES` (Lee & Oh, 2024), achieving a state-of-the-art theoretical guarantee for randomized exploration algorithms in the GLM setting. Compared with EVILL (Janz et al., 2024b), we also improve the number of rounds of warm-up from $d^9$ to $d^{9/2}$. We also develop novel analysis techniques to avoid the adapted perturbation requirement as in Liu (2023), where the distribution of perturbation changes in each round. This makes our algorithm more efficient and easier to implement.

## 5.2 REGRET BOUND OF NEURAL-ES

For the regret analysis of `Neural-ES`, we first introduce the following notations. A detailed introduction to neural tangent kernel is provided in (Jacot et al., 2018), here we only list the important notations for our theoretical results. We use $\mathbf{H}$ to denote the neural tangent kernel (NTK) matrix defined on the context set $\mathcal{X}$ and $\mathbf{h} = (h(X_1), ..., h(X_K))$. Then, we use notation $S_{\text{Neural}}$ as the upper bound of $\sqrt{2\mathbf{h}^\top \mathbf{H}^{-1}\mathbf{h}}$. The effective dimension $\widetilde{d}$ of the neural NTK matrix is defined as

$$\widetilde{d} = \frac{\log\det(I + TH/\lambda)}{\log(1 + TK/\lambda)}. \tag{5.2}$$

Now we present the frequentist regret bound of `Neural-ES` as follows. The complete proof and expression of the regret bound is presented in Appendix C.

**Theorem 5.7** (Regret Bound for Neural-ES). Let $\widetilde{d}$ be the effective dimension of the neural tangent kernel matrix. Assume $|\mathcal{X}| = K < \infty$ and run `Neural-ES` with ensemble size $m = \Omega(K\log T)$, regularization parameter $\lambda \geq \max\{1, S_{\text{Neural}}^{-2}\}$, perturbation distribution $\mathcal{P}_R = \mathcal{N}(0, \sigma_R^2)$ with $\sigma_R$ given by (C.6). Then, the cumulative regret of `Neural-ES` is bounded by

$$\mathbb{E}\big[R(T)\big] = \widetilde{\mathcal{O}}(\widetilde{d}\sqrt{T}). \tag{5.3}$$

**Remark 5.8.** The regret bound of `Neural-ES` matches the order of `Neural-PHE` (Jia et al., 2022), `Neural-TS` (Zhang et al., 2021) and `Neural-UCB` (Zhou et al., 2020), achieving the state-of-the-art theoretical guarantee in the neural bandit setting. Compared to `Neural-PHE`, the computational cost per model is considerably reduced because we do not resample all perturbations in each round. Compared to `Neural-TS` and `Neural-UCB`, we do not need to construct high-probability confidence sets for exploration, which is relatively difficult to implement and typically involves very high computational cost.

## 6 EXTENSION TO ANYTIME ALGORITHMS

In this section, we present how to use doubling trick (Besson & Kaufmann, 2018) to extend ensemble sampling into anytime algorithms while keeping the asymptotic behavior of regret bound. Here we present the algorithm design and theoretical guarantee of anytime versions of ensemble sampling, the analysis of regret bound are presented in Appendix E.

To apply doubling trick, we choose a sequence of time steps $\{T_i\} = \{T_0, T_1, T_2, ...\}$ and fully restart the original (non-anytime) algorithm when we reach $t = T_i + 1$. Therefore, after each reset, the algorithm runs from $T_i + 1$ until $\min\{T_{i+1}, T\}$, and we can initialize the $T$-dependent parameters for $\tau_i = T_{i+1} - T_i$ rounds. The number of rounds follows the sequence:

$$\{\tau_i\} = \{T_0, T_1 - T_0, T_2 - T_1, ...\}.$$

The doubling trick approach treats the original non-anytime algorithm as a black box, thus we can easily extend both `GLM-ES` and `Neural-ES` into anytime algorithms using the same method.

Our main theory of anytime versions of ensemble sampling is as follows.

**Theorem 6.1** (Regret Bound of Doubling Trick). Set sequence $T_i = \lfloor T_0 b^i \rfloor$, where $T_0 \geq 100$ and $b = (3 + \sqrt{5})/2 \approx 2.6$. Use $R(T, \delta)$ to denote the regret bound of non-anytime algorithms `GLM-ES` or `Neural-ES` that holds with probability at least $1 - \delta$, and $R^{DT}(T, \delta)$ to denote its corresponding anytime version using doubling trick. Then, by applying doubling trick for `GLM-ES` and `Neural-ES`, with probability at least $1 - \delta$, the cumulative regret is bounded by

$$R^{DT}(T, \delta) \leq 3.3\, R(T, \delta).$$

**Remark 6.2.** By directly applying doubling trick, we obtain the same asymptotic behavior of regret bound. By properly setting the sequence $\{T_i\}$, extending ensemble sampling to anytime algorithm comes with the cost of only a constant factor. This constant factor is determined by the parameters $T_0$ and $b$, the choice of these parameters can considerably affect the empirical performance.

## 7 EXPERIMENTS

We conduct experiments to demonstrate the practicality of ensemble sampling and its anytime variants in bandit settings. We use linear bandit environment to test `Lin-ES`, logistic bandit to test `GLM-ES`, then distance bandit and quadratic form bandit to test `Neural-ES`. Our result shows that ensemble sampling can give competitive cumulative regret with reduced computational cost.

### 7.1 ENVIRONMENT SETUP

**Linear Bandit** We assess the empirical performance of `Lin-ES` using linear bandit environment, where the reward is generated by $Y_t = X_t^\top \theta^* + \eta_t$. Specifically, we set the number of arms $K = 50$, dimension of feature vector $d = 20$ and total steps $T = 10^4$. The noise of reward are generated from Gaussian distribution $\eta_t \sim \mathcal{N}(0, \sigma^2)$ with $\sigma = 0.5$. In the `Lin-ES` algorithm, we set ensemble size $m = 25$, regularization $\lambda = 1.0$, and perturbation distribution $\mathcal{N}(0, \sigma_R^2)$ with $\sigma_R = 0.1$. There is no warm-up procedure in `Lin-ES`.

**Generalized Linear Bandit** We assess the empirical performance of `GLM-ES` using logistic bandit environment, where the link function is given by $Y_t = 1/(1 + \exp(-X_t^\top \theta^*)) + \eta_t$. We set the number of arms $K = 50$, dimension of feature vector $d = 20$, total steps $T = 10^4$. The noise of reward are generated from Gaussian distribution $\eta_t \sim \mathcal{N}(0, \sigma^2)$ with $\sigma = 0.5$.

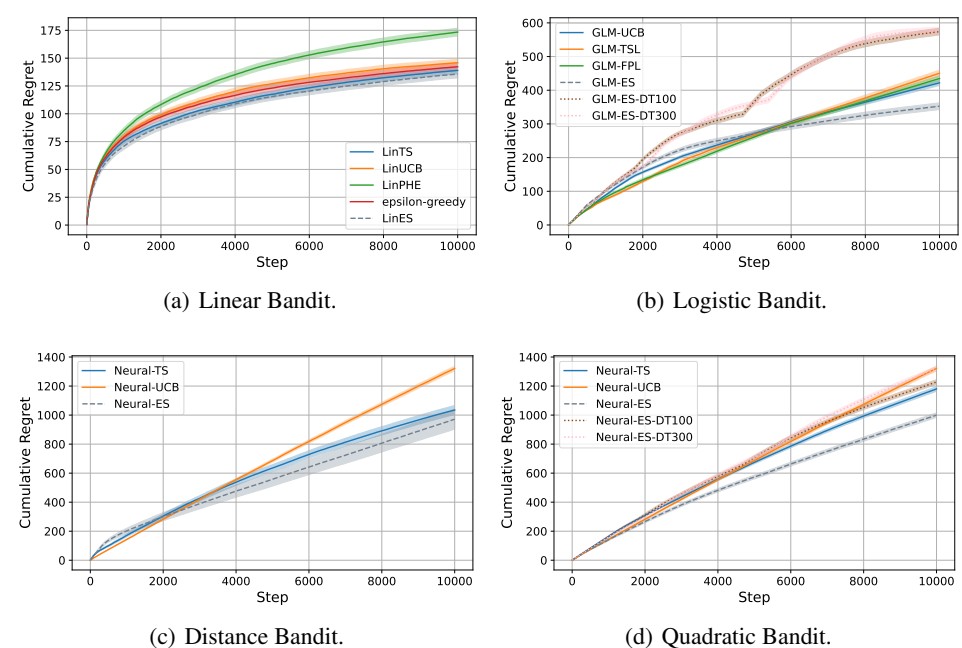

(a) Linear Bandit.

(b) Logistic Bandit.

(c) Distance Bandit.

(d) Quadratic Bandit.

Figure 1: Experiment results in various bandit settings.

**Neural Bandit** We test `Neural-ES` on two nonlinear reward models: (1) distance bandit: $h_1(X) = -||X - \theta^*||_2$; and (2) quadratic bandit: $h_2(X) = 10^{-2}(X^\top AA^\top X)$, where $A$ is a $d \times d$ matrix with each entry randomly sampled from $\mathcal{N}(0, 1)$. In the experiments, we set number of arms $K = 50$, dimension of feature vector $d = 20$ and total steps $T = 10^4$. The noise of reward are generated from Gaussian distribution $\eta_t \sim \mathcal{N}(0, \sigma^2)$ with $\sigma = 0.5$.

## 7.2 IMPLEMENTATIONS OF ALGORITHMS

**GLM-ES** We implement `GLM-ES` assuming that the link function $\mu(\cdot)$ is known to the agent. The parameter $\theta$ is learned through gradient descent of the $\lambda$-regularized negative likelihood. Specifically, we use 100 iterations of gradient descent with step size 0.01. In the non-doubling-trick experiment, we set ensemble size $m = 10$, perturbation distribution $\mathcal{N}(0, \sigma_R^2)$ with $\sigma_R = 0.1$, warm-up steps $\tau = 500$ and regularization $\lambda = 1.0$. In the experiment with doubling trick, for number of rounds $\tau_i$, we set ensemble size $m = 2 \times \log\tau_i$, perturbation distribution $\mathcal{N}(0, \sigma_R^2)$ with $\sigma_R = 0.02 \times \log\tau_i$, warm-up steps $\tau = 500$ and regularization $\lambda = 1.0$.

**Neural-ES** We implement the fully connected neural network $f(X; \theta)$ using $L = 3$ layers, the width is set to $N = 20$ for each layer and we use ReLU as the activation function. The network structure is the same for `Neural-ES`, `Neural-TS` and `Neural-UCB`. We optimize the loss function using gradient descent with 100 steps and learning rate 0.01. Similar to `GLM-ES`, in the non-doubling-trick experiment, we set ensemble size $m = 10$, perturbation distribution $\mathcal{N}(0, \sigma_R^2)$ with $\sigma_R = 0.1$, warm-up steps $\tau = 50$ and regularization $\lambda = 1.0$. In the experiment with doubling trick, for number of rounds $\tau_i$, we set ensemble size $m = 2 \times \log\tau_i$, perturbation distribution $\mathcal{N}(0, \sigma_R^2)$ with $\sigma_R = 0.02 \times \log\tau_i$, warm-up steps $\tau = 50$ and regularization $\lambda = 1.0$.

## 7.3 RESULTS AND DISCUSSIONS

The empirical results are plotted in Figure 1, The cumulative regret of ensemble sampling are plotted in gray dashed lines, the results of ensemble sampling with doubling trick are also plotted in dashed lines. Specifically, "DT100" and "DT300" indicate that we choose $T_0 = 100$ and $T_0 = 300$ in doubling trick, respectively. We set $b = (3 + \sqrt{5})/2 \approx 2.6$ for all doubling trick simulations. All numerical results are averaged over 10 problem instances.

According to simulation results, the cumulative regret of ensemble sampling is competitive compared to baseline algorithms in both linear and nonlinear bandit settings, the design of keeping previous perturbations also considerably reduces per-model computational cost at large time step $t$. Additionally, the doubling trick variants of ensemble sampling demonstrates its practicality by giving competitive cumulative regret while removing the requirement of total rounds $T$ prior to running the algorithm. Overall, our experiment result shows that ensemble sampling is practical in both linear and nonlinear bandit settings, its cumulative regret is similar to or outperforms baseline models in different bandit environments.

## 8  CONCLUSION

In this work, we studied ensemble sampling in nonlinear contextual bandit settings. We proposed a general framework of algorithm design for ensemble sampling in bandit problems, then discussed two realizations `GLM-ES` and `Neural-ES` for generalized linear bandit and neural bandit, respectively. We proved high-probability regret bound $\widetilde{\mathcal{O}}(d^{3/2}\sqrt{T} + d^{9/2})$ for `GLM-ES` and $\widetilde{\mathcal{O}}(\widetilde{d}\sqrt{T})$ for `Neural-ES`, both match the state-of-the-art result of randomized exploration algorithms in bandit problems. We used synthetic bandit environments to evaluate the performance of the proposed algorithms in terms of cumulative regret and computational cost. The empirical results demonstrate that ensemble sampling and its anytime variants can achieve competitive cumulative regret with considerably reduced computational cost. Our work establishes ensemble sampling as a provable and practical algorithm framework in bandit problems.

## REPRODUCIBILITY STATEMENT

We provide detailed descriptions of all methodologies used in this paper, including algorithm design and implementation, theoretical guarantee of proposed ensemble sampling algorithms and experiment instances. For theoretical results in this work, we provide detailed proof in the Appendix. For experiment results in this work, we provide the source code as supplementary materials including the implementations of the bandit environment and algorithms. With the parameters listed in Section 7, our numerical results are reproducible.

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

# A    PROOF OF REGRET BOUND OF GLM-ES

## A.1    PRELIMINARY ANALYSIS

In this section, we present basic technical lemmas directly from the definition and assumptions of the GLM setting. The technical lemmas and notations introduced in this section are fundamental and will be extensively utilized in the following analysis.

In the GLM setting with exponential family distribution assumption, given feature vector $X \in \mathbb{R}^d$, the conditional distribution of reward $Y$ is given by

$$p(Y|X) = b_0(Y) \exp\big[Y \cdot X^\top \theta - b(X^\top \theta)\big],$$

where $\theta \in \mathbb{R}^d$ is a fixed parameter for a given model. We define the link function $\mu(\cdot)$ as

$$\mu(\cdot) = \dot{b}(\cdot),$$

the expected value of $Y$ is given by $\mu(X^\top \theta)$. We make additional assumptions that the link function $\mu(\cdot)$ is *strictly increasing* and the derivative of $\mu$ is *bounded* as follows:

$$\dot{\mu}(s) > 0, \quad \forall s \in \mathbb{R}; \qquad 0 < \dot{\mu}_{\min} \le \dot{\mu}(s) \le \dot{\mu}_{\max}.$$

Using the mean value theorem, for each pair $s_1 < s_2$, we have

$$\mu(s_2) - \mu(s_1) = \dot{\mu}(s_0)(s_2 - s_1), \quad s_0 \in (s_1, s_2).$$

Therefore, we have that

$$\dot{\mu}_{\min}(s_2 - s_1) \le \mu(s_2) - \mu(s_1) \le \dot{\mu}_{\max}(s_2 - s_1).$$

We add regularization to the algorithm design for the analysis of the warm-up procedure. Given input data set $\mathcal{D}_t = \{(X_l, Y_l)\}_{l=1}^t$, we define the following $\lambda$-regularized negative log-likelihood:

$$L_{\mathrm{GLM}}(\theta; \mathcal{D}_t) = \frac{\lambda}{2}\|\theta\|^2 - \sum_{l=1}^t \big(Y_l \cdot X_l^\top \theta - b(X_l^\top \theta)\big). \tag{A.1}$$

Then, for time step $t$, we can define the (regularized) MLE as follows:

$$\bar{\theta}_t = \operatorname*{argmin}_{\theta \in \mathbb{R}^d} L_{\mathrm{GLM}}(\theta; \mathcal{D}_t). \tag{A.2}$$

Based on (A.1), we can compute the gradient

$$\nabla_\theta L_{\mathrm{GLM}}(\theta; \mathcal{D}_t) = \sum_{l=1}^t \mu(X_l^\top \theta)X_l + \lambda\theta - \sum_{l=1}^t X_l Y_l = f_t(\theta) - \sum_{l=1}^t X_l Y_l,$$

where we define $f_t(\theta)$ as

$$f_t(\theta) := \sum_{l=1}^t \mu(X_l^\top \theta)X_l + \lambda\theta. \tag{A.3}$$

Note that $\bar{\theta}_t = \operatorname{argmin}_{\theta \in \mathbb{R}^d} L_{\mathrm{GLM}}(\theta; \mathcal{D}_t)$, thus $\nabla_\theta L_{\mathrm{GLM}}(\bar{\theta}_t; \mathcal{D}_t) = 0$, we have

$$f_t(\bar{\theta}_t) = \sum_{l=1}^t X_l Y_l.$$

We further define the Hessian matrix as follows

$$H(\theta; \mathcal{D}_t) = \nabla_\theta^2 L_{\mathrm{GLM}}(\theta; \mathcal{D}_t) = \sum_{l=1}^t \dot{\mu}(X_l^\top \theta)X_l X_l^\top + \lambda I. \tag{A.4}$$

For simplicity of notation, we define the Hessian matrix at true parameter $\theta^*$ and MLE $\bar{\theta}_t$ as

$$H_t = H(\theta^*; \mathcal{D}_t), \quad \bar{H}_t = H(\bar{\theta}_t; \mathcal{D}_t).$$

Next, we introduce a "secant approximation" of Hessian matrix $H(\theta; \mathcal{D}_t)$. We define the secant function as follows:

$$\alpha(u, u') = \begin{cases} \dot{\mu}(u) & \text{if } u = u', \\ \frac{\mu(u) - \mu(u')}{u - u'} & \text{otherwise.} \end{cases}$$

Note that $\lim_{u \to u'} \alpha(u, u') = \dot{\mu}(u)$. Then we give the following approximation of $H(\theta; \mathcal{D}_t)$,

$$Q(\theta, \theta'; \mathcal{D}_t) = \sum_{l=1}^{t} \alpha\big(X_l^\top \theta, X_l^\top \theta'\big) X_l X_l^\top + \lambda I.$$

Note that $Q(\theta, \theta'; \mathcal{D}_t) \to H(\theta; \mathcal{D}_t)$ as $\theta \to \theta'$.

Then, we can obtain the following technical lemma that provides upper and lower bounds for the secant function $\alpha(u, u')$.

**Lemma A.1** (Corollary 2 in Sun & Tran-Dinh (2019)). Given that $\mu(\cdot)$ is $M$-self-concordant, for $u, u' \in \mathbb{R}$, we have

$$\dot{\mu}(u) h_s(-M|u - u'|) \leq \alpha(u, u') \leq \dot{\mu}(u) h_s(M|u - u'|), \tag{A.5}$$

where we define

$$h_s(x) = \begin{cases} \frac{e^x - 1}{x} & \text{if } x \neq 0, \\ 1 & x = 0. \end{cases}$$

Based on Lemma A.1, we can obtain the following upper bound on $H(\theta; \mathcal{D}_t)$ in terms of $Q(\theta, \theta'; \mathcal{D}_t)$:

**Lemma A.2** (Lemma 3.9 in Liu (2023)). For any $\theta, \theta' \in \mathbb{R}^d$, we have

$$H(\theta; \mathcal{D}_t) \preceq (1 + MD(\theta - \theta')) Q(\theta, \theta'; \mathcal{D}_t) = (1 + MD(\theta - \theta')) Q(\theta', \theta; \mathcal{D}_t), \tag{A.6}$$

where $D(v)$ is defined as $D(v) = \max_{X \in \mathcal{X}} |X^\top v|$ for input vector $v \in \mathbb{R}^d$.

From the definition of $D(v)$, we note the following inequality:

$$D(v) = \max_{X \in \mathcal{X}} |X^\top v| \leq \max_{X \in \mathcal{X}} ||X||_2 ||v||_2 \leq ||v||_2,$$

where we used the assumption $||X||_2 \leq 1$. This inequality is very useful in the following analysis. By definition, we can directly obtain the following relation, which is critical in the later analysis:

$$f_t(\theta) - f_t(\theta') = Q\big(\theta, \theta'; \mathcal{D}_t\big)(\theta - \theta') = Q\big(\theta', \theta; \mathcal{D}_t\big)(\theta - \theta'). \tag{A.7}$$

According to this relation, finding upper bound of $||\theta - \theta'||$ can be decomposed to bounding $||f_t(\theta) - f_t(\theta')||$ and $Q\big(\theta, \theta'; \mathcal{D}_t\big)$, which provides convenience in the following analysis.

We also define an auxiliary matrix $G_t$ as follows:

$$G_t = \sum_{l=1}^{t} X_l X_l^\top + \frac{\lambda}{\dot{\mu}_{\min}} I.$$

Since we assumed that $0 < \dot{\mu}_{\min} \leq \dot{\mu}(\cdot) \leq \dot{\mu}_{\max}$, we have

$$\dot{\mu}_{\min} G_t \preceq H(\theta; \mathcal{D}_t) \preceq \dot{\mu}_{\max} G_t, \quad \forall \theta \in \mathbb{R}^d. \tag{A.8}$$

## A.2 TECHNICAL LEMMAS

In this section, we list the main technical lemmas required in the analysis of regret bound, these technical lemmas are proved in Appendix B. The main idea of analyzing GLM is that we use the lower and upper bound of $\dot{\mu}(\cdot)$ to find similarities between GLM and linear bandit. Despite the similarities of linear bandit and generalized linear bandit settings, the GLM setting is considerably more challenging to analyze due to the fact that we do not have closed form solution for parameter

estimate $\bar{\theta}_t$ and $\theta_t^j$. In our analysis, we introduce the $M$-concordant assumption with novel analysis tools to overcome these difficulties.

We list the concentration results of the estimated parameters $\theta_t^j$ as follows. Recall that we use $\theta^*$ for the true parameter of the generalized linear model, $\bar{\theta}_t$ for the MLE and $\theta_t^j$ for the estimate of model $j \in [m]$ in the ensemble at the end of round $t \in [T]$, and $\theta_t = \theta_{t-1}^{j_t}$ for the chosen parameter at round $t \in [T]$.

**Lemma A.3** (Concentration of MLE). Fix $\delta \in (0, 1]$. Define the following parameters:

$$\gamma_t(\delta, \lambda) = \sqrt{\lambda}\left(\frac{1}{2M} + S\right) + \frac{4M}{\sqrt{\lambda}}\left(d + \frac{d}{2}\log\left(1 + \frac{t\dot{\mu}_{\max}}{d\lambda}\right) + \log\frac{1}{\delta}\right),$$

$$\beta_t(\delta, \lambda) = M\frac{\gamma_t(\delta, \lambda)^2}{\lambda} + \sqrt{\frac{\gamma_t(\delta, \lambda)^2}{\lambda}}.$$

For $t \in \mathbb{N}$, define a sequence of events

$$\bar{\mathcal{E}}_t := \left\{\|\bar{\theta}_t - \theta^*\|_{H_t} \leq (1 + M\beta_T(\delta, \lambda))\gamma_T(\delta, \lambda)\right\},$$

and their intersection $\bar{\mathcal{E}} = \cap_{t=1}^T \bar{\mathcal{E}}_t$. Then, we have $\mathbb{P}(\bar{\mathcal{E}}) \geq 1 - \delta$.

**Lemma A.4** (Concentration of Perturbation). Fix $\delta \in (0, 1]$ and $t \in [T]$. Define the following parameter:

$$\widetilde{\gamma}_t(\delta, \lambda) = \sqrt{d\log\left(1 + \frac{t\dot{\mu}_{\min}}{d\lambda}\right) + 2\log\frac{T}{\delta}}.$$

For each round $t \in \mathbb{N}$, define the following event

$$\widetilde{\mathcal{E}}_{1,t} := \left\{\left\|\theta_{t-1}^{j_t} - \bar{\theta}_{t-1}\right\|_{H_{t-1}} \leq \dot{\mu}_{\max}^{1/2}\dot{\mu}_{\min}^{-1}\sigma_R\widetilde{\gamma}_T(\delta, \lambda)\right\}.$$

Then, $\widetilde{\mathcal{E}}_{1,t}$ holds with probability at least $1 - \delta/T$.

Combining the concentration results Lemma A.3 and Lemma A.4, we have the following result of the concentration of $\theta_{t-1}^{j_t}$.

**Lemma A.5** (Concentration of Estimation $\theta_{t-1}^{j_t}$). Define a sequence of events as

$$\mathcal{E}_{1,t} := \left\{\left\|\theta_{t-1}^{j_t} - \theta^*\right\|_{H_{t-1}} \leq \gamma\right\}, \quad \text{where } \gamma = (1 + M\beta_T)\gamma_T(\delta, \lambda) + \dot{\mu}_{\max}^{1/2}\dot{\mu}_{\min}^{-1}\sigma_R\widetilde{\gamma}_T(\delta, \lambda),$$

and their intersections $\mathcal{E}_1 = \cap_{t=1}^T \mathcal{E}_{1,t}$. Then, $\mathcal{E}_1$ holds with probability at least $1 - 2\delta$.

Next, the following lemma demonstrates that optimism holds with constant probability in GLM-ES.

**Lemma A.6** (Optimism). Let constant $p_N = 1 - \Phi(1) \approx 0.16$, and a sequence of events defined as

$$\mathcal{E}_{2,t} = \left\{\mathbb{P}\left(X^{*\top}\theta^* \leq X_t^\top\theta_{t-1}^{j_t} \text{ and } \widetilde{\mathcal{E}}_{1,t} \mid \mathcal{F}_{t-1}\right) \geq p_N/4\right\},$$

and their intersection $\mathcal{E}_2 = \cap_{t=1}^T \mathcal{E}_{2,t}$. Then, $\mathcal{E}_2$ holds with probability at least $1 - \delta$.

We need the following lemma to introduce the notations of $\theta_t^-$ and $X_t^-$, which will be used in Appendix A.3.

**Lemma A.7.** Assume the conditions of Theorem 5.5. Let $J(\theta) = \sup_{X \in \mathcal{X}} \mu(X^\top\theta)$, where $\theta \in \mathbb{R}^d$. Let $\Theta_t = \{\theta \in \mathbb{R}^d \mid \|\theta - \theta^*\|_{H_{t-1}} \leq \gamma\}$. Define $\theta_t^- = \operatorname{argmin}_{\theta \in \Theta_t} J(\theta)$ and $X_t^- = \operatorname{argmax}_{X \in \mathcal{X}} \mu(X^\top\theta_t^-)$. For any $\theta \in \mathbb{R}^d$ and an event $\mathcal{E}'$, we introduce the following notation:

$$g_t(\theta, \mathcal{E}') = (J(\theta) - J(\theta_t^-))\mathbb{1}\{\mathcal{E}'\}.$$

Then, $g_t(\theta^*, \mathcal{E}') \geq 0$ holds for any event $\mathcal{E}' \in \mathcal{F}$, and $g_t(\theta_t, \mathcal{E}'') \geq 0$ holds almost surely for any event such that $\mathcal{E}'' \subset \mathcal{E}_{1,t}$.

Finally, we need the following lemma to calculate the summation in Appendix A.3.

**Lemma A.8** (Lemma 2 in Li et al. (2017)). Let $\{X_t\}_{t=1}^\infty$ be a sequence in $\mathbb{R}^d$ satisfying $\|X_t\| \leq 1$. Define $X_0 = \mathbf{0}$ and $G_{t-1} = \lambda I + \sum_{i=0}^{t-1} X_i X_i^\top$. Suppose there is an integer $n$ such that $\lambda_{\min}(G_n) \geq 1$, then for all $T > n$,

$$\sum_{t=n+1}^T \|X_t\|_{G_{t-1}^{-1}} \leq \sqrt{2(T-n)d\log\left(1 + \frac{T}{d\lambda}\right)}.$$

### A.3 REGRET ANALYSIS

With the technical lemmas listed above, we prove Theorem 5.5 as follows.

*Proof of Theorem 5.5.* Recall that the cumulative regret is defined as

$$R(T) = \sum_{t=1}^{T} \left( \mu(x^{*\top}\theta^*) - \mu(X_t^\top \theta^*) \right).$$

In the warm-up procedure, the regret for each round is bounded as

$$\Delta_{\max} = \sup_{x \in \mathcal{X}} \mu(x^\top \theta^*) - \inf_{x \in \mathcal{X}} \mu(x^\top \theta^*).$$

Therefore, we have

$$R(T) \leq \tau \Delta_{\max} + \sum_{t=\tau+1}^{T} \left( \left( \mu(x^{*\top}\theta^*) - \mu(X_t^\top \theta^*) \right) \right).$$

We only consider $t \in [\tau + 1, T]$ in the following discussions.

We first bound the instantaneous regret at time $t$. Define event $\mathcal{E}$ as $\mathcal{E} = \cap_{t=1}^{T} (\mathcal{E}_{1,t} \cap \mathcal{E}_{2,t})$, we discuss the per-round regret under event $\mathcal{E}$.

$$\left( \mu(x^{*\top}\theta^*) - \mu(X_t^\top \theta^*) \right) \mathbb{1}\{\mathcal{E}\} = \underbrace{\left( \mu(x^{*\top}\theta^*) - \mu(X_t^\top \theta_t) \right) \mathbb{1}\{\mathcal{E}\}}_{I_1} + \underbrace{\left( \mu(X_t^\top \theta_t) - \mu(X_t^\top \theta^*) \right) \mathbb{1}\{\mathcal{E}\}}_{I_2}.$$

From the expression, $I_1$ is related to optimism, $I_2$ is related to concentration of $\theta_t$. $I_2$ is directly bounded under $\mathcal{E}$:

$$\begin{aligned}
I_2 &= \left( \mu(X_t^\top \theta_t) - \mu(X_t^\top \theta^*) \right) \mathbb{1}\{\mathcal{E}\} \\
&\leq \dot{\mu}_{\max} \left| X_t^\top \theta_t - X_t^\top \theta^* \right| \mathbb{1}\{\mathcal{E}\} \\
&\leq \dot{\mu}_{\max} \|X_t\|_{H_t^{-1}} \|\theta_t - \theta^*\|_{H_t} \mathbb{1}\{\mathcal{E}\} \\
&\leq \dot{\mu}_{\max} \gamma \|X_t\|_{H_t^{-1}}.
\end{aligned}$$

Now we consider $I_1$. Using the definitions in Lemma A.7, we have

$$\begin{aligned}
I_1 &= \left( \mu(x^{*\top}\theta^*) - \mu(X_t^\top \theta_t) \right) \mathbb{1}\{\mathcal{E}\} \\
&= \left( \mu(x^{*\top}\theta^*) - \mu(X_t^{-\top}\theta_t^-) \right) \mathbb{1}\{\mathcal{E}\} + \left( \mu(X_t^{-\top}\theta_t^-) - \mu(X_t^\top \theta_t) \right) \mathbb{1}\{\mathcal{E}\} \\
&= \left( J(\theta^*) - J(\theta_t^-) \right) \mathbb{1}\{\mathcal{E}\} + \left( J(\theta_t^-) - J(\theta_t) \right) \mathbb{1}\{\mathcal{E}\} \\
&= g_t(\theta^*, \mathcal{E}) - g_t(\theta_t, \mathcal{E}) \\
&\leq g_t(\theta^*, \mathcal{E}) \leq g_t(\theta^*, \mathcal{E}_{2,t}).
\end{aligned}$$

We use Markov's inequality to bound $g_t(\theta^*, \mathcal{E}_{2,t})$:

$$g_t(\theta^*, \mathcal{E}_{2,t}) \mathbb{P}\left( g_t(\theta_t, \mathcal{E}_{1,t} \cap \mathcal{E}_{2,t}) \geq g_t(\theta^*, \mathcal{E}_{2,t}) \mid \mathcal{F}_{t-1} \right) \leq \mathbb{E}\left[ g_t(\theta_t, \mathcal{E}_{1,t} \cap \mathcal{E}_{2,t}) \mid \mathcal{F}_{t-1} \right].$$

Now we need the lower bound for $\mathbb{P}\left( g_t(\theta_t, \mathcal{E}_{1,t} \cap \mathcal{E}_{2,t}) \geq g_t(\theta^*, \mathcal{E}_{2,t}) \mid \mathcal{F}_{t-1} \right)$ (we relate this to the probability of optimism) and the upper bound for $g_t(\theta_t, \mathcal{E}_{1,t} \cap \mathcal{E}_{2,t})$.

Using the assumption that $\mu(x)$ is strictly increasing, under the event $\left( x^{*\top}\theta^* \leq X_t^\top \theta_t \right) \wedge \mathcal{E}_{1,t} \wedge \mathcal{E}_{2,t}$, we have the following results:

$$\begin{aligned}
x^{*\top}\theta^* \leq X_t^\top \theta_t &\Leftrightarrow \mu(x^{*\top}\theta^*) - \mu(X_t^{-\top}\theta_t^-) \leq \mu(X_t^\top \theta_t) - \mu(X_t^{-\top}\theta_t^-) \\
&\Leftrightarrow \left( \mu(x^{*\top}\theta^*) - \mu(X_t^{-\top}\theta_t^-) \right) \mathbb{1}\{\mathcal{E}_{2,t}\} \leq \left( \mu(X_t^\top \theta_t) - \mu(X_t^{-\top}\theta_t^-) \right) \mathbb{1}\{\mathcal{E}_{1,t} \cap \mathcal{E}_{2,t}\} \\
&\Leftrightarrow g_t(\theta^*, \mathcal{E}_{2,t}) \leq g_t(\theta_t, \mathcal{E}_{1,t} \cap \mathcal{E}_{2,t}).
\end{aligned}$$

Therefore, we have

$$\mathbb{P}\Big(g_t\big(\theta_t, \mathcal{E}_{1,t} \cap \mathcal{E}_{2,t}\big) \geq g_t(\theta^*, \mathcal{E}_{2,t}) \mid \mathcal{F}_{t-1}\Big) \geq p_N/4.$$

Next, we consider the upper bound of $g_t(\theta_t, \mathcal{E}_{1,t} \cap \mathcal{E}_{2,t})$. Using the definition, we have

$$\begin{aligned}
g_t(\theta_t, \mathcal{E}_{1,t} \cap \mathcal{E}_{2,t}) &= \big(\mu(X_t^\top \theta_t) - \mu(X_t^{-\top} \theta_t^-)\big)\, \mathbb{1}\{\mathcal{E}_{1,t} \cap \mathcal{E}_{2,t}\} \\
&\leq \big(\mu(X_t^\top \theta_t) - \mu(X_t^\top \theta_t^-)\big)\, \mathbb{1}\{\mathcal{E}_{1,t} \cap \mathcal{E}_{2,t}\} \\
&\leq \dot{\mu}_{\max}\big|X_t^\top \theta_t - X_t^\top \theta_t^-\big|\, \mathbb{1}\{\mathcal{E}_{1,t} \cap \mathcal{E}_{2,t}\} \\
&\leq \dot{\mu}_{\max}||X_t||_{H_{t-1}^{-1}}||\theta_t - \theta_t^-||_{H_{t-1}}\, \mathbb{1}\{\mathcal{E}_{1,t} \cap \mathcal{E}_{2,t}\} \\
&\leq 2\gamma \dot{\mu}_{\max}||X_t||_{H_{t-1}^{-1}}.
\end{aligned}$$

Using the results above, the instantaneous regret at time $t \in [\tau+1, T]$ is bounded as:

$$\begin{aligned}
&\Big(\mu(x^{*\top}\theta^*) - \mu(X_t^\top \theta^*)\Big)\, \mathbb{1}\{\mathcal{E}\} \\
&\leq \gamma \dot{\mu}_{\max}||X_t||_{H_{t-1}^{-1}} + \frac{8\gamma\dot{\mu}_{\max}}{p_N}\mathbb{E}\Big[||X_t||_{H_{t-1}^{-1}} \mid \mathcal{F}_{t-1}\Big] \\
&= \gamma \dot{\mu}_{\max}\Big(1 + \frac{8}{p_N}\Big)||X_t||_{H_{t-1}^{-1}} + \frac{8\gamma\dot{\mu}_{\max}}{p_N}\Big(\mathbb{E}\Big[||X_t||_{H_{t-1}^{-1}} \mid \mathcal{F}_{t-1}\Big] - ||X_t||_{H_{t-1}^{-1}}\Big).
\end{aligned}$$

The cumulative regret is bounded as:

$$\begin{aligned}
R(T)\, \mathbb{1}\{\mathcal{E}\} \leq &\,\tau \Delta_{\max} + \gamma \dot{\mu}_{\max}\Big(1 + \frac{8}{p_N}\Big) \sum_{t=\tau+1}^{T} ||X_t||_{H_{t-1}^{-1}} \\
&+ \frac{8\gamma\dot{\mu}_{\max}}{p_N} \sum_{t=\tau+1}^{T} \Big(\mathbb{E}\Big[||X_t||_{H_{t-1}^{-1}} \mid \mathcal{F}_{t-1}\Big] - ||X_t||_{H_{t-1}^{-1}}\Big).
\end{aligned} \tag{A.9}$$

Next, we calculate the two summations in (A.9). From the fact that $\dot{\mu}_{\min} G_t \preceq H(\theta; \mathcal{D}_t) \preceq \dot{\mu}_{\max} G_t$, for any $x \in \mathbb{R}^d$, we have

$$\frac{1}{\sqrt{\dot{\mu}_{\max}}}||x||_{G_t^{-1}} \leq ||x||_{H_t^{-1}} \leq \frac{1}{\sqrt{\dot{\mu}_{\min}}}||x||_{G_t^{-1}}.$$

Therefore, the first summation in (A.9) is bounded by

$$\sum_{t=\tau+1}^{T} ||X_t||_{H_{t-1}^{-1}} \leq \frac{1}{\sqrt{\dot{\mu}_{\min}}} \sum_{t=\tau+1}^{T} ||X_t||_{G_{t-1}^{-1}}.$$

From the fact that $\bar{H}_t \succeq \lambda I$, we have $\lambda_{\min}(\bar{H}_t) \geq \lambda$. The first summation in (A.9) is bounded as follows by using Lemma A.8:

$$\sum_{t=\tau+1}^{T} ||X_t||_{G_{t-1}^{-1}} \leq \sqrt{2(T-\tau)d \log\Big(1 + \frac{T\dot{\mu}_{\min}}{d\lambda}\Big)}.$$

Now we consider the second summation in (A.9). Note that

$$0 \leq ||X_t||_{H_{t-1}^{-1}} \leq \sqrt{\lambda_{\max}(H_{t-1}^{-1})}||X_t||_2 \leq 1,$$

according to Azuma-Hoeffding inequality, we can bound the second summation in (A.9) by

$$\sum_{t=\tau+1}^{T} \Big(\mathbb{E}\Big[||X_t||_{H_{t-1}^{-1}} \mid \mathcal{F}_{t-1}\Big] - ||X_t||_{H_{t-1}^{-1}}\Big) \leq \sqrt{\frac{T-\tau}{2}\log\frac{1}{\delta}},$$

with probability at least $1 - \delta$. Combining these results, with probability at least $1 - 4\delta$, we have

$$\begin{aligned}
&R(T)\, \mathbb{1}\{\mathcal{E}\} \\
&\leq \gamma \frac{\dot{\mu}_{\max}}{\sqrt{\dot{\mu}_{\min}}}\Big(1 + \frac{8}{p_N}\Big)\sqrt{2d(T-\tau)\log\Big(1 + \frac{T\dot{\mu}_{\min}}{d\lambda}\Big)} + \frac{4\gamma\dot{\mu}_{\max}}{p_N}\sqrt{2(T-\tau)\log\frac{1}{\delta}} + \tau\Delta_{\max},
\end{aligned}$$

Combining the fact that the warm up rounds $\tau = \widetilde{\mathcal{O}}(d^{9/2})$ and set regularization $\lambda = 1 \vee (2dM/S) \log \left( e\sqrt{1 + TL/d} \vee 1/\delta \right)$, the final regret bound is

$$R(T) \leq \widetilde{\mathcal{O}}\big(d^{3/2}\sqrt{T} + d^{9/2}\big),$$

with probability at least $1 - 4\delta$. This completes the proof. $\qquad\square$

## B PROOF OF TECHNICAL LEMMAS IN GLM-ES

### B.1 PROOF OF LEMMA A.3 (CONCENTRATION OF MLE)

In this section, we study the distance between the MLE $\bar{\theta}_t$ and the true parameter $\theta^*$. Our main goal is to obtain a high probability upper bound of $\|\theta^* - \bar{\theta}_t\|_{H_t}$. From the previous sections, the definitions of $H_t$ and $\bar{H}_t$ are as follows:

$$H_t = H(\theta^*; \mathcal{D}_t) = \nabla_\theta^2 L(\theta^*; \mathcal{D}_t) = \sum_{l=1}^t \dot{\mu}(X_l^\top \theta^*) X_l X_l^\top + \lambda I,$$

$$\bar{H}_t = H(\bar{\theta}_t; \mathcal{D}_t) = \nabla_\theta^2 L(\bar{\theta}_t; \mathcal{D}_t) = \sum_{l=1}^t \dot{\mu}(X_l^\top \bar{\theta}_t) X_l X_l^\top + \lambda I.$$

To achieve the upper bound of $\|\theta^* - \bar{\theta}_t\|_{H_t}$, we first focus on upper bounding $\|f_t(\bar{\theta}_t) - f_t(\theta^*)\|_{H_t^{-1}}$. Recall from definition, we have that

$$f_t(\bar{\theta}_t) = \sum_{l=1}^t Y_l X_l, \quad f_t(\theta^*) = \sum_{l=1}^t \mu(X_l^\top \theta^*) X_l + \lambda \theta^*.$$

By defining $\epsilon_l = Y_l - \mu(X_l^\top \theta^*)$, $S_t = \sum_{l=1}^t \epsilon_l X_l$, we have

$$f_t(\bar{\theta}_t) - f_t(\theta^*) = \sum_{l=1}^t (Y_l - \mu(X_l^\top \theta^*)) X_l - \lambda \theta^* = S_t - \lambda \theta^*.$$

Therefore, we have the following decomposition:

$$\|f_t(\bar{\theta}_t) - f_t(\theta^*)\|_{H_t^{-1}} \leq \|S_t\|_{H_t^{-1}} + \lambda \|\theta^*\|_{H_t^{-1}} \leq \|S_t\|_{H_t^{-1}} + \sqrt{\lambda} S,$$

where we used the fact that $\lambda I \preceq H_t$ and $\|\theta^*\| \leq S$.

Next, we need the upper bound for $\|S_t\|_{H_t^{-1}}$. We need the following result from Janz et al. (2024b).

**Lemma B.1** (Theorem 2 in Janz et al. (2024b)). Fix $\lambda, M > 0$. Let $(X_t)_{t \in \mathbb{N}^+}$ be a $B_2^d$-valued random sequence, $(\nu_t)_{t \in \mathbb{N}}$ be a non-negative valued random sequence. Let $\mathbb{F}' = (\mathbb{F}'_t)_{t \in \mathbb{N}}$ be filtration such that (i) $(X_t)_{t \in \mathbb{N}^+}$ is $\mathbb{F}'$-predictable and (ii) $(Y_t)_{t \in \mathbb{N}^+}$ and $(\nu_t)_{t \in \mathbb{N}}$ are $\mathbb{F}'$-adapted. Let $\epsilon_t = Y_t - \mathbb{E}[Y_t \mid \mathbb{F}'_{t-1}]$ and assume that the following condition holds:

$$\mathbb{E}[\exp(s\epsilon_t) \mid \mathbb{F}'_{t-1}] \leq \exp(s^2 \nu_{t-1}), \quad \forall |s| \leq 1/M \text{ and } t \in \mathbb{N}^+. \tag{B.1}$$

Then, for $\widetilde{H}_t = \sum_{l=1}^t \nu_{l-1} X_l X_l^\top + \lambda I$ and $S_t = \sum_{l=1}^t \epsilon_l X_l$ and any $\delta > 0$,

$$\mathbb{P}\Big(\exists t \in \mathbb{N}^+ : \|S_t\|_{\widetilde{H}_t^{-1}} \geq \frac{\sqrt{\lambda}}{2M} + \frac{2M}{\sqrt{\lambda}} \log\Big(\frac{\det(\widetilde{H}_t)^{1/2} \lambda^{-d/2}}{\delta}\Big) + \frac{2M}{\sqrt{\lambda}} d \log(2)\Big) \leq \delta.$$

According to previous discussions, we choose the sequences in the lemma as follows:

$$\nu_{t-1} \to \dot{\mu}(X_t^\top \theta^*), \quad \widetilde{H}_t \to H_t.$$

To apply Lemma B.1, we need to prove that $\epsilon_t$ satisfies the sub-exponential condition Eq. (B.1). Note that now we have $\epsilon_t = Y_t - \mu(X_t^\top \theta^*)$, we need to utilize the assumption that the reward $Y$ given feature vector $X$ has an exponential-family distribution to prove this property.

From definition, we can write (we use notation $u := X^\top \theta^*$ for simplicity)

$$\mathbb{E}[\exp(s\epsilon_t) \,|\, \mathbb{F}'_{t-1}] = \int_y \exp(s(y - \mu(u))\, p(y \,|\, u)\, \mathrm{d}y.$$

From previous sections,

$$p(y \,|\, u) = h(y) \exp[y \cdot u - b(u)] \quad \rightarrow \quad \int_y h(y) \exp(uy)\, \mathrm{d}y = \exp(b(u)).$$

Therefore, we have

$$\mathbb{E}[\exp(s\epsilon_t) \,|\, \mathbb{F}'_{t-1}] = \exp(-s\mu(u) - b(u)) \int_y h(y) \exp((s + u)y)\, \mathrm{d}y$$
$$= \exp(-s\mu(u) - b(u) + b(s + u)).$$

Next, observing the expressions of (B.1), we need to prove that

$$-s\mu(u) - b(u) + b(s + u) \le s^2 \dot{\mu}(u) \quad \rightarrow \quad b(s + u) - b(u) \le s\mu(u) + s^2 \dot{\mu}(u).$$

By definition, we have $\mu(\cdot) = \dot{b}(\cdot)$, the expression above is equivalent to

$$b(s + u) - b(u) \le s\dot{b}(u) + s^2 \ddot{b}(u).$$

Using Taylor expansion, we can write

$$b(s + u) - b(u) = s\dot{b}(u) + \frac{s^2}{2}\ddot{b}(c), \quad c \in [u, s + u].$$

By self-concordance assumption, for $|s| \le \log(2)/M$, we can bound $\ddot{b}(c)$ as

$$\ddot{b}(c) \le \ddot{b}(u) \exp(M|c - u|) \le \ddot{b}(u) \exp(M|s|) \le 2 \cdot \ddot{b}(u).$$

Therefore, we proved that $\ddot{b}(c) \le 2\ddot{b}(u)$ for $|s| \le \log(2)/M$, the sub-exponential condition holds in this setting.

Applying Lemma B.1 with $M \to M/\log(2)$, we have that

$$||S_t||_{H_t^{-1}} \le \frac{\sqrt{\lambda}}{2M} + \frac{4M}{\sqrt{\lambda}} \log\Big( \frac{\det(H_t)^{1/2}\lambda^{-d/2}}{\delta} \Big) + \frac{2M}{\sqrt{\lambda}}d, \quad \forall t \in [T]$$

holds with probability at least $1 - \delta$. From the definition of $H_t$, we have

$$\det(H_t) \le \lambda^d (1 + \frac{t\dot{\mu}_{\max}}{\lambda d})^d.$$

Therefore, we can write the high probability upper bound as

$$||S_t||_{H_t^{-1}} \le \frac{\sqrt{\lambda}}{2M} + \frac{4M}{\sqrt{\lambda}}\Big(d + \frac{d}{2}\log\Big(1 + \frac{t\dot{\mu}_{\max}}{\lambda d}\Big) + \log\frac{1}{\delta}\Big).$$

We can further write the upper bound of $\|f_t(\bar{\theta}_t) - f_t(\theta^*)\|_{H_t^{-1}}$ as follows:

$$\|f_t(\bar{\theta}_t) - f_t(\theta^*)\|_{H_t^{-1}} \le \sqrt{\lambda}\Big(\frac{1}{2M} + S\Big) + \frac{4M}{\sqrt{\lambda}}\Big(d + \frac{d}{2}\log\Big(1 + \frac{t\dot{\mu}_{\max}}{\lambda d}\Big) + \log\frac{1}{\delta}\Big) =: \gamma_t(\delta, \lambda) \tag{B.2}$$

holds for all $t \in [T]$ with probability at least $1 - \delta$.

Now we are ready to bound $\|\theta^* - \bar{\theta}_t\|_{H_t}$. Recall that we have the following relation:

$$f_t(\theta) - f_t(\theta') = Q(\theta', \theta; \mathcal{D}_t)(\theta - \theta'), \quad Q(\theta, \theta'; \mathcal{D}_t) := \sum_{l=1}^{t} \alpha\big(X_l^\top \theta, X_l^\top \theta'\big) X_l X_l^\top + \lambda I.$$

We denote $Q_t = Q(\bar{\theta}_t, \theta^*; \mathcal{D}_t)$, then $\bar{\theta}_t - \theta^* = Q_t^{-1}(f_t(\bar{\theta}_t) - f_t(\theta^*))$. We have the following bound:

$$\left\|\bar{\theta}_t - \theta^*\right\|^2 = \left\|Q_t^{-1}(f_t(\bar{\theta}_t) - f_t(\theta^*))\right\|^2 \leq \frac{1}{\lambda}\|f_t(\bar{\theta}_t) - f_t(\theta^*)\|_{Q_t^{-1}}^2$$

$$\leq \left(1 + MD(\bar{\theta}_t - \theta^*)\right)\frac{1}{\lambda}\|f_t(\bar{\theta}_t) - f_t(\theta^*)\|_{H_t^{-1}}^2$$

$$\leq \left(1 + M\|\bar{\theta}_t - \theta^*\|\right)\frac{1}{\lambda}\gamma_t(\delta, \lambda)^2.$$

The first inequality holds due to the fact $\lambda I \preceq Q_t$. The second inequality holds because $H(\theta^*; \mathcal{D}_t) \preceq (1 + MD(\theta^* - \bar{\theta}_t))Q(\theta^*, \bar{\theta}_t; \mathcal{D}_t)$ (Lemma A.2). The last inequality holds because $D(v) \leq \|v\|$ for $v \in \mathbb{R}^d$ and is under the assumption $\|f_t(\bar{\theta}_t) - f_t(\theta^*)\|_{H_t^{-1}} \leq \gamma_t(\delta, \lambda)$. Note that for any $b, c \geq 0, x \in \mathbb{R}$, if $x^2 \leq bx + c$, we have $x \leq b + \sqrt{c}$. Based on this result, we have

$$\|\bar{\theta}_t - \theta^*\| \leq M\frac{\gamma_t(\delta, \lambda)^2}{\lambda} + \sqrt{\frac{\gamma_t(\delta, \lambda)^2}{\lambda}} =: \beta_t(\delta, \lambda). \tag{B.3}$$

Next, we can write

$$\|\bar{\theta}_t - \theta^*\|_{H_t} = \left\|Q_t^{-1}(f_t(\bar{\theta}_t) - f_t(\theta^*))\right\|_{H_t}$$

$$= \sqrt{(f_t(\bar{\theta}_t) - f_t(\theta^*))^\top Q_t^{-1} H_t Q_t^{-1}(f_t(\bar{\theta}_t) - f_t(\theta^*))}$$

$$\leq \left(1 + MD(\bar{\theta}_t - \theta^*)\right)\|f_t(\bar{\theta}_t) - f_t(\theta^*)\|_{H_t^{-1}}$$

$$\leq \left(1 + M\|\bar{\theta}_t - \theta^*\|\right)\gamma_t(\delta, \lambda)$$

$$\leq (1 + M\beta_t(\delta, \lambda))\gamma_t(\delta, \lambda).$$

The first inequality holds because $H(\theta^*; \mathcal{D}_t) \preceq (1 + MD(\theta^* - \bar{\theta}_t))Q(\theta^*, \bar{\theta}_t; \mathcal{D}_t)$ (Lemma A.2). The second inequality holds because $D(v) \leq \|v\|$ for $v \in \mathbb{R}^d$ and is under the assumption $\|f_t(\bar{\theta}_t) - f_t(\theta^*)\|_{H_t^{-1}} \leq \gamma_t(\delta, \lambda)$. The last inequality holds because of (B.3). Therefore, we have

$$\|\bar{\theta}_t - \theta^*\|_{H_t} \leq (1 + M\beta_t(\delta, \lambda))\gamma_t(\delta, \lambda), \quad \forall t \in [T]$$

holds with probability at least $1 - \delta$.

While this concentration result of MLE is valid and tight, we need $\|\bar{\theta}_t - \theta^*\|_{H_t}$ to be bounded by a constant to prove the regret bound. Therefore, we need to take the upper bound of $\gamma_t(\delta, \lambda)$. From the definitions, both $\gamma_t(\delta, \lambda)$ and $\beta_t(\delta, \lambda)$ increase as $t$ increases, thus $\gamma_t(\delta, \lambda) \leq \gamma_T(\delta, \lambda)$, $\beta_t(\delta, \lambda) \leq \beta_T(\delta, \lambda)$. Therefore, we have the following concentration result:

$$\|\bar{\theta}_t - \theta^*\|_{H_t} \leq (1 + M\beta_T(\delta, \lambda))\gamma_T(\delta, \lambda), \quad \forall t \in [T]$$

holds with probability at least $1 - \delta$. This completes the proof.

## B.2 PROOF OF LEMMA A.4 (CONCENTRATION OF PERTURBATION)

In this section, our main goal is to find a high probability upper bound of $\|\theta_t^j - \bar{\theta}_t\|_{H_t}$, where $\theta_t^j = \operatorname{argmin}_{\theta \in \mathbb{R}^d} L(\theta; \mathcal{D}_t^j)$ is the estimate of $\theta^*$ from ensemble element $j$, $\mathcal{D}_t^j = \{(X_l, Y_l + Z_l^j)\}_{l=1}^t$ is the perturbed dataset used by element $j$, and $Z_l^j \sim \mathcal{N}(0, \sigma_R^2)$ are perturbations of rewards. Recall from the definition, we have

$$f_t(\bar{\theta}_t) = \sum_{l=1}^t \mu(X_l^\top \bar{\theta}_t)X_l + \lambda\bar{\theta}_t = \sum_{l=1}^t X_l Y_l,$$

$$f_t(\theta_t^j) = \sum_{l=1}^t \mu(X_l^\top \theta_t^j)X_l + \lambda\theta_t^j = \sum_{l=1}^t X_l(Y_l + Z_l^j).$$

Thus we have $f_t(\theta_t^j) - f_t(\bar{\theta}_t) = \sum_{l=1}^t Z_l^j X_l$. We use the following lemma to provide a high probability upper bound.

**Lemma B.2** (Lemma 8 of Lee & Oh (2024)). Let $\{\mathcal{F}_t\}$ be a filtration. Let $\{\eta_t\}_{t=1}^{\infty}$ be a sequence of real-valued random variables such that $\eta_t$ is $\mathcal{F}_{t-1}$-conditionally $\sigma$-sub-Gaussian for some $\sigma \geq 0$. Let $\{X_t\}_{t=1}^{\infty}$ be a sequence of $\mathbb{R}^d$-valued random vectors such that $X_t$ is $\mathcal{F}_{t-1}$-measurable and $\|X_t\|_2 \leq 1$ almost surely for all $t \geq 1$. Fix $\lambda \geq 1$. Let $V_t = \lambda I + \sum_{l=1}^{t} X_l X_l^{\top}$. Then, for any $\delta \in (0, 1]$, with probability at least $1 - \delta$, the following inequality holds for all $t \geq 0$:

$$\left\| \sum_{l=1}^{t} \eta_l X_l \right\|_{V_t^{-1}} \leq \sigma \sqrt{d \log\left(1 + \frac{t}{d\lambda}\right) + 2\log\frac{1}{\delta}}.$$

Applying Lemma B.2, we have that

$$\left\| f_t(\theta_t^j) - f_t(\bar{\theta}_t) \right\|_{G_t^{-1}} = \left\| \sum_{l=1}^{t} Z_l^j X_l \right\|_{G_t^{-1}} \leq \sigma_R \sqrt{d \log\left(1 + \frac{t\dot{\mu}_{\min}}{d\lambda}\right) + 2\log\frac{T}{\delta}} =: \sigma_R \widetilde{\gamma}_t(\delta, \lambda)$$

(B.4)

holds for all $t \geq 0$ with probability at least $1 - \delta/T$.

Next, we use this result to prove a high probability upper bound of $\|\theta_t^j - \bar{\theta}_t\|_{H_t}$. Recall from (A.7), we have

$$f_t(\theta_t^j) - f_t(\bar{\theta}_t) = Q(\theta_t^j, \bar{\theta}_t; \mathcal{D}_t)(\theta_t^j - \bar{\theta}_t) =: \widetilde{Q}_t^j(\theta_t^j - \bar{\theta}_t).$$

From the definition,

$$\widetilde{Q}_t^j = Q(\theta_t^j, \bar{\theta}_t; \mathcal{D}_t) = \sum_{l=1}^{t} \alpha\big(X_l^{\top}\theta_t^j, X_l^{\top}\bar{\theta}_t\big) X_l X_l^{\top} + \lambda I.$$

Based on the definition of $\alpha(\cdot, \cdot)$ and mean value theorem, there exists $\zeta(l) = aX_l^{\top}\theta_t^j + (1-a)X_l^{\top}\bar{\theta}_t \in \mathbb{R}$ with $0 < a < 1$ such that

$$\alpha\big(X_l^{\top}\theta_t^j, X_l^{\top}\bar{\theta}_t\big) = \frac{\mu(X_l^{\top}\theta_t^j) - \mu(X_l^{\top}\bar{\theta}_t)}{X_l^{\top}\theta_t^j - X_l^{\top}\bar{\theta}_t} = \dot{\mu}(\zeta(l)).$$

Note that we have assumed $0 < \dot{\mu}_{\min} \leq \dot{\mu}(\cdot) \leq \dot{\mu}_{\max}$, thus we have

$$\dot{\mu}_{\min} G_t \preceq \widetilde{Q}_t^j \preceq \dot{\mu}_{\max} G_t.$$

Combining the discussions above and the fact $\dot{\mu}_{\min} G_t \preceq H(\theta; \mathcal{D}_t) \preceq \dot{\mu}_{\max} G_t$, when (B.4) holds, we have the following upper bound:

$$\begin{aligned}
\left\| \theta_t^j - \bar{\theta}_t \right\|_{H_t} &= \left\| \big(\widetilde{Q}_t^j\big)^{-1} (f_t(\theta_t^j) - f_t(\bar{\theta}_t)) \right\|_{H_t} \\
&= \sqrt{\big(f_t(\theta_t^j) - f_t(\bar{\theta}_t)\big)^{\top} \big(\widetilde{Q}_t^j\big)^{-1} H_t \big(\widetilde{Q}_t^j\big)^{-1} \big(f_t(\theta_t^j) - f_t(\bar{\theta}_t)\big)} \\
&\leq \dot{\mu}_{\max}^{1/2} \dot{\mu}_{\min}^{-1} \left\| f_t(\theta_t^j) - f_t(\bar{\theta}_t) \right\|_{G_t^{-1}} \\
&\leq \dot{\mu}_{\max}^{1/2} \dot{\mu}_{\min}^{-1} \sigma_R \widetilde{\gamma}_t(\delta, \lambda).
\end{aligned}$$

The result above is for fixed ensemble element $j \in [m]$. We further define the following event for each round $t$:

$$\widetilde{\mathcal{E}}_{1,t} := \{\left\| \theta_t^{j_t} - \bar{\theta}_t \right\|_{H_t} \leq \dot{\mu}_{\max}^{1/2} \dot{\mu}_{\min}^{-1} \sigma_R \widetilde{\gamma}_T(\delta, \lambda)\}.$$

Since $j_t$ is sampled independently, we have

$$\mathbb{P}(\widetilde{\mathcal{E}}_{1,t}) = \sum_{j=1}^{m} \mathbb{P}(j_t = j, \left\| \theta_t^j - \bar{\theta}_t \right\|_{H_t} \leq \dot{\mu}_{\max}^{1/2} \dot{\mu}_{\min}^{-1} \sigma_R \widetilde{\gamma}_T(\delta, \lambda)) \geq 1 - \frac{\delta}{T}.$$

Therefore, for each round $t \in [T]$, the concentration of perturbation holds with high probability. This completes the proof.

## B.3 Sufficient Condition of Optimism

In this section, our main goal is to obtain a sufficient condition of optimism $X_t^\top \theta_t \geq X^{*\top} \theta^*$. The motivation is that we need to find a lower bound of probability of optimism for each round, but analyzing this probability directly is very challenging. To overcome these difficulties, we need to find a sufficient condition for optimism using variables $\{X_l\}_{l=1}^t$ and $\{Z_l^j\}_{l=1}^t$ explicitly. By analyzing the probability of this sufficient condition, we can obtain a lower bound of probability of optimism. To achieve this, for each member $j \in [m]$ in the ensemble, we consider the following inequality:

$$
\begin{aligned}
X_t^\top \theta_t^j - X^{*\top} \theta^* &\geq X^{*\top} \theta_t^j - X^{*\top} \theta^* \\
&= X^{*\top} (\widetilde{\theta}_t^j + \bar{\theta}_t - \theta^*) \\
&= X^{*\top} \widetilde{\theta}_t^j + X^{*\top} (\bar{\theta}_t - \theta^*),
\end{aligned}
\tag{B.5}
$$

where $\widetilde{\theta}_t^j = \theta_t^j - \bar{\theta}_t$. Next, we introduce the following notations:

$$
\mathbf{Z}_t^j = \big(\underbrace{0, \cdots, 0}_{d}, Z_1^j, \cdots, Z_t^j\big)^\top \in \mathbb{R}^{d+t}, \quad \Phi_t = \big(\sqrt{\lambda/\dot{\mu}_{\min}} I_d, X_1, \cdots, X_t\big) \in \mathbb{R}^{d \times (d+t)},
$$

then we have $\Phi_t \mathbf{Z}_t^j = \sum_{l=1}^t X_l Z_l^j$. Recall that we have

$$
\widetilde{\theta}_t^j = \theta_t^j - \bar{\theta}_t = \big(\widetilde{Q}_t^j\big)^{-1} \big(f_t(\theta_t^j) - f_t(\bar{\theta}_t)\big) = \big(\widetilde{Q}_t^j\big)^{-1} \sum_{l=1}^t X_l Z_l^j,
$$

we obtain the relation $\widetilde{\theta}_t^j = \big(\widetilde{Q}_t^j\big)^{-1} \Phi_t \mathbf{Z}_t^j$. We further define

$$
U_t^\top = X^{*\top} \bar{H}_t^{-1} \Phi_t \in \mathbb{R}^t,
$$

then we have

$$
\|X^*\|_{\bar{H}_t^{-1}}^2 = X^{*\top} \bar{H}_t^{-1} X^* \leq \dot{\mu}_{\max}^2 X^{*\top} \bar{H}_t^{-1} \Phi_t \Phi_t^\top \bar{H}_t^{-1} X^* = \dot{\mu}_{\max}^2 \|U_t\|_2^2.
$$

This relates the norm of $X^*$ to $U_t$.

With these notations and results, we can continue analyzing B.5. First, we can use concentration to directly bound $X^{*\top} (\bar{\theta}_t - \theta^*)$ as follows:

$$
X^{*\top} (\bar{\theta}_t - \theta^*) \geq -\|X^*\|_{\bar{H}_t^{-1}} \|\bar{\theta}_t - \theta^*\|_{\bar{H}_t} \geq -\dot{\mu}_{\max}^2 \|U_t\|_2^2 \|\bar{\theta}_t - \theta^*\|_{\bar{H}_t}.
$$

Assume that $\|\bar{\theta}_t - \theta^*\|_{\bar{H}_t} \leq c$, where $c > 0$ is a constant, then

$$
X^{*\top} (\bar{\theta}_t - \theta^*) \geq -c \dot{\mu}_{\max}^2 \|U_t\|_2^2.
$$

Next, we analyze $X^{*\top} \widetilde{\theta}_t^j$. From previous discussions, we have the following expression:

$$
X^{*\top} \widetilde{\theta}_t^j = X^{*\top} \big(\widetilde{Q}_t^j\big)^{-1} \Phi_t \mathbf{Z}_t^j.
$$

Recall that from the definition,

$$
\widetilde{Q}_t^j = Q(\theta_t^j, \bar{\theta}_t; \mathcal{D}_t) = \sum_{l=1}^t \alpha\big(X_l^\top \theta_t^j, X_l^\top \bar{\theta}_t\big) X_l X_l^\top + \lambda I.
$$

Since $\widetilde{Q}_t^j$ includes the information of $\{Z_l^j\}_{l=1}^t$, the randomness from perturbation is not fully decoupled and we cannot directly use tail bound of the distribution of $Z_t^j$. To overcome this difficulty, we define $W_t^j = \widetilde{Q}_t^j - \bar{H}_t$, and we have the following relation:

$$
\big(\widetilde{Q}_t^j\big)^{-1} = \big(\bar{H}_t + W_t^j\big)^{-1} = \bar{H}_t^{-1} - \bar{H}_t^{-1} W_t^j \big(\bar{H}_t + W_t^j\big)^{-1}.
$$

Note that $\bar{H}_t$ does not include information of perturbations. Based on (B.5), to guarantee that optimism holds, it suffices to satisfy that

$$X^{*\top}\widetilde{\theta}_t^j + X^{*\top}(\bar{\theta}_t - \theta^*)$$

$$= X^{*\top}\bar{H}_t^{-1}\Phi_t\mathbf{Z}_t^j - X^{*\top}\big(\bar{H}_t^{-1}W_t^j(\bar{H}_t + W_t^j)^{-1}\big)\Phi_t\mathbf{Z}_t^j + X^{*\top}(\bar{\theta}_t - \theta^*) \geq 0.$$

Then it suffices to satisfy that

$$X^{*\top}\bar{H}_t^{-1}\Phi_t\mathbf{Z}_t^j \geq \left|X^{*\top}\left((\bar{H}_t^{-1}W_t^j(\bar{H}_t + W_t^j)^{-1}\right)\Phi_t\mathbf{Z}_t^j\right| + \left|X^{*\top}(\bar{\theta}_t - \theta^*)\right|.$$

Note that in $X^{*\top}\bar{H}_t^{-1}\Phi_t\mathbf{Z}_t^j$, the randomness of $\{Z_l^j\}_{l=1}^t$ only appears in $\mathbf{Z}_t^j$, thus we decoupled the randomness of perturbation and we can utilize the tail bound of $Z_l^j$.

Next, we need to upper bound the right-hand-side of the inequality to obtain a sufficient condition of optimism. We already have the result

$$\left|X^{*\top}(\bar{\theta}_t - \theta^*)\right| \leq c\,\dot{\mu}_{\max}^2\|U_t\|_2^2.$$

We upper bound $\left|X^{*\top}(\bar{H}_t^{-1}W_t^j(\bar{H}_t + W_t^j)^{-1})\Phi_t\mathbf{Z}_t^j\right|$ as follows.

$$\left|X^{*\top}\big(\bar{H}_t^{-1}W_t^j(\widetilde{Q}_t^j)^{-1}\big)\Phi_t\mathbf{Z}_t^j\right| = \left|\big(\bar{H}_t^{-1/2}X^*\big)^\top\big(\bar{H}_t^{-1/2}W_t^j\bar{H}_t^{-1/2}\big)\big(\bar{H}_t^{1/2}(\widetilde{Q}_t^j)^{-1}\Phi_t\mathbf{Z}_t^j\big)\right|$$

$$\leq \big\|\bar{H}_t^{-1/2}X^*\big\|_2\big\|\bar{H}_t^{-1/2}W_t^j\bar{H}_t^{-1/2}\big\|_2\big\|\bar{H}_t^{1/2}(\widetilde{Q}_t^j)^{-1}\Phi_t\mathbf{Z}_t^j\big\|_2$$

$$= \|X^*\|_{\bar{H}_t^{-1}}\big\|\bar{H}_t^{-1/2}W_t^j\bar{H}_t^{-1/2}\big\|_2\big\|(\widetilde{Q}_t^j)^{-1}\Phi_t\mathbf{Z}_t^j\big\|_{\bar{H}_t}$$

$$= \|X^*\|_{\bar{H}_t^{-1}}\big\|\bar{H}_t^{-1/2}(\widetilde{Q}_t^j - \bar{H}_t)\bar{H}_t^{-1/2}\big\|_2\big\|\widetilde{\theta}_t^j\big\|_{\bar{H}_t}.$$

According to the definition, for symmetric matrix $A$, we have

$$\|A\|_2 = \max_{\|v\|=1}\|Av\|_2 = |\lambda_{\max}(A)|.$$

From definition,

$$\bar{H}_t^{-1/2}\big(\widetilde{Q}_t^j - \bar{H}_t\big)\bar{H}_t^{-1/2} \preceq \bar{H}_t^{-1/2}\bigg(\sum_{l=1}^t\Big|\alpha(X_l^\top\bar{\theta}_t, X_l^\top\theta_t^j) - \dot{\mu}(X_l^\top\bar{\theta}_t)\Big|X_lX_l^\top\bigg)\bar{H}_t^{-1/2}.$$

From Lemma A.1, we have

$$\alpha\big(X_l^\top\bar{\theta}_t, X_l^\top\theta_t^j\big) \leq \dot{\mu}\big(X_l^\top\bar{\theta}_t\big)\frac{\exp(MD_t) - 1}{MD_t},$$

where $D_t = \max_{l<t}|X_l^\top(\bar{\theta}_t - \theta_t^j)|$. Then we have

$$\bar{H}_t^{-1/2}\big(\widetilde{Q}_t^j - \bar{H}_t\big)\bar{H}_t^{-1/2} \preceq \bar{H}_t^{-1/2}\bigg(\sum_{l=1}^t\Big|\frac{\exp(MD_t) - 1}{MD_t} - 1\Big|\dot{\mu}(X_l^\top\bar{\theta}_t)X_lX_l^\top\bigg)\bar{H}_t^{-1/2}.$$

Let $R_t = \frac{\exp(MD_t) - 1}{MD_t} - 1$, then for any $v \in \mathbb{R}^d$ with $\|v\| = 1$, we have

$$v^\top\Big(\bar{H}_t^{-1/2}\big(\widetilde{Q}_t^j - \bar{H}_t\big)\bar{H}_t^{-1/2}\Big)v \leq R_t v^\top\bar{H}_t^{-1/2}(\bar{H}_t - \lambda I)\bar{H}_t^{-1/2}v \leq R_t.$$

We then obtain that

$$\big\|\bar{H}_t^{-1/2}\big(\widetilde{Q}_t^j - \bar{H}_t\big)\bar{H}_t^{-1/2}\big\|_2 = \big|\lambda_{\max}\big(\bar{H}_t^{-1/2}\big(\widetilde{Q}_t^j - \bar{H}_t\big)\bar{H}_t^{-1/2}\big)\big| \leq R_t.$$

From concentration results, under event $\mathcal{E}_1$, we have

$$\|\bar{\theta}_t - \theta^*\|_{H_t} \leq (1 + M\beta_T)\gamma_T, \tag{B.6}$$

$$\|\widetilde{\theta}_t^j\|_{H_t} \leq \dot{\mu}_{\max}^{1/2}\dot{\mu}_{\min}^{-1}\widetilde{\gamma}_T\sigma_R, \tag{B.7}$$

for all $t \in [T]$, and $\mathcal{E}_1$ holds with probability at least $1 - 2\delta$. Here we should note that we need the concentration bound with $\bar{H}_t$ instead of $H_t$. From the $M$-self-concordant assumption, we have

$$e^{-MD(\bar{\theta}_t - \theta^*)}\bar{H}_t \preceq H_t \preceq e^{MD(\bar{\theta}_t - \theta^*)}\bar{H}_t.$$

Therefore, we have the following concentration bounds.

$$\|\bar{\theta}_t - \theta^*\|_{\bar{H}_t} \leq e^{MD(\bar{\theta}_t - \theta^*)/2}(1 + M\beta_T)\gamma_T, \tag{B.8}$$

$$\|\widetilde{\theta}_t^j\|_{\bar{H}_t} \leq e^{MD(\bar{\theta}_t - \theta^*)/2}\dot{\mu}_{\max}^{1/2}\dot{\mu}_{\min}^{-1}\widetilde{\gamma}_T\sigma_R. \tag{B.9}$$

We then have

$$\left|X^{*\top}\left(\bar{H}_t^{-1}W_t^j(\bar{H}_t + W_t^j)^{-1}\right)\Phi_t\mathbf{Z}_t^j\right| + \left|X^{*\top}(\bar{\theta}_t - \theta^*)\right|$$

$$\leq \left((1 + M\beta_T)\gamma_T + R_t\dot{\mu}_{\max}^{1/2}\dot{\mu}_{\min}^{-1}\widetilde{\gamma}_T\sigma_R\right)e^{MD(\bar{\theta}_t - \theta^*)/2}\|X^*\|_{\bar{H}_t^{-1}}.$$

Finally, we obtain the sufficient condition for optimism as follows:

$$X^{*\top}\bar{H}_t^{-1}\Phi_t\mathbf{Z}_t^j \geq \left((1 + M\beta_T)\gamma_T + R_t\dot{\mu}_{\max}^{1/2}\dot{\mu}_{\min}^{-1}\widetilde{\gamma}_T\sigma_R\right)e^{MD(\bar{\theta}_t - \theta^*)/2}\|X^*\|_{\bar{H}_t^{-1}}. \tag{B.10}$$

Recall the definition $U_t^\top = X^{*\top}\bar{H}_t^{-1}\Phi_t \in \mathbb{R}^t$ and the result

$$\|X^*\|_{\bar{H}_t^{-1}}^2 = X^{*\top}\bar{H}_t X^* \leq \dot{\mu}_{\max}^2\|U_t\|_2^2,$$

we obtain the final expression of the sufficient condition for optimism:

$$U_t^\top\mathbf{Z}_t^j \geq \dot{\mu}_{\max}\left((1 + M\beta_T)\gamma_T + R_t\dot{\mu}_{\max}^{1/2}\dot{\mu}_{\min}^{-1}\widetilde{\gamma}_T\sigma_R\right)e^{MD(\bar{\theta}_t - \theta^*)/2}\|U_t\|_2.$$

To simplify the notations, define $b_1' = \dot{\mu}_{\max}(1 + M\beta_T)\gamma_T$ and $b_2' = \dot{\mu}_{\max}^{3/2}\dot{\mu}_{\min}^{-1}\widetilde{\gamma}_T$, we have the following expression:

$$U_t^\top\mathbf{Z}_t^j \geq (b_1' + b_2'R_t\sigma_R)e^{MD(\bar{\theta}_t - \theta^*)/2}\|U_t\|_2.$$

Next, we discuss the order of the term $\exp(MD(\bar{\theta}_t - \theta^*)/2)$. From the warm-up procedure and Lemma B.5, $\max_{X \in \mathcal{X}}\|X\|_{H_t^{-1}} = \widetilde{\mathcal{O}}(M^{-1}d^{-2})$. Then we have

$$MD(\bar{\theta}_t - \theta^*) = M\max_{s<t}\left|X_s^\top(\bar{\theta}_t - \theta^*)\right| \leq M\max_{s<t}\|X_s\|_{H_t^{-1}}\|\bar{\theta}_t - \theta^*\|_{H_t} = \widetilde{\mathcal{O}}(d^{-3/2}).$$

Therefore, $\exp(MD(\bar{\theta}_t - \theta^*)/2) \sim \widetilde{\mathcal{O}}(1)$ and does not affect the regret bound analysis. In the following analysis, we assume that $\exp(MD(\bar{\theta}_t - \theta^*)/2) \leq C^*$, where $C^* > 1$ is a constant. Then we have the following sufficient condition for optimism:

$$U_t^\top\mathbf{Z}_t^j \geq C^*(b_1' + b_2'R_t\sigma_R)\|U_t\|_2.$$

We summarize these discussions to the following lemma.

**Lemma B.3** (Sufficient Condition for Optimism). For $t \in [T]$ and $j \in [m]$, define vector $U_t^\top = X^{*\top}\bar{H}_t^{-1}\Phi_t \in \mathbb{R}^t$, $b_1 = C^*\dot{\mu}_{\max}(1 + M\beta_T)\gamma_T$ and $b_2 = C^*\dot{\mu}_{\max}^{3/2}\dot{\mu}_{\min}^{-1}\widetilde{\gamma}_T$, where $C^* > 1$ is a constant. Then, under the concentration event $\mathcal{E}_1$, optimism $X^{*\top}\theta^* \leq X_t^\top\theta_t$ holds if the following inequality is satisfied:

$$U_t^\top\mathbf{Z}_t^j \geq (b_1 + b_2R_t\sigma_R)\|U_t\|_2. \tag{B.11}$$

### B.4 Proof of Lemma A.6 (Probability of Optimism)

With the sufficient condition of optimism Lemma B.3 and the fact that perturbations only appear in $\mathbf{Z}_t^j$, we can apply the tail bound of perturbations to obtain the probability of optimism. We use Gaussian noise $Z_l^j \sim \mathcal{N}(0, \sigma_R^2)$ for reward perturbation, thus we can apply the following tail bound: for any $u \in \mathbb{R}^n$ and $\mathbf{Z} \sim \mathcal{N}(0, \sigma_R^2 I_n)$, we have the following inequality

$$\mathbb{P}\left(u^\top\mathbf{Z} \geq \sigma_R\|u\|_2\right) \geq p_N,$$

where $p_N$ is a constant and $p_N \geq 0.15$. Based on this result, if we fix the sequence of pulled arms $\{X_l\}_{l=1}^t$ and element $j \in [m]$, (B.11) in Lemma B.3 holds with probability at least $p_N$ if we choose $\sigma_R$ that satisfies

$$\sigma_R \geq b_1 + b_2R_t\sigma_R. \tag{B.12}$$

Next, we define the indicator function that optimism and concentration holds at the beginning of round $t$: $I_t^j := \mathbb{1}\{(U_{t-1}^\top \mathbf{Z}_{t-1}^j \geq (b_1 + b_2 R_{t-1}\sigma_R)\|U_{t-1}\|_2) \cap \widetilde{\mathcal{E}}_{1,t-1}\}$. We proved that $\widetilde{\mathcal{E}}_{1,t-1}$ holds with probability at least $1 - \delta/T$, thus we have

$$\mathbb{P}(I_t^j = 1) \geq \mathbb{P}(U_{t-1}^\top \mathbf{Z}_{t-1}^j \geq (b_1 + b_2 R_{t-1}\sigma_R)\|U_{t-1}\|_2) - \frac{\delta}{T} \geq p_N - \frac{\delta}{T} \geq \frac{p_N}{2},$$

where we assumed that $\delta/T \leq p_N/2$. Note that $I_t^j$ is $\mathcal{F}_{t-1}$-measurable: given the history up to round $(t-1)$, the value of $I_t^j$ is determined for each model $j \in [m]$. Now we add the randomness of choosing arm $j_t$ at round $t$. Conditioning on $\mathcal{F}_{t-1}$, the probability of optimism after uniformly randomly choosing model $j_t$ is given by

$$\mathbb{P}\Big((U_{t-1}^\top \mathbf{Z}_{t-1}^{j_t} \geq (b_1 + b_2 R_{t-1}\sigma_R)\|U_{t-1}\|_2) \cap \widetilde{\mathcal{E}}_{1,t-1} \mid \mathcal{F}_{t-1}\Big) = \frac{1}{m}\sum_{j=1}^m I_t^j.$$

Using Azuma-Hoeffding inequality, we have the following result:

$$\mathbb{P}\Big(\frac{1}{m}\sum_{j=1}^m I_t^j < \frac{p_N}{4}\Big) \leq \exp\Big(-\frac{p_N^2 m}{8}\Big).$$

The results above are obtained under fixed sequence of pulled arms $\{X_l\}_{l=1}^t$, we use the following result to count equivalent permutations of $\{X_l\}_{l=1}^t$.

**Lemma B.4** (Claim 1 in Lee & Oh (2024)). There exists an event $\mathcal{E}^*$ such that under $\mathcal{E}^*$, $\frac{1}{m}\sum_{j=1}^m I_t^j \geq p_N/4$ holds for all $t \in [T]$, and $\mathbb{P}(\bar{\mathcal{E}}^*) \leq T^K \exp(-p_N^2 m/8)$.

Therefore, by choosing ensemble size

$$m \geq \frac{8}{p_N^2}\Big(K\log T + \log\frac{1}{\delta}\Big),$$

we have $\mathbb{P}(\bar{\mathcal{E}}^*) \leq 1 - \delta$, with high probability, optimism holds with constant probability. This completes the proof.

## B.5 PROOF OF LEMMA A.7

The proof of this technical lemma mostly follows the proof of Lemma 6 in (Lee & Oh, 2025). From the definitions of $g_t(\theta^*, \mathcal{E}')$:

$$g_t(\theta^*, \mathcal{E}') = (J(\theta^*) - J(\theta_t^-))\mathbb{1}\{\mathcal{E}'\}, \quad J(\theta_t^-) = \inf_{\theta \in \Theta_t} J(\theta).$$

Therefore, we have

$$J(\theta^*) \geq J(\theta_t^-) \quad \rightarrow \quad g_t(\theta^*, \mathcal{E}') \geq 0,$$

holds for any event $\mathcal{E}' \in \mathcal{F}$. Now we consider event $\mathcal{E}'' \subset \mathcal{E}_{1,t}$. If $\mathcal{E}''$ does not hold, $g_t(\theta_t, \mathcal{E}'') = 0 \geq 0$. If $\mathcal{E}''$ holds, from the concentration result Lemma A.5, we have $\theta_t \in \Theta_t$. Then we have

$$J(\theta_t) \geq J(\theta_t^-) \quad \rightarrow \quad g_t(\theta_t, \mathcal{E}'') \geq 0.$$

Therefore, $g_t(\theta_t, \mathcal{E}'') \geq 0$ holds almost surely for any event such that $\mathcal{E}'' \subset \mathcal{E}_{1,t}$. This completes the proof.

## B.6 WARM-UP PROCEDURE

In the analysis of optimism, we assumed that (B.12) has a solution. In this section, we analyze how to guarantee that solution of (B.12) exists. From the expression, we need to set warm-up parameter $\tau$ to make sure that $R_t$ is sufficiently small, so that the solution is approximately $\sigma_R = b_1$. Recall from the definition of $R_t$, we have

$$R_t = \frac{\exp(MD_t) - 1}{MD_t} - 1 \leq \frac{1 + MD_t + (MD_t)^2 - 1}{MD_t} - 1 = MD_t,$$

where the inequality holds when $MD_t \leq 1$. Therefore, we should use warm-up procedure to reduce $D_t$ until sufficiently small. Note that

$$D_t = \max_{s<t} \left| X_s^\top (\bar{\theta}_t - \theta_t^j) \right| \leq \max_{s<t} \|X_s\|_{H_t^{-1}} \|\bar{\theta}_t - \theta_t^j\|_{H_t},$$

where the inequality holds due to Cauchy-Schwarz inequality. Therefore, we should focus on upper bounding $\max_{X \in \mathcal{X}} \|X\|_{H_t}$. This is achieved by the following warm-up procedures.

---

**Algorithm 2** Warm-up of GLM-ES

---

**Input:** arm set $\mathcal{X} \subset \mathbb{R}^d, \tau, \epsilon \in (0,1)$
1: Set $\zeta = \operatorname{argmin}_{\zeta \in \Delta_\mathcal{X}} \max_{X \in \mathcal{X}} \|X\|_{V(\zeta)^{-1}}^2$
2: Pull $x_1, \ldots, x_\tau = \operatorname{round}(\tau, \zeta, \epsilon)$
3: Observe rewards $y_1, \ldots, y_\tau$
**Return:** $\{(x_s, y_s)\}_{s=1}^\tau$

---

**Algorithm 3** Rounding procedure of Algorithm 2

---

**Input:** $\tau, \zeta$ and $\epsilon$
1: $r(\epsilon) \leftarrow (d(d+1)/2 + 1)/\epsilon$
2: $p \leftarrow |\operatorname{supp}(\zeta)|$
3: $\{x_i\}_{i=1}^p \leftarrow \operatorname{supp}(\zeta)$
4: $N_i \leftarrow \lceil (\tau - p/2)\zeta_i \rceil$, for all $i \leq p$
5: **while** $\sum_{i=1}^p N_i \neq \tau$ **do**
6:    **if** $\sum_{i=1}^p N_i < \tau$ **then**
7:       $j \leftarrow \operatorname{argmin}_{i \leq p}(N_i - 1)/\zeta_i$
8:       $N_j \leftarrow N_j + 1$
9:    **end if**
10:   **if** $\sum_{i=1}^p N_i > \tau$ **then**
11:      $j \leftarrow \operatorname{argmax}_{i \leq p}(N_i - 1)/\zeta_i$
12:      $N_j \leftarrow N_j - 1$
13:   **end if**
14: **end while**
15: $N_i \leftarrow \max\left(N_i, r(\epsilon)/p\right)$
**Return:** $N_1, ..., N_p$ for all $i \leq p$

---

In GLM-ES, Algorithm 1 is initialized by calling a warm-up procedure Algorithm 2 that approximates a G-optimal design. This is necessary to guarantee the sufficient condition for optimism (B.11) happens with constant probability. Here, $\zeta \in \Delta_\mathcal{X}$ is a probability measure over arm set $\mathcal{X}$, which is set to minimize

$$\max_{X \in \mathcal{X}} \|X\|_{V(\zeta)^{-1}}^2, \quad \text{where } V(\zeta) = \sum_{X \in \mathcal{X}} \zeta(X) X X^\top.$$

We then sample $X_1, \ldots, X_\tau$ based on G-optimal design $\zeta$ by following the rounding procedure (Line 2 in Algorithm 2). The complete rounding procedure is given in Algorithm 3, which is proposed in Chapter 12 of Pukelsheim (2006) and is detailed in Fiez et al. (2019).

We provide a theoretical guarantee for the warm-up procedure as follows.

**Lemma B.5** (Warm-up). Let $\zeta$ be the $G$-optimal design solution over a compact feature set $\mathcal{X}$ whose span is $\mathbb{R}^d$. Let $\iota > 0$ and $\epsilon = 0.5$. By setting the number of warm-up rounds as $\tau = \lceil 1.5\iota^2 d/\dot{\mu}_{\min} \rceil$, Algorithm 2 returns a dataset $\{(X_s, Y_s)\}_{s=1}^\tau$ such that for all $t > \tau$, we have

$$\max_{X \in \mathcal{X}} \|X\|_{H_t^{-1}} \leq 1/\iota. \tag{B.13}$$

*Proof of Lemma B.5.* The proof mainly follows the proof of (Liu, 2023, Lemma 4.10), we write it here for completeness. Note that for all $t > \tau$, we have $H_t^{-1} \preceq H_\tau^{-1}$. Then we have

$$\max_{X \in \mathcal{X}} \|X\|_{H_t^{-1}}^2 \leq \max_{X \in \mathcal{X}} \|X\|_{H_\tau^{-1}}^2.$$

We further define $H(\zeta, \theta) = \sum_{X \in \mathcal{X}} \zeta(X) \dot{\mu}(X^\top \theta) X X^\top$ where $\zeta$ is a probability measure over $\mathcal{X}$. Then it is sufficient to prove that for all $t > \tau$, $\max_{X \in \mathcal{X}} \|X\|_{H_\tau^{-1}} \leq 1/\iota$ holds.

$$\max_{X \in \mathcal{X}} \|X\|_{H_\tau^{-1}}^2 \leq \frac{1+\epsilon}{\tau} \max_{X \in \mathcal{X}} \|X\|_{H^{-1}(\zeta, \theta^*)}^2$$

$$\leq \frac{1+\epsilon}{\tau \dot{\mu}_{\min}} \max_{X \in \mathcal{X}} \|X\|_{V^{-1}(\zeta)}^2$$

$$\leq \frac{d(1+\epsilon)}{\tau \dot{\mu}_{\min}},$$

where the first inequality holds because of (Jun et al., 2021, Lemma 13), the second inequality holds because $H(\zeta, \theta^*) \succeq \dot{\mu}_{\min} V(\zeta)$ and the last inequality holds due to (Lattimore & Szepesvári, 2020, Theorem 21.1 (Kiefer–Wolfowitz)). The proof is completed by setting $\tau = \iota^2 d(1+\epsilon)/\dot{\mu}_{\min}$. □

To guarantee that (B.12) holds, we desire to have $R_t \leq MD_t \leq \sigma_R^{-\varepsilon} < 1$, where $\varepsilon \in (0,1)$. Therefore, it suffices to have

$$\iota \geq M b_2 \dot{\mu}_{\max}^{-1} \sigma_R^{1+\varepsilon}.$$

Also note that the solution for $\sigma_R \geq b_1 + b_2 \sigma_R^{1-\varepsilon}$ satisfies (B.12), and we have $\sigma_R \gtrsim (d \log T)^{\frac{1}{2\varepsilon}}$.

Recall that $\tau = \lceil 1.5 \iota^2 d / \dot{\mu}_{\min} \rceil$, then we can obtain the order of warm-up parameter $\tau$ as follows

$$\tau \gtrsim C d^2 \log T (d \log T)^{\frac{1+\varepsilon}{\varepsilon}},$$

where $C$ is a constant with respect to $(M, \dot{\mu}_{\max}, \dot{\mu}_{\min})$. By setting $\varepsilon = \frac{2}{3}$, we can obtain

$$\tau = \widetilde{O}\big(d^{\frac{9}{2}}\big). \tag{B.14}$$

Note that in Liu (2023), the warm-up parameter $\tau = \widetilde{O}(d^9)$, which is much larger than (B.14) that we derive.

In summary, to guarantee that (B.12) has a solution, we should set warm-up rounds $\tau = \widetilde{O}\big(d^{9/2}\big)$ with Algorithm 2. Accordingly, the variance of perturbation $\sigma_R$ should satisfy $\sigma_R \geq b_1 + b_2 \sigma_R^{1/3}$, we should set $\sigma_R = \Theta\big(b_1 + b_2^{3/2}\big)$.

## C  PROOF OF REGRET BOUND OF NEURAL-ES

### C.1  PRELIMINARY ANALYSIS

Recall that we adopt a fully connected neural network $f(X; \theta) : \mathbb{R}^d \to \mathbb{R}$ to approximate the true reward model $h(X)$:

$$f(X; \theta) = \sqrt{N} W_L \phi\Big(W_{L-1} \phi\big(\cdots \phi(W_1 X)\big)\Big),$$

where $N$ is the width of each layer (we assume that the widths of each layer are the same), $\theta = [\text{vec}(W_1), \cdots, \text{vec}(W_L)] \in \mathbb{R}^{d'}$ is the concatenation of all learnable parameters. We use $g(X; \theta_0) = \nabla_\theta f(X; \theta_0) \in \mathbb{R}^{d'}$ to denote the gradient of the initial neural network. For convenience and comparison with linear contextual bandit, we define the following mapping:

$$\psi(X) = \frac{g(X; \theta_0)}{\sqrt{N}} : \quad \mathbb{R}^d \to \mathbb{R}^{d'}. \tag{C.1}$$

As we will see in the following analysis, after we map the $d$-dimensional feature vector $X$ into the $d'$-dimensional $\psi(X)$, the analysis of `Neural-ES` is similar to that of `Lin-ES`.

We define the covariance matrix $A_t$ at round $t$ as

$$A_t = \frac{1}{N} \sum_{l=1}^{t-1} g(X_l; \theta_0) g(X_l; \theta_0)^\top + \lambda I_{d'} = \sum_{l=1}^{t-1} \psi(X_l) \psi(X_l)^\top + \lambda I_{d'}, \tag{C.2}$$

and the following auxiliary vectors

$$\bar{\mathbf{b}}_t = \frac{1}{\sqrt{N}} \sum_{l=1}^{t-1} Y_l \, g(X_l; \theta_0) = \sum_{l=1}^{t-1} Y_l \, \psi(X_l),$$

$$\mathbf{b}_t^j = \frac{1}{\sqrt{N}} \sum_{l=1}^{t-1} (Y_l + Z_l^j) \, g(X_l; \theta_0) = \sum_{l=1}^{t-1} (Y_l + Z_l^j) \, \psi(X_l).$$

Then, similar to linear contextual bandit, the MLE without perturbation is $\bar{\theta}_t = A_t^{-1} \bar{\mathbf{b}}_t$, the least square solution for parameter $\theta_t^j$ (of model $j \in [m]$) is $A_t^{-1} \mathbf{b}_t^j$. These notations and results will be utilized extensively in the following analysis.

## C.2 TECHNICAL LEMMAS

In this section, we list the main technical lemmas required to obtain the high-probability regret bound of `Neural-ES`. We present the proof of the technical lemmas in Appendix D.

First, we need the following common assumption and well-known result to demonstrate that our neural network $f(X; \theta)$ and its gradient $g(X; \theta_0)$ can accurately approximate the true reward model $h(X)$.

**Assumption C.1.** We use $\mathbf{H}$ to denote the neural tangent kernel (NTK) matrix on the context set. We assume that $\mathbf{H} \succeq \lambda_0 I$. Moreover, for any $X \in \mathcal{X}$, $\|X\|_2 \leq 1$ and $[X]_j = [X]_{j+d/2}$.

**Lemma C.2.** (Lemma 4.1 in Jia et al. (2022)) There exists a positive constant $\bar{C}$ such that for any $\delta \in (0, 1)$, with probability at least $1 - \delta$, when $N \geq \bar{C} K^4 L^6 \log(K^2 L/\delta)/\lambda_0^4$, there exists a $\theta^* \in \mathbb{R}^{d'}$ such that for all $X \in \mathcal{X}$,

$$h(X) = \langle g(X; \theta_0), \theta^* - \theta_0 \rangle = \langle \psi(X), \sqrt{N}(\theta^* - \theta_0) \rangle, \quad \sqrt{N} \|\theta^* - \theta_0\|_2 \leq \sqrt{2\mathbf{h}^\top \mathbf{H}^{-1} \mathbf{h}} \leq S,$$

where $\mathbf{H}$ is the NTK matrix defined on the context set $\mathcal{X}$ and $\mathbf{h} = \big( h(X_1), ..., h(X_K) \big)$, $S$ is the upper bound of $\sqrt{2\mathbf{h}^\top \mathbf{H}^{-1} \mathbf{h}}$.

Compare this result with linear contextual bandit, where the mean reward is given by $X^\top \theta^*$, we can see that when the neural network is wide enough, the mean reward at contexts $X \in \mathcal{X}$ can be well approximated by a linear function of $\psi(X)$. If our learned parameter $\theta_t$ is close to $\theta^*$, the distance between neural network approximation $f(X; \theta_t)$ and true mean reward $h(X)$ can be controlled by the distance $\big| f(X_t; \theta_t) - \langle g(X_t; \theta_0), \bar{\theta}_t \rangle \big|$ and $\big| \langle g(X_t; \theta_0), \bar{\theta}_t \rangle - h(X_t) \big|$. Following these intuitions, we need the following lemmas to provide concentration of these distances.

**Lemma C.3.** (Lemma 4.2 in Jia et al. (2022)) Define a sequence of events $\mathcal{E}_{1,t}$ as follows:

$$\mathcal{E}_{1,t} := \left\{ \forall X_t \in \mathcal{X}, \big| \langle g(X_t; \theta_0), A_t^{-1} \bar{\mathbf{b}}_t / \sqrt{N} \rangle - h(X_t) \big| \leq \alpha_t \|g(X_t; \theta_0)\|_{A_t^{-1}} \right\},$$

where $\alpha_t$ is a scalar scale of deviation. Then, for any $t \in [T]$, $\lambda > 0$ and $\delta > 0$, with $\alpha_t$ set as

$$\alpha_t = \sqrt{\sigma^2 \log\big( \det(A_t)/(\delta^2 \det(\lambda I)) \big)} + \sqrt{\lambda} S, \tag{C.3}$$

where $S$ is the upper bound of $\sqrt{2\mathbf{h}^\top \mathbf{H}^{-1} \mathbf{h}}$, we have $\mathbb{P}(\mathcal{E}_{1,t} \mid \mathcal{F}_{t-1}) \geq 1 - \delta$.

Under event $\mathcal{E}_{t,1}$, the distance $\big| \langle \psi(X_t), \bar{\theta}_t \rangle - h(X_t) \big|$ is bounded by $\alpha_t \|g(X_t; \theta_0)\|_{A_t^{-1}}$. Recall that we assume the random noise $\eta_t$ is $\sigma$-sub-Gaussian, thus $\alpha_t$ reflects the deviation caused by the noise in the observed reward. This result is valid for any algorithms in the neural contextual bandit setting, as it does not involve any perturbations. The estimate $\bar{\theta}_t$ is only based on the sequence of pulled arms $\{X_l\}_{l=1}^{t-1}$ and observed rewards $\{Y_l\}_{l=1}^{t-1}$. Next, we need the following lemma to upper bound the distance between neural network output and linear approximation $\big| f(X_t; \theta_{t-1}^j) - \langle \psi(X_t), \bar{\theta}_t \rangle \big|$. The following lemma is adapted from Lemma 4.3 in Jia et al. (2022) and the proof is mostly the same. Note that in our lemma, the probability $\mathbb{P}(\mathcal{E}_{2,t}^j)$ is not conditioned on the history of the perturbations.

**Lemma C.4.** For each model $j \in [m]$, define a sequence of events $\mathcal{E}_{2,t}^j$ as follows:

$$\mathcal{E}_{2,t}^j = \left\{ \forall X_t \in \mathcal{X}, \left| f(X_t; \theta_{t-1}^j) - \langle g(X_t; \theta_0), A_t^{-1}\bar{\mathbf{b}}_t/\sqrt{N} \rangle \right| \leq \epsilon(N) + \beta_t \|g(X_t; \theta_0)\|_{A_t^{-1}} \right\},$$

where $\beta_t$ is a scalar scale of the deviation, and $\epsilon(N)$ is defined as

$$\begin{aligned} \epsilon(N) = &C_{\epsilon,1}N^{-1/6}T^{2/3}\lambda^{-2/3}L^3\sqrt{\log N} + C_{\epsilon,2}(1 - \eta N\lambda)^J\sqrt{TL/\lambda} \\ &+ C_{\epsilon,3}N^{-1/6}T^{5/3}\lambda^{-5/3}L^4\sqrt{\log N}\left(1 + \sqrt{T/\lambda}\right), \end{aligned} \tag{C.4}$$

where $\eta$ is learning rate, $J$ is the number of steps for gradient descent in neural network learning, $\{C_i\}_{i=1}^3$ are constants. Then, there exist positive constants $\{C_i\}_{i=1}^3$, such that with step size $\eta$ and the neural network satisfy the following conditions:

$$\eta = C_1(N\lambda + NLT)^{-1},$$

$$N \geq C_2\sqrt{\lambda}L^{-3/2}\left[\log(TKL^2/\delta)\right]^{3/2},$$

$$N[\log N]^{-3} \geq C_3 \max\left\{ TL^{12}\lambda^{-1}, T^7\lambda^{-8}L^{18}(\lambda + LT)^6, L^{21}T^7\lambda^{-7}(1 + \sqrt{T/\lambda})^6 \right\},$$

and set $\beta_t$ as

$$\beta_t = \sigma_R\sqrt{4\log t + 2\log K}, \tag{C.5}$$

we have $\mathbb{P}(\mathcal{E}_{2,t}^j) \geq 1 - t^{-2}$.

Recall that the perturbations $Z_t^j$ are sampled from $\mathcal{N}(0, \sigma_R^2)$, thus $\beta_t$ reflects the deviation caused by perturbations. After uniformly randomly choosing model $j_t \in [m]$ from the ensemble, we further define the event for round $t$:

$$\mathcal{E}_{2,t} = \left\{ \forall X_t \in \mathcal{X}, \left| f(X_t; \theta_{t-1}^{j_t}) - \langle g(X_t; \theta_0), A_t^{-1}\bar{\mathbf{b}}_t/\sqrt{N} \rangle \right| \leq \epsilon(N) + \beta_t \|g(X_t; \theta_0)\|_{A_t^{-1}} \right\},$$

Then, at round $t$, we have the probability for $\mathcal{E}_{2,t}$ as:

$$\mathbb{P}(\mathcal{E}_{2,t}) = \sum_{j=1}^m \mathbb{P}(j_t = j, \mathcal{E}_{2,t}^j) \geq 1 - \frac{1}{t^2}.$$

Next, we present the optimism condition (anti-concentration) in the following two lemmas. First, we fix the sequence of pulled arms $\{X_l\}_{l=1}^t$ and model $j \in [m]$, the only source of randomness is the perturbations $\{Z_l^j\}$, we can apply tail bound of Gaussian distribution to lower bound the probability of optimism. Next, we add randomness of choosing arm $j_t$ at round $t$, and apply the Azuma-Hoeffding inequality. Finally, we count equivalent sequences of $\{X_l\}_{l=1}^t$ and prove the probability of optimism considering all randomness of the procedure.

**Lemma C.5.** Fix the model $j \in [m]$. For any $t \in [T]$, fix the sequence of pulled arms $\{X_l\}_{l=1}^t$. Define a sequence of events $\mathcal{E}_{3,t}^j$ as follows:

$$\mathcal{E}_{3,t}^j := \left\{ f(X_t; \theta_{t-1}^j) \geq h(X^*) - \epsilon(N) \right\}.$$

Then, by choosing variance of perturbations $\sigma_R$ as

$$\sigma_R = \alpha_T(1 - \lambda\lambda_K^{-1}(A_K))^{-1/2}, \tag{C.6}$$

where $A_K$ is the covariance matrix constructed after the initial $K$-round arm pulling, and $\lambda_K(A_K)$ represents the $K$-th largest eigenvalue of matrix $A_K$, we have $\mathbb{P}(\mathcal{E}_{t,3}^j) \geq (4e\sqrt{\pi})^{-1} =: p_N'$.

**Lemma C.6.** Fix the ensemble size $m$ as

$$m \geq \frac{2}{p_N'^2}\left( K\log T + \log\frac{1}{\delta} \right).$$

Define a sequence of events $\mathcal{E}_{3,t}$ as follows:

$$\mathcal{E}_{3,t} := \left\{ f(X_t; \theta_{t-1}^{j_t}) \geq h(X^*) - \epsilon(N) \right\}.$$

Then, for any $t \in [T]$, we have $\mathbb{P}(\mathcal{E}_{3,t}) \geq p_N'/2$.

Finally, we need the following lemma to calculate the summation.

**Lemma C.7** (Lemma 4.6 in Jia et al. (2022)). With $\widetilde{d}$ as the effective dimension of the NTK matrix **H**, we have

$$\sum_{t=1}^{T} \min\left\{ ||g(X_t; \theta_0)/\sqrt{N}||_{A_{t-1}^{-1}}^2, 1 \right\} \leq 2\left(\widetilde{d}\log(1 + TK/\lambda) + 1\right). \tag{C.7}$$

Note that by setting $\lambda \geq \max\{1, S^{-2}\}$, we have

$$||g(X_t; \theta_0)/\sqrt{N}||_{A_{t-1}^{-1}}^2 \leq \frac{1}{\lambda}||g(X_t; \theta_0)/\sqrt{N}||_2^2 \leq 1,$$

thus we can apply this lemma to compute the cumulative regret.

## C.3 Regret Analysis

From the design of the algorithm, we use the first $K$ rounds to pull each arm once, thus the cumulative regret can be decomposed as (assuming that $T > K$):

$$R(T) \leq \sum_{t=K+1}^{T} \left(h(X^*) - h(X_t)\right) + K.$$

Next, we analyze the per-round regret bound for $t \in [K + 1, T]$. We assume that $\mathcal{E}_{1,t}$ holds for the rest of the analysis. For each round $t$, the regret can be written as

$$h(X^*) - h(X_t) \leq \left(h(X^*) - h(X_t)\right) \mathbb{1}\{\mathcal{E}_{t,2}\} + \mathbb{P}\left(\bar{\mathcal{E}}_{t,2}\right),$$

where we used the assumption that $h(X)$ in bounded in $[0, 1]$. From the technical lemmas, $\mathbb{P}\left(\bar{\mathcal{E}}_{t,2}\right) \sim t^{-2}$, thus we only need to focus on $\left(h(X^*) - h(X_t)\right)$ under event $\mathcal{E}_{t,1} \cap \mathcal{E}_{t,2}$.

For the following analysis, we introduce the set of sufficiently sampled arms in round $t$:

$$\Omega_t = \left\{\forall X_t \in \mathcal{X} : 2\epsilon(N) + (\alpha_t + \beta_t)||g(X_t; \theta_0)||_{A_t^{-1}} \leq h(X^*) - h(X_t)\right\}.$$

We also define the set of under-sampled arms $\bar{\Omega}_t = \mathcal{X}\backslash\Omega_t$. From the definition, the per-round regret of $X_t \in \bar{\Omega}_t$ is bounded by $\left(2\epsilon(N) + (\alpha_t + \beta_t)||g(X_t; \theta_0)||_{A_t^{-1}}\right)$, we expect that the pulled arm $X_t$ should come from $\bar{\Omega}_t$ with high probability as the algorithm proceeds. We further define the least uncertain and under-sampled arm $X_t^{(e)}$ at round $t$ as

$$X_t^{(e)} = \text{argmin}_{X \in \bar{\Omega}_t}||g(X; \theta_0)/\sqrt{N}||_{A_t^{-1}}. \tag{C.8}$$

Now we can write the per-round regret

$$\begin{aligned}
&\left(h(X^*) - h(X_t)\right) \mathbb{1}\{\mathcal{E}_{2,t}\} \\
=&\left(h(X^*) - h(X_t^{(e)}) + h(X_t^{(e)}) - h(X_t)\right) \mathbb{1}\{\mathcal{E}_{2,t}\} \\
\leq&\left(2\epsilon(N) + (\alpha_t + \beta_t)||g(X_t; \theta_0)||_{A_t^{-1}}\right) + \left(h(X_t^{(e)}) - h(X_t)\right) \mathbb{1}\{\mathcal{E}_{2,t}\} \\
\leq&4\epsilon(N) + (\alpha_t + \beta_t)\left(2||g(X_t^{(e)}; \theta_0)/\sqrt{N}||_{A_t^{-1}} + ||g(X_t; \theta_0)/\sqrt{N}||_{A_t^{-1}}\right).
\end{aligned}$$

From the per-round regret bound, we should upper bound $||g(X_t^{(e)}; \theta_0)/\sqrt{N}||_{A_t^{-1}}$ using expression of $||g(X_t; \theta_0)/\sqrt{N}||_{A_t^{-1}}$, then we can apply Lemma C.7 to compute the cumulative regret. We have the following result

$$\mathbb{E}\left[||g(X_t; \theta_0)/\sqrt{N}||_{A_t^{-1}}\right] \geq ||g(X_t^{(e)}; \theta_0)/\sqrt{N}||_{A_t^{-1}}\mathbb{P}\left(X_t \in \bar{\Omega}_t\right),$$

$$\Rightarrow ||g(X_t^{(e)}; \theta_0)/\sqrt{N}||_{A_t^{-1}} \leq \frac{\mathbb{E}\left[||g(X_t; \theta_0)/\sqrt{N}||_{A_t^{-1}}\right]}{\mathbb{P}(X_t \in \bar{\Omega}_t)}.$$

As mentioned previously, we should lower bound the probability of $\mathbb{P}(X_t \in \bar{\Omega}_t)$.

$$\begin{aligned}
\mathbb{P}(X_t \in \bar{\Omega}_t) &\geq \mathbb{P}\big(\exists X \in \bar{\Omega}_t : f(X; \theta_t) > \max_{X' \in \Omega_t} f(X'; \theta_t)\big) \\
&\geq \mathbb{P}\big(f(X^*; \theta_t) > \max_{X' \in \Omega_t} f(X'; \theta_t), \mathcal{E}_{2,t}\big) \\
&\geq \mathbb{P}\big(f(X^*; \theta_t) > h(X^*) - \epsilon(N)\big) - \mathbb{P}(\bar{\mathcal{E}}_{2,t}) \\
&\geq \mathbb{P}(\mathcal{E}_{3,t}) - \mathbb{P}(\bar{\mathcal{E}}_{2,t}).
\end{aligned}$$

Therefore, the per-round regret can be bounded as

$$\mathbb{E}\big[h(X^*) - h(X_t)\big]$$
$$\leq \mathbb{P}(\bar{\mathcal{E}}_{2,t}) + 4\epsilon(N) + (\alpha_t + \beta_t)\Big(1 + \frac{2}{\mathbb{P}(\mathcal{E}_{3,t}) - \mathbb{P}(\bar{\mathcal{E}}_{2,t})}\Big)||g(X_t; \theta_0)/\sqrt{N}||_{A_t^{-1}}.$$

Now we sum up the per-round regret bound and compute the cumulative regret $R(T)$. From Lemma C.3, $\mathcal{E}_{1,t}$ holds for all $t \in [T]$ with probability at least $1 - \delta$. Taking $\delta = 1/T$, we can compute the summation under as follows.

$$\mathbb{E}\big[R(T)\big] \leq 1 + K + 4T\epsilon(N)$$
$$+ \sum_{t=K+1}^{T} \mathbb{P}(\bar{\mathcal{E}}_{2,t}) + (\alpha_t + \beta_t)\Big(1 + \frac{2}{\mathbb{P}(\mathcal{E}_{3,t}) - \mathbb{P}(\bar{\mathcal{E}}_{2,t})}\Big)||g(X_t; \theta_0)/\sqrt{N}||_{A_t^{-1}}.$$

For $\mathbb{P}(\bar{\mathcal{E}}_{2,t})$, we have the following bound:

$$\sum_{t=K+1}^{T} \mathbb{P}(\bar{\mathcal{E}}_{2,t}) \leq \sum_{t=K+1}^{T} \frac{1}{t^2} \leq \frac{\pi^2}{6}.$$

From $\mathbb{P}(\mathcal{E}_{3,t}) \geq p'_N/2$, we have

$$1 + \frac{2}{\mathbb{P}(\mathcal{E}_{3,t}) - \mathbb{P}(\bar{\mathcal{E}}_{2,t})} \leq 1 + \frac{8}{p'_N}.$$

Combining these results, we have

$$\mathbb{E}\big[R(T)\big] \leq K + 4T\epsilon(N) + \frac{\pi^2}{6} + 1 + \Big(1 + \frac{8}{p'_N}\Big)(\alpha_T + \beta_T) \sum_{t=K+1}^{T} ||g(X_t; \theta_0)/\sqrt{N}||_{A_t^{-1}}$$

$$\leq K + 4T\epsilon(N) + \frac{\pi^2}{6} + 1 + \Big(1 + \frac{8}{p'_N}\Big)(\alpha_T + \beta_T)\sqrt{T}\sqrt{\sum_{t=1}^{T} ||g(X_t; \theta_0)/\sqrt{N}||_{A_t^{-1}}^2}$$

$$\leq K + 4T\epsilon(N) + \frac{\pi^2}{6} + 1 + \Big(1 + \frac{8}{p'_N}\Big)(\alpha_T + \beta_T)\sqrt{2\widetilde{d}T\log(1 + TK/\lambda) + 2T}.$$

With parameters $\eta = C_1(NTL + N\lambda)^{-1}$, $N[\log N]^{-3} \geq C_3 \max\{TL^{12}\lambda^{-1}, T^7\lambda^{-8}L^{18}(\lambda + LT)^6, L^{21}T^7\lambda^{-7}(1 + \sqrt{T/\lambda})^6\}$ and $\lambda \geq \max\{1, S^{-2}\}$, we have $T\epsilon(N) \sim \widetilde{\mathcal{O}}(1)$, $\alpha_T, \beta_T \sim \widetilde{\mathcal{O}}(\widetilde{d}^{1/2})$. Therefore, the regret bound $\mathbb{E}\big[R(T)\big] = \widetilde{\mathcal{O}}(\widetilde{d}\sqrt{T})$.

# D   PROOF OF TECHNICAL LEMMAS IN NEURAL-ES

## D.1   PROOF OF LEMMA C.5

Under event $\mathcal{E}_{1,t} \cap \mathcal{E}_{2,t}^j$, we have

$$h(X_t) \leq \langle g(X_t; \theta_0), A_t^{-1}\bar{\mathbf{b}}_t/\sqrt{N}\rangle + \alpha_t||g(X_t; \theta_0)/\sqrt{N}||_{A_t^{-1}}.$$

Therefore, we have the following sufficient condition for optimism $f(X_t; \theta_t^j) > h(X^*) - \epsilon(N)$:

$$\langle g(X_t; \theta_0), A_t^{-1}\mathbf{b}_t/\sqrt{N}\rangle \geq \langle g(X_t; \theta_0), A_t^{-1}\bar{\mathbf{b}}_t/\sqrt{N}\rangle + \alpha_t||g(X_t; \theta_0)/\sqrt{N}||_{A_t^{-1}}. \qquad \text{(D.1)}$$

From the definitions of covariance matrix and auxiliary vectors, we have

$$\langle g(X_t; \theta_0), A_t^{-1}(\mathbf{b}_t - \bar{\mathbf{b}}_t)/\sqrt{N}\rangle = \frac{1}{N}\sum_{l=1}^{t} Z_l^j g(X_l; \theta_0)^\top A_t^{-1} g(X_l; \theta_0) =: U_t. \tag{D.2}$$

Fix the sequence of pulled arms $\{X_l\}_{l=1}^t$, then the only randomness comes from the perturbation sequence $\{Z_l^j\}_{l=1}^t$. Since the perturbations are i.i.d., we can use the following tail bound:

$$\mathbb{P}\Big(\frac{X-\mu}{\sigma} > \beta\Big) \geq \frac{\exp(-\beta^2)}{4\sqrt{\pi}\beta}.$$

The mean of $U_t$ is zero, the variance is bounded by

$$\begin{aligned}
\mathrm{Var}[U_t] &= \sigma_R^2\Big(\sum_{l=1}^{t}\Big(\frac{1}{N}g(X_l;\theta_0)^\top A_t^{-1} g(X_l;\theta_0)\Big)^2\Big)\\
&= \sigma_R^2\|g(X_l;\theta_0)/\sqrt{N}\|_{A_t^{-1}}^2 - \lambda\sigma_R^2\|g(X_l;\theta_0)/\sqrt{N}\|_{A_t^{-2}}^2\\
&\geq \sigma_R^2(1 - \lambda\lambda_{\min}^{-1}(A_t))\|g(X_t;\theta_0)/\sqrt{N}\|_{A_t^{-1}}^2\\
&\geq \sigma_R^2(1 - \lambda\lambda_K^{-1}(A_K))\|g(X_t;\theta_0)/\sqrt{N}\|_{A_t^{-1}}^2.
\end{aligned}$$

Therefore, by choosing variance of perturbation

$$\sigma_R = \alpha_t\big(1 - \lambda\lambda_K^{-1}(A_K)\big)^{-1/2},$$

we have the probability of optimism:

$$\mathbb{P}\Big(f(X_t;\theta_t) > h(X^*) - \epsilon(N)\Big) \geq \mathbb{P}\Big(U_t > \alpha_t\|g(X_t;\theta_0)/\sqrt{N}\|_{A_t^{-1}}\Big) \geq \frac{1}{4e\sqrt{\pi}}. \tag{D.3}$$

### D.2 Proof of Lemma C.6

Define random variable $I_t^j$ for round $t$ and model $j$ as $I_t^j = \mathbb{1}\{\mathcal{E}_{3,t}^j\}$. Then, from previous technical lemmas, we have $\mathbb{P}(I_t^j = 1) \geq p_N'$. For each round $t$, we further define $I_t = I_t^{j_t}$. After uniformly choosing the model $j \in [m]$, we have

$$\mathbb{P}(I_t = 1) = \frac{1}{m}\sum_{j=1}^{m}\mathbb{P}(I_t^j = 1). \tag{D.4}$$

Apply Azuma-Hoeffding inequality, we have the following probability of optimism:

$$\mathbb{P}\Big(\frac{1}{m}\sum_{j=1}^{m} I_t^j < \frac{p_N'}{2}\Big) \leq \exp\Big(-\frac{p_N'^2 m}{2}\Big). \tag{D.5}$$

Following the discussions in Lee & Oh (2024), we have the following lemma, which is adapted from Claim 1 in the paper.

**Lemma D.1.** There exists an event $\mathcal{E}^*$ such that under $\mathcal{E}^*$, $\frac{1}{m}\sum_{j=1}^{m} I_t^j \geq p_N'/2$ holds for all $t \in [T]$, and $\mathbb{P}(\bar{\mathcal{E}}^*) \leq T^K \exp(-p_N'^2 m/2)$.

Therefore, setting ensemble size as follows:

$$m \geq \frac{2}{p_N'^2}\Big(K\log T + \log\frac{1}{\delta}\Big),$$

then we have $\mathbb{P}(\mathcal{E}^*) \geq 1 - \delta$. In summary, by setting the ensemble size $m = \Omega(K\log T)$, we have

$$\mathbb{P}(\mathcal{E}_{3,t}) = \mathbb{P}\big(f(X_t;\theta_{t-1}^{j_t}) \geq h(X^*) - \epsilon(N)\,|\,\mathcal{F}_{t-1}, \mathcal{E}_{t,1}\big) \geq p_N'/2. \tag{D.6}$$

# E    ANALYSIS OF DOUBLING TRICK

In this section, we provide theoretical analysis of regret bound with doubling trick. To preserve the asymptotic regret bound of ensemble sampling, we need to properly set the schedule $\{T_i\}$. We need the following result of the regret bound of doubling trick.

**Lemma E.1** (Theorem 4 in Besson & Kaufmann (2018)). If an algorithm $\mathcal{A}$ satisfies $R_T(\mathcal{A}_T) \leq cT^\gamma (\log T)^\delta + f(T)$, for $0 < \gamma < 1$, $\delta \geq 0$ and for $c > 0$, and an increasing function $f(t) = o(t^\gamma (\log t)^\delta)$ (at $t \rightarrow \infty$), then the anytime version $\mathcal{A}' := \mathcal{DT}(\mathcal{A}, (T_i)_{i \in \mathbb{N}})$ with the geometric sequence $(T_i)_{i \in \mathbb{N}}$ of parameters $T_0 \in \mathbb{N}^*$, $b > 1$ (i.e., $T_i = \lfloor T_0 b^i \rfloor$) with the condition $T_0(b-1) > 1$ if $\delta > 0$, satisfies,

$$R_T(\mathcal{A}') \leq l(\gamma, \delta, T_0, b)cT^\gamma (\log T)^\delta + g(T),$$

with a increasing function $g(t) = o(t^\gamma (\log t)^\delta)$, and a constant loss $l(\gamma, \delta, T_0, b) > 1$,

$$l(\gamma, \delta, T_0, b) := \left( \frac{\log(T_0(b-1)+1)}{\log(T_0(b-1))} \right)^\delta \times \frac{b^\gamma (b-1)^\gamma}{b^\gamma - 1}.$$

With Lemma E.1, the regret bound analysis is straightforward. From the regret bound of ensemble sampling, the dependency on $T$ is given by $\mathcal{O}\left( (\log T)^{\frac{3}{2}} \sqrt{T} \right)$. Therefore, apply Lemma E.1 to ensemble sampling algorithms (GLM-ES or Neural-ES), we have $\gamma = 1/2$, $\delta = 3/2$. We can minimize the expression of $l(\gamma, \delta, T_0, b)$ by properly selecting the parameter of the doubling sequence $T_0$ and $b$. The optimal choice of $b$ is $(3 + \sqrt{5})/2 \approx 2.6$ and we can choose $T_0$ large enough to reduce the other factor. For example, if we choose $T_0 = 100$, the extra factor $l(\gamma, \delta, T_0, b) \approx 3.3$.

From the discussions above, we can choose the sequence

$$\{T_i\} = \{T_0, \, T_0 b, \, T_0 b^2, ...\}.$$

The number of rounds follows the sequence:

$$\{\tau_i\} = \{T_0, \, T_0(b-1), \, T_0(b-1)b, \, T_0(b-1)b^2, ...\}.$$

From the discussions above, by directly applying doubling trick, we obtain the same asymptotic regret bound with the cost of a constant factor.

# F    ADDITIONAL EXPERIMENTS

In this section, we present additional experiment results to explain how the performance of ensemble sampling is affected by hyper-parameters of the algorithm and the environment.

First, we consider the performance of GLM-ES and Neural-ES with different ensemble size, regularization, perturbation distribution and network structure. The experiment results are plotted in Figure 2. For GLM-ES, our baseline algorithm uses the following parameters: we use 100 iterations of gradient descent with step size 0.01, ensemble size $m = 10$, perturbation distribution $\mathcal{N}(0, \sigma_R^2)$ with $\sigma_R = 0.1$, warm-up steps $\tau = 500$ and regularization $\lambda = 1.0$. For Neural-ES, our baseline algorithm uses the following parameters: we use fully connected neural network with $L = 3$ layers and width $N = 20$, we optimize the loss function using gradient descent with 100 steps and learning rate 0.01, ensemble size $m = 10$, perturbation distribution $\mathcal{N}(0, \sigma_R^2)$ with $\sigma_R = 0.1$, warm-up steps $\tau = 50$ and regularization $\lambda = 1.0$. To study the effects of $m$, $\lambda$, $\sigma_R$ and network structure, in each experiment in Figure 2, we change one hyper-parameter while keeping other parameters the same as the baseline algorithm. In Figure 2, the baseline algorithm is plotted in gray solid line, additional experiment results are plotted in dashed lines. We discuss the experiment results and effect of each hyper-parameter as follows.

(i) Ensemble size $m$. For very small ensemble size $m$, it is likely that most models are trapped in sub-optimal arms, thus the cumulative regret grows fast for large $T$ compared to experiments with moderate or large ensemble sizes. We also observe higher variance for $m = 5$, thus the algorithm becomes unstable at very small ensemble size. For very large ensemble size ($m = 50$), we observe marginal improvements on the cumulative regret compared to $m = 10$. Since the computational

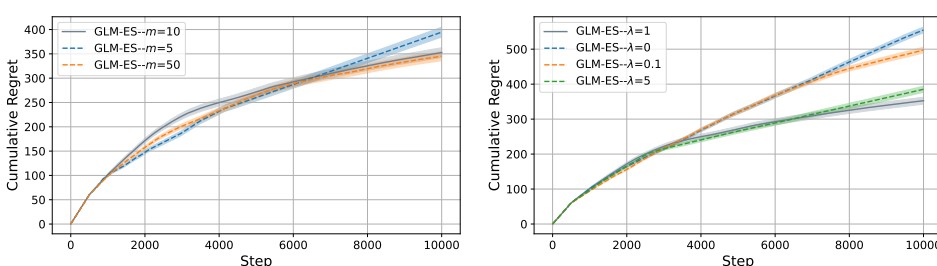

(a) Logistic bandit environment with different ensemble size $m$.

(b) Logistic bandit environment with different regularization $\lambda$.

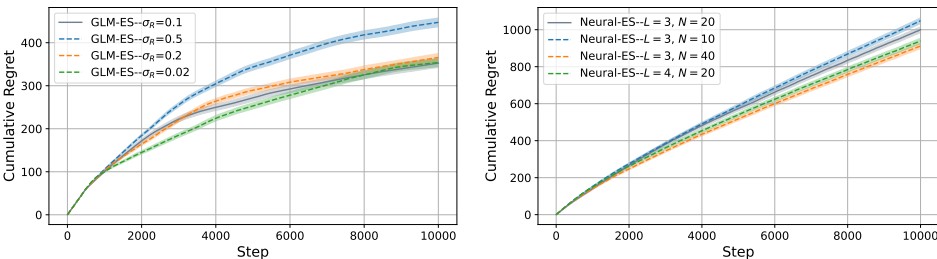

(c) Logistic bandit environment with different perturbation variance $\sigma_R$.

(d) Quadratic bandit environment with different network structure.

Figure 2: Experiment results with different parameter setup and model structures.

cost grows linearly with $m$, moderate ensemble size such as $m = 10$ suffices to reach competitive performance for `GLM-ES`.

(ii) Regularization $\lambda$. For very small regularization $\lambda \sim 0$, the MLE estimates of the true parameter becomes unstable. Therefore, the predictions of reward become noisy and could cause unnecessary exploration. For very large regularization, the loss function is dominated by the regularization term and the estimated parameter becomes very small. This could cause estimated rewards to be very similar across different arms, making identifying the optimal arm more difficult. From the experiment results, $\lambda \sim 1$ is a proper choice, the cumulative regret becomes higher when regularization is too small or too large.

(iii) Perturbation variance $\sigma_R$. For very small perturbation variance $\sigma_R$, the algorithm behaves greedy and the perturbed rewards for each model are nearly identical. Without sufficient exploration from perturbations, the algorithm is likely to be trapped in sub-optimal arms, resulting in fast growth of cumulative regret for large $T$. For very large perturbations, the true reward is dominated by random perturbations and the model cannot effectively learn the true parameters of the environment. This results in large cumulative regret, especially at small $t$. From the experiment results, $\sigma_R \sim 0.1$ is a proper choice of perturbation variance that provides sufficient exploration without introducing too much noise.

(iv) Network structure $L$ and $N$. For very small neural networks, the model has limited capacity to approximate the true reward model, the cumulative regret grows fast compared to larger networks. For very large networks, while the neural network can accurately approximate the true reward model, with limited data from interactions, the model could overfit the random noise and added perturbations. From experiments results, the performance is not significantly improved with larger network structures. We also note that expanding the width of the neural network is generally more effective than adding more layers. This agrees with our theoretical analysis. As in Theorem 5.7, wider network can considerably reduce the cumulative regret.

Next, we consider the performance of anytime versions of ensemble sampling with different schedules $b$ and $T_0$. The experiment results are plotted in Figure 3. From theoretical analysis, for anytime

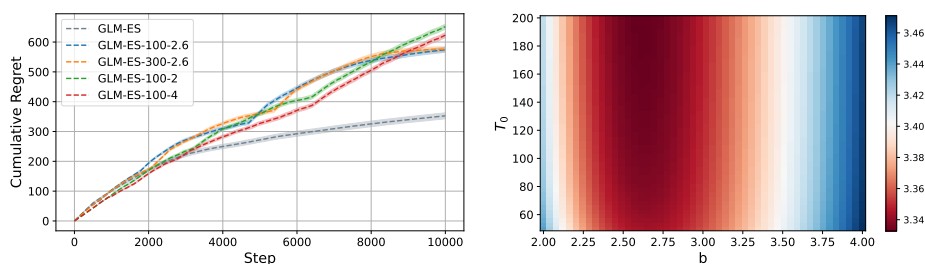

(a) Logistic bandit with different anytime schedules (GLM-ES-$T_0$-$b$).

(b) Constant factor $l(T_0, b)$ in doubling trick.

Figure 3: Experiment results with different anytime schedules.

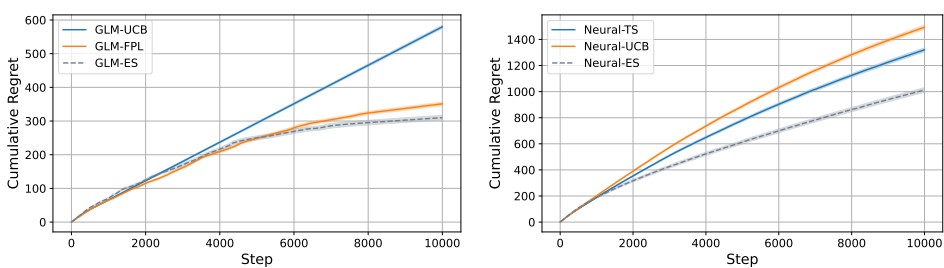

(a) Logistic bandit with larger feature spaces.

(b) Quadratic bandit with larger feature spaces.

Figure 4: Experiment results with larger feature spaces.

versions of ensemble sampling, the regret bound increases by a constant factor $l(T_0, b)$:

$$l(T_0, b) := \left( \frac{\log(T_0(b-1) + 1)}{\log(T_0(b-1))} \right)^{\delta} \times \frac{b^{\gamma}(b-1)^{\gamma}}{b^{\gamma} - 1},$$

where $\gamma = 1/2$, $\delta = 3/2$. The value of $l(T_0, b)$ is plotted in Figure 3. From this plot, the regret bound becomes smaller as we choose higher $T_0$ and $b = (3 + \sqrt{5})/2 \approx 2.6$. We also observe that $T_0$ has minor effect on the performance of the algorithm, while moving away from the optimal value of $b$ can considerably cause greater cumulative regret. This agrees with the experiment results. Setting different $T_0$ results in nearly identical cumulative regret, while setting $b$ away from $(3 + \sqrt{5})/2$ has more significant impact on the performance of the algorithm. Therefore, for practicality, we should keep $b$ at the optimal value $b = (3 + \sqrt{5})/2 \approx 2.6$ and set $T_0$ in the order of $10^2$.

Finally, we use synthetic environment with higher-dimensional features and more arms in the arm set $\mathcal{X}$. Specifically, we use $d = 50$, $K = 500$ in the experiments. The results are plotted in Figure 4. From the experiment results, the ensemble sampling algorithms remain competitive in larger feature spaces compared to baseline algorithms.

## USAGE OF LLM

All ideas and research are conducted by the authors, and the paper itself is written by the authors. The LLM is used as a tool for polishing the written content of the paper and checking the grammar errors.