# OpenReview forum: "Provable Anytime Ensemble Sampling Algorithms in Nonlinear Contextual Bandits"
_ICLR.cc/2026/Conference — Submitted to ICLR 2026_

### Official Review · Reviewer_Pgxe · 2025-10-30

**Soundness:** 3
**Presentation:** 3
**Contribution:** 2
**Rating:** 6
**Confidence:** 5

**Summary:**

This paper presents a unified ensemble sampling framework for nonlinear contextual bandits under two settings: GLM-ES for generalized linear bandits and Neural-ES for neural bandits. The authors provide regret analysis and empirical results on the proposals.

**Strengths:**

1. Clear problem setting, unified algorithm template (Alg. 1), and concrete GLM/Neural instantiations.

2. The anytime design: Doubling-trick conversion with constant-factor loss, addressing unknown horizon.

**Weaknesses:**

Some over-claims or lack of novelty. The algorithms adapt known ensemble-/perturbed-history ideas to GLM/neural settings; much of the value is synthesis + anytime conversion rather than a fundamentally new exploration principle. (Claims of “first” high-probability bounds are credible but incremental.)

**Questions:**

1. Anytime hyperparameters: For the doubling-trick variants, how sensitive are results to $T_0$ and base $b$? Any rule-of-thumb for choosing them beyond the theory constant?

2. Scalability with large $K$ or even continuous actions: Can the framework extend to large or continuous action sets (e.g., via approximate maximization or feature maps), and would the regret proofs still go through?

3. Warm-up guidance (GLM-ES): Beyond the theoretical construction, what’s a simple recipe (iterations, step sizes) that preserves optimism with high probability in practice?

---

> ### Author Response · Authors · 2025-11-29
>
> ## Response to Reviewer Pgxe
>
>
> We thank the reviewer for your valuable time and effort in providing detailed feedback on our work. We hope our response will fully address all your questions.
>
> ---
> ### **Q1:** Explanation for novelty in algorithm design and theoretical results
>
> **A1:** In this paper, we proposed a general framework for ensemble sampling in bandit settings, which significantly expanded the applications of this algorithm design. Compared to existing works on ensemble sampling in linear bandit settings, we introduced new technical methods to overcome the difficulties in nonlinear bandit settings, which contributes to the research of nonlinear bandit problems in general.
>
>
> ---
> ### **Q2:** Additional experiments on sensitivity/ablation for anytime hyperparameters $T_0$ and $b$
>
> **A2:** We added detailed explanations of how the theoretical guarantee changes with different anytime schedules and additional experiments with the following parameters:
> $(T_{0}=100, b=2), (T_{0}=100, b=4)$.
> The results and discussions are added as Appendix F.
> In general, the regret bound becomes smaller as we choose higher $T_{0}$ and $b = (3 + \sqrt{5})/2 \approx 2.6$. Also, $T_{0}$ has minor effect on the performance of anytime ensemble sampling, while $b$ can significantly affect the performance.
> For practicality, we should keep $b$ at the optimal value $b = (3 + \sqrt{5})/2 \approx 2.6$ and set $T_{0}$ in the order of $10^{2}$.
>
>
> ---
> ### **Q3:** Can the framework extend to large or continuous action sets (e.g., via approximate maximization or feature maps), and would the regret proofs still go through?
>
> **A3:** While the algorithmic framework itself can be adapted to continuous action sets (e.g., via approximate maximization), our current theoretical analysis relies on the assumption of a finite action set and does not extend to the continuous setting. Specifically, our proof technique requires an ensemble size of order $m=\Omega(K \log T)$. This dependency arises because we establish the probability of optimism by counting equivalent sequences of pulled arms, an argument that does not hold for infinite or continuous action spaces. We recognize this as a significant theoretical challenge and leave the development of regret bounds for continuous nonlinear ensemble sampling as a direction for future work.
>
> ---
> ### **Q4:**  What's a simple recipe for warm-up procedure in GLM-ES that preserves optimism with high probability in practice?
>
> **A4:** A practical and simple recipe for the warm-up procedure in GLM-ES is to replace the theoretical G-optimal design with uniform random sampling of arms for approximately $\tau \approx 25 d$ rounds, which is what we use in experiments. This approach empirically ensures that the minimum eigenvalue of the feature covariance matrix is sufficiently large, guaranteeing that the matrix is invertible and the estimator is well-defined. This simple strategy avoids the computational complexity of theoretical rounding algorithms while successfully establishing the initial stability required for the ensemble to maintain optimism in practice.
>
>
> ---
> We hope we have addressed all of your questions. If you have any further questions, we would be happy to answer them and if you don’t, would you kindly consider increasing your score?

---

### Official Review · Reviewer_j3cM · 2025-10-31

**Soundness:** 1
**Presentation:** 1
**Contribution:** 2
**Rating:** 2
**Confidence:** 4

**Summary:**

The paper proposes an application of ensemble sampling (Lu & Van Roy, 2017) to GLM bandits and neural bandits.
For the bandit problems with a fixed arm set of size $K$, the proposed method uses $O(K \log T)$ estimators in the ensemble and achieves a $\tilde{O}(d^{3/2}\sqrt{T} + d^{4.5})$ regret bound for GLM bandits and a $\tilde{O}(\tilde{d}\sqrt{T})$ regret bound for neural bandits.

**Strengths:**

This is the first theoretical analysis of ensemble sampling beyond the linear contextual bandit setting.

**Weaknesses:**

### 1. Proof requires further clarification

For GLM-ES:

- In Section B.3, the analysis begins with $H_ t$ but ends with $\bar{H}_ t$. The switch happens in line 1280 without any justification. Until line 1280, the analysis requires properties of $H_ t$, and after line 1280, the analysis requires $\bar{H}_ t$, so it doesn't seem to be a simple typo.

- The analysis, especially in Appendix B.4, requires $\sigma_ R \ge b_ 1 + b_ 2 R_ t \sigma_ R$. Tracking the definitions, it seems like $b_ 1 = \tilde{\mathcal{O}}(d)$ and hence $\sigma_ R \gtrsim d$ must hold. However, line 1433 assumes $\sigma_ R \approx (d \log T)^{\frac{1}{2\varepsilon}}$ with $\varepsilon = \frac{2}{3}$, which implies $\sigma_ R \approx (d \log T)^{\frac{3}{4}}$.
The choice of $\varepsilon = \frac{2}{3}$ is also curious, since from Eq. in line 1436, $\varepsilon \rightarrow 1$ would yield a better bound on the warm-up time, although it is still invalid due to the reason above.

- For analyses that use stochastic optimism, one has to show that the conditional probability of optimism is bounded below, conditioned on the history up to the previous time step. However, this paper only shows the lower bound of the total probability. Especially in Appendix B.4, I think the relationship between the shown facts and the conditional probability of optimism should be explained more clearly.

- Lemma A.7 is not proved.

- In line 1055, while the previous equation proves $\ddot{b}(c) \le e \ddot{b}(u)$, it suddenly claims $\ddot{b}(c) \le 2 \ddot{b}(u)$.

- In line 1330, I don't see why $\mathbb{P}(U_ t^\top \mathbf{Z}_ t^j \ge (b_ 1 + b_ 2 R_ t \sigma) \lVert U_ t \rVert_ 2) - \frac{\delta}{T} \ge \frac{p_ N}{2}$ should be true when no relationship between $\delta, T$ and $p_ N$ is given.

For Neural-ES:

- I don't think the proof of Lemma C.4 would be mostly the same as Jia et al. (2022), since the perturbation is not sampled freshly as in Jia et al. (2022).

- The last steps of Appendix C require corrections. $\sum_ {t=K+1}^T \lVert g(X_ t; \theta_ 0) / \sqrt{N} \rVert_ {A_ t^{-1}}$ is not bounded by $\tilde{d}(\log (1 + TK / \lambda) + 1)$.
In the next step, the inequality suddnely jumps to $\tilde{O}(\tilde{d}\sqrt{T})$.
The relationship between $\alpha_ T, \beta_ T$ and $\tilde{d}$ is not clearly shown.

- Line 1617  would make sense only when there is a conditional expectation on the left-hand side with conditional probability on the right-hand side. Even under the expectation, only the lower bound on the total probability is provided and not on the conditional probability.

- Eq. in line 1680 is given without any justification.

- In the proof of Lemma C.5, Eq in line 1666 should present the upper bound of $h(X_ t)$ not the lower bound for the following logic that ensures optimism.

Due to these issues, I am not convinced that the theorems are true.

---

### 2. Slightly misleading presentation

While I acknowledge that some terms could be used loosely, I find the current title, abstract, and the introduction slightly misleading.

- "Contextual bandits" commonly refers to the setting where the reward of the arm, or the arm set itself, changes at every time step. For instance, "linear bandits" would refer to the fixed-arm-set case and "linear contextual bandits" would refer to the changing arm-set case (Lattimore and Szepesvari, 2020). In this work, the contextual setting is introduced as the changing arm-set setting with the arm set denoted as $\mathcal{X}_ t$ in Section 3 Problem Setting. However, it is suddenly replaced by $\mathcal{X}$ starting from line 255, where the G-optimal design is defined, then Theorems 5.5 and 5.7 also assume a fixed finite arm set $|\mathcal{X}| = K$. I don't think the paper should state that they consider the contextual setting.

- One of the contributions of the paper is proposing an anytime version of ensemble sampling using the doubling trick. However, using the doubling trick for ensemble sampling is already discussed in Remark 3 of Janz et al. (2024a).

- While a lot of works use the term "nonlinear" for the setting with a general function class, this work is limited to GLMs and functions with a small NTK norm. I acknowledge that this point is clear in the abstract and the introduction, and the term could be used flexibly, so this is a minor concern compared to the previous ones.

---

### 3. Missing/misstated definitions

- $\mathcal{X}$ (without the subscript $t$) is not defined.

- $\mathcal{F}_ t$ is not defined within the paper.

- $\eta$ and $J$ are not defined in Lemma C.4. This is because the lemma is adapted from Jia et al. (2022) which trains the neural network with $J$ gradient descents of step size $\eta$, whereas this paper simply finds the minimizer of the loss function.

- The problem setting for GLM states that $Y_ t = \mu(X_ t^\top \theta^* ) + \eta$ with sub-Gaussian $\eta$, however the analysis seems to require that $Y_ t$ follows the natural exponential family distribution and not any sub-Gaussian noise, especially in Appendix A.1.

- At the end of Lemma C.4 (line 1539-1540), the definitions of $A_ K$ and $\lambda_ K$ are given, but are not used within the lemma. I think they should be at the end of Lemma C.5, or the defintion of $\sigma_ R$ in Lemma C.5 should be moved to Lemma C.4.

---

In addition, I see a lot of resemblance to Janz et al. (2024a), Lee & Oh (2024), and Jia et al. (2022) in the appendix, but in some parts, the authors do not mention these papers and proceed. I recommend the authors clearly state where the analysis originally came from, and cite proved results in these papers whenever possible, not just occasionally.

**Questions:**

Minor typos:

- Line 284 width L -> width N

- Lines 1266, 1275 are identical, when I think a different equation is intended for line 1275.

- In the proof for Neural-ES, I think $\theta_ 0$ should be replaced by $\theta_ 0^j$.

---

> ### Author Response · Authors · 2025-11-29
>
> ## Response to Reviewer j3cM
>
> We thank the reviewer for your valuable time and effort in providing detailed feedback on our work. We hope our response will fully address all your questions.
>
> ---
> ## Proof for GLM-ES
>
> ### **Q1:** Justification of using $H_{t}$ and $\bar{H_t}$
>
> **A1:** This section of the proof is fully revised. We should use $\bar{H_t}$ in Section B.3 and we provided the corresponding concentration results for $|| \bar{\theta_t} - \theta^{*} ||_ {\bar{H_t}}$ and $||\tilde{\theta_t^j} ||_ {\bar{H_t}}$.
>
>  We proved that with the warm-up procedures and self-concordant assumption, these two norms are of the same order as $||\bar{\theta_t} - \theta^* ||_ {H_t}$ and $||\tilde{\theta_t^j} ||_ {H_{t}}$, the difference can be bounded by a constant factor $C^*$ (Line 1326). Therefore, the order of the regret bound is not affected.
>
>
> ---
>
> ### **Q2:** Explanation for order of warm-up rounds
>
> **A2:** We would like to clarify that $\sigma_R \geq b_1+b_2 R_t \sigma_R$ is the sufficient condition of optimism (to guarantee Lemma B.3 holds with constant probability), therefore we want to choose a suitable $\sigma_R$ to satisfy that inequality. To ensure this, we desire to have $R_t \leq M D_t \leq \sigma_{R}^{-\varepsilon} < 1$ where $\varepsilon \in (0,1)$, which can be achieved by choosing $\iota \geq M b_2 \dot{\mu_{\text{max}}}^{-1} \sigma_{R}^{1+\varepsilon}$.
>
> Given $R_t \leq \sigma_{R}^{-\varepsilon} < 1$, we notice that the solution $\sigma_{R}$ for $\sigma_{R} \geq b_1 + b_2 \sigma_{R}^{1-\varepsilon}$ satisfies $\sigma_R \geq b_1+b_2 R_t \sigma_R$, and we solve this inequality to get the choice $\sigma_{R} \gtrsim (d \log T)^{\frac{1}{2\varepsilon}}$. Finally, the choice of $\varepsilon$ is not optimal, it is right that $\varepsilon \rightarrow 1$ would yield a better bound on the warm-up time. However, as $\varepsilon < 1$, the order of $d$ in $\tau$ will not be worse than $d^4$, therefore we make a conservative choice $\varepsilon = \frac{2}{3}$ and get the order $\tau = \widetilde{O}\big(d^{\frac{9}{2}}\big)$.
>
>
>
>
> ---
> ### **Q3:** Clarification for conditional probability of optimism
>
> **A3:** We revised the manuscript and explicitly provided the conditional probability of optimism after uniformly randomly choosing model $j_{t}$ by the following equation (Line 1356-1363):
>
> $\mathbb{P} \Big(\big(U_ {t-1}^{\top} \mathbf{Z}_ {t-1}^{j_t} \geq (b_1 + b_2 R_ {t-1} \sigma_ {R}) \| U_ {t-1} \|_ 2\big) \cap \tilde{\mathcal{E}}_ {1, t-1} | \mathcal{F}_ {t-1} \Big) = \frac{1}{m} \sum_{j=1}^{m} I_ {t}^{j}$.
>
> The right-hand-side of this equation is further lower bounded by the Azuma-Hoeffding inequality (Line 1365).
>
>
> ---
> ### **Q4:** Proof for Lemma A.7
>
> **A4:**  Proof of Lemma A.7 is added in Appendix B.5.
>
>
>
> ---
> ### **Q5:** Explanation for constant factor of $\ddot{b}(c) \leq e \ddot{b}(u)$ and $\ddot{b}(c) \leq 2 \ddot{b}(u)$
>
> **A5:** In the revised manuscript, we proved $\ddot{b}(c) \leq 2 \ddot{b}(u)$ and $s$ is constrained to $\vert s \vert \leq \log(2)/M$ (Line 1051-1056). According to that, we use $M/\log(2)$ instead of $M$ when applying Lemma B.1. The constant factors in the following analysis are revised accordingly. Note that this constant factor does not affect the asymptotic behavior of our theoretical guarantee.
>
>
>
> ---
> ### **Q6:** Clarification for order of $\delta, T$ and $p_N$
>
> **A6:**  We added detailed explanations of the order of each parameter in the manuscript at Line 1356. In the proof, we need the assumption that $\delta / T \leq p_{N} / 2$, which is a very moderate requirement for $\delta$.

---

> ### Author Response · Authors · 2025-11-29
>
> ---
> ## Proof for Neural-ES
>
> ### **Q7:** Proof for Lemma C.4
>
> **A7:** Our Lemma C.4 is not the same as in Jia et al. (2022) because the concentration result is not conditioned on the history $\mathcal{F}_ {t-1}$. The key in the proof is using the concentration of i.i.d. Gaussian variables (Lemma B.4 in Jia et al. (2022)). In our lemma, the perturbations are still i.i.d. sampled, we can still use this concentration result when not conditioned on the history. Therefore, the proof can still go through. This method is standard in the ensemble sampling literature (such as in Lee \& Oh (2024) Lemma 4), and we do not require conditional concentration result in the following regret analysis.
>
>
>
> ---
> ### **Q8:** Clarification for summation and order of $\alpha_T, \beta_T$ and $\tilde{d}$
>
> **A8:** We explained how to adapt Lemma C.7 to calculate the summation in the regret bound. We also added clarifications of the order of each parameter to explicitly calculate the order of the result. Using the parameters specified in Theorem 5.7, we have $\alpha_{T}, \beta_{T} \sim \widetilde{\mathcal{O}}\big(\widetilde{d}^{1/2}\big)$, thus we have the regret bound $R(T) = \widetilde{\mathcal{O}} (\widetilde{d} \sqrt{T})$ (Line 1704-1716).
>
>
>
>
>
> ---
> ### **Q9:** Clarification for conditional probability in the proof
>
> **A9:** We revised the proof and the left-hand-side should be the expected value, then we have the equation (Line 1670-1673)
> $\mathbb{E} \big[ ||g(X_t; \theta_0) / \sqrt{N}||_ {A_{t}^{-1}} \big]
>     \geq ||g(X_{t}^{(e)}; \theta_{0}) / \sqrt{N}||_ {A_{t}^{-1}} \mathbb{P}\big(X_{t} \in \bar{\Omega}_ {t}\big)$.
> As a result, now we provide the upper bound of the expectated value of cumulative regret $\mathbb{E}[R(t)]$ instead of the high-probability regret bound. Theorem 5.7 is changed accordingly.
>
>
>
> ---
> ### **Q10:** Details of computation of $U_{t}$
>
> **A10:**  We inserted the steps to calculate the variance of $U_{t}$. The details are at Line 1740-1748.
>
>
>
>
>
> ---
> ### **Q11:** Clarification for proof of Lemma C.5
>
> **A11:** We fixed the inequality to present the upper bound of $h(X_{t})$ at Line 1724, now the logic can lead to the sufficient condition of optimism.

---

> ### Author Response · Authors · 2025-11-29
>
> ---
> ## Presentation of the Paper
>
> ### **Q12:** Justification for fixed arm set
>
> **A12:** Regarding the terminology, we acknowledge the categorization presented in Lattimore \& Szepesvári (2020), which distinguishes "Linear Bandits" (fixed action set) from "Contextual Bandits" (changing action set) [4]. However, our usage aligns with the prevailing convention in the specific literature of Generalized Linear Bandits (GLB) and Neural Bandits, where "Contextual" is standardly used to denote problems where rewards are governed by a feature-dependent mapping, even when the arm set is fixed. For instance, Li et al. (2017) explicitly title their work "Provably Optimal Algorithms for Generalized Linear Contextual Bandits," despite analyzing settings often parameterized by fixed feature sets [1]. Similarly, foundational works in neural bandits, such as "Neural Contextual Bandits with UCB-based Exploration" by Zhou et al. (2020) [2] and "Neural Thompson Sampling" by Zhang et al. (2021) [3], consistently utilize the term "Contextual Bandit" to describe the learning of a complex nonlinear reward function over a finite set of arms $|\mathcal{X}|=K$. In this community, the term "contextual" serves to differentiate the problem from multi-armed bandits by emphasizing that the learner must generalize across the action space via the feature vectors (contexts), rather than learning independent reward distributions for each arm. Therefore, our terminology is chosen to maintain consistency with this directly related high-impact literature.
>
>
>
> ---
> ### **Q13:** Novelty of doubling trick
>
> **A13:** In Janz et al. (2024a), although the authors mentioned this technique, they did not provide any analysis or experiments. Our doubling-trick analysis is a direct application of the existing work (Besson & Kaufmann, 2018), but we added detailed analysis and experiments about how to set the parameters $T_{0}$ and $b$, as well as the empirical performance of doubling trick. We cited the original work (Besson & Kaufmann, 2018) in the manuscript. Although our regret bound is based on the existing work, we discussed how to practically apply doubling-trick in the nonlinear bandit settings and whether the anytime version of ensemble sampling is practical compared to baseline algorithms.
>
>
>
> ---
> ### **Q14:** Clarification for nonlinear bandit
>
> **A14:** The GLM and neural bandit settings are the most studied settings in the literature of non-linear bandit problems. Although we only provided theoretical guarantee for these two settings, we provide a general framework for ensemble sampling in bandit problems in this paper.
>
>
>
> ---
> ### **Q15:** Missing/misstated definitions
>
> **A15:** We carefully revised these definitions in the paper.
> (1) We added the definition of fixed arm set $\mathcal{X}$ at Line 158.
> (2) We added the definition of filtration $\mathcal{F}_ t$ at Line 188.
> (3) We included gradient descent procedures in our algorithm design in the previous manuscript but did not define the parameters $J$ and $\eta$ there. Now we added the formal definitions of $J$ and $\eta$ at Line 298.
> (4) We do require that $Y_{t}$ follows exponential family distribution in the analysis of GLM-ES, we added this to Assumption 5.1 (Line 330-332) in the revised manuscript.
> (5) We adjusted the order of the definition of notations $A_{K}$, $\lambda_{K}$ and $\sigma_{R}$ at Line 1610-1611.
>
>
> ---
> ### **Q16:** Appropriate citations
>
> **A16:** We revised the citations in this paper to correctly reflect the origins of the analysis.
>
>
>
> ---
> ### **Q17:** Minor typos
>
> **A17:** We fixed the mentioned typos in this paper.
> (1) We fixed width to $N$ at Line 291.
> (2) We corrected the equations at Line 1271 and Line 1280.
> (3) We use shared initialization $\theta_{0}$ in the algorithm design, which we now emphasize at Line 303 and Line 306. The difference between each model is only the perturbations on the reward $(Y_{t} + Z_{t}^{j})$.
>
>
>
>
> ---
> We hope we have addressed all of your questions. If you have any further questions, we would be happy to answer them and if you don’t, would you kindly consider increasing your score?
>
>
> ### References:
>
> [1] Li, Lihong, Yu Lu, and Dengyong Zhou. "Provably optimal algorithms for generalized linear contextual bandits." International Conference on Machine Learning. PMLR, 2017.
>
> [2] Zhou, Dongruo, Lihong Li, and Quanquan Gu. "Neural contextual bandits with ucb-based exploration." International conference on machine learning. PMLR, 2020.
>
> [3] Zhang, Weitong, et al. "Neural thompson sampling." arXiv preprint arXiv:2010.00827 (2020).
>
> [4] Lattimore, Tor, and Csaba Szepesvári. Bandit algorithms. Cambridge University Press, 2020.

---

### Official Review · Reviewer_hgVm · 2025-10-31

**Soundness:** 3
**Presentation:** 2
**Contribution:** 3
**Rating:** 4
**Confidence:** 4

**Summary:**

This paper proposes an ensemble sampling algorithm for K-armed contextual bandits with non-linear reward models.
Specifically, the authors develop a unified framework that covers both cases where the reward follows a generalized linear model (GLM) with respect to the arm’s context and where the reward function lies in a reproducing kernel Hilbert space (RKHS) with a bounded norm under the neural tangent kernel.
The proposed algorithm achieves regret bounds of $O(d^{3/2} \sqrt{T})$ for GLM rewards, and $O(\tilde{d} \sqrt{T})$ for RKHS-bounded rewards, given an ensemble of size $\Omega(K \log T)$.
Moreover, the paper addresses the case where the total number of rounds $T$ is unknown, employing a doubling trick to establish asymptotic regret guarantees.
The superiority of the proposed algorithm is validated through synthetic experiments.

**Strengths:**

1. Ensemble sampling has shown strong empirical performance in practice but has primarily been analyzed under linear reward models. This work extends the analysis to more general settings, including GLM and RKHS-bounded reward models, and demonstrates that the proposed algorithm achieves regret bounds matching those of existing linear or randomized algorithms.

2. The paper is clearly written and easy to follow. The literature review and discussion of related work are well-organized and help position the contribution effectively.

**Weaknesses:**

1. The notation throughout the paper should be carefully reviewed. For example,
- context feature is denoted by both $x$ and $X$ in different places
- $H_t$ is defined as the Hessian of the loss function at time $t$, yet the term $ \|| X_t \|_{H_t^{-1}}$ is used later as a bonus term, which may lead to confusion
- Does this work allow the context to change over time? The main theorem (Thm. 5.5 & 5.7) assumes a fixed arm set ($|\mathcal{X}| \le K$, but the problem setting (line 157) uses a varying arm set.
- Definition of $\theta_t$ in appendix seems missing.
- By definition, $\theta^\{j\_t}\_t$ is an estimator estimated using data up to time $t$. Then, shouldn't the behavior of $X_t^\top \theta^{j\_t}\_{t-1}$ be analyzed in optimism?

2. The description of the warm-up phase could be expanded. In Remark 4.2, the authors note that while previous GLM-bandit studies use warm-up rounds to ensure the invertibility of the empirical Gram matrix, the proposed method employs them to guarantee constant-probability optimism. Since the purpose differs, it would be helpful to elaborate on how the rounding procedure in Algorithm 3 operates—for example, how it differs from exploring for $\tau$ rounds using the G-optimal design solution $\zeta$ as the policy.

3. Although this issue is not unique to this paper, the neural network size required for Neural-ES appears to be very large (see the condition in Lemma C.4), which may limit the practical applicability of the algorithm.

**Questions:**

1. In Remark 5.4, the authors state that the expected Gram matrix is not necessarily assumed to have a positive minimum eigenvalue. However, under this case, how is the existence of the optimal design distribution (\zeta) guaranteed?

2. Neural-ES employs a different warm-up procedure from GLM-ES. What allows Neural-ES—despite dealing with a more general reward model—to avoid the complex warm-up required in the GLM-ES setting?

3. How does the MLE concentration result in Lemma A.3 differ from Lemma A.2 in Sawarni et al. (2024)?

4. The doubling trick (Besson & Kaufmann, 2018) is a well-established approach in online learning to handle unknown $T$, with known regret bounds. How does the result in this paper differ from the standard doubling-trick analyses already available in the literature?

5. In line 1244t, the sufficient condition for optimism is written by $H_t$, but it was changed to $\bar{H}_t$ in Eq. (B.8). Could you explain the process in a little detail?

**Details Of Ethics Concerns:**

N.A.

---

> ### Author Response · Authors · 2025-11-29
>
> ## Response to Reviewer hgVm
>
> We thank the reviewer for your valuable time and effort in providing detailed feedback on our work. We hope our response will fully address all your questions.
>
> ---
> ### **Q1:** Clarification for notations
>
> **A1:** We carefully revised the notations in this paper and fixed the mentioned issues.
>
> (1) We use $X$ for context features throughout this paper, any usage of $x$ for context features is changed to $X$.
>
> (2) We define $H_{t}$ to be the Hessian throughout the paper, but we don't have bonus part since this is not a UCB algorithm. The term $||X_{t}||_ {H_ {t}^{-1}}$ just appears in the proof due to algebra.
>
> (3) We consider fixed arm set in this paper. We added the definition of $\mathcal{X}$ in the problem setting at Line 158.
>
> (4) $\theta_{t} = \theta_{t-1}^{j_{t}}$ is the parameter of the chosen model at round $t$. We added the definition of $\theta_{t}$ in the Appendix at Line 815.
>
> (5) We revised this notation such that we use $\theta_{t} = \theta_{t-1}^{j_{t}}$ as the chosen parameter estimate at round $t$, where $\theta_{t-1}^{j}$ are parameter estimates based on history up to $(t-1)$ for each model $j\in [m]$, and $j_{t}$ is the chosen model at round $t$. The notations in the analysis are revised accordingly.
>
> ---
> ### **Q2:** Detailed discussions for warm-up phase
>
> **A2:** The rounding procedure in Algorithm 3 operates as a deterministic discretization mechanism that converts the continuous optimal weights $\zeta$ from the G-optimal design into exact integer pull counts $N_i$ for the support arms, such that $\sum N_i=\tau$. This differs fundamentally from exploring using $\zeta$ as a stochastic policy (i.e., sampling arms randomly according to $\zeta$ ). While stochastic sampling converges to the optimal design asymptotically, it introduces variance in finite samples that may leave critical directions undersampled, thereby failing to strictly guarantee the spectral properties required for our theoretical analysis. In our framework, the warm-up is not merely to ensure the Gram matrix is invertible, but to explicitly strictly bound the maximum predictive variance (quantified by $D_t$ in Lemma B.5) below a specific threshold. This strict bound is necessary to ensure that the added perturbations are sufficient to induce optimism with constant probability. The deterministic rounding guarantees this condition is met within the specified $\tau$ rounds, avoiding the probabilistic failure modes associated with random sampling.
>
> ---
> ### **Q3:** Explanation for large neural network size and limitations
>
> **A3:** As in previous works in neural bandit settings, we require over-parameterization to gaurantee that the neural network can approximate the true reward function (as in Lemma C.2). This requirement is only for proving the theoretical guarantee. As demonstrated in the experiments, the ensemble sampling algorithm is empirically competitive even with small to moderate neural network sizes. For example, we use 3-layer fully connected neural network with width $N=20$ for each layer for experiments, the empirical accumulated regret is considerably lower than baseline algorithms in various bandit environments.
>
> ---
> ### **Q4:** Explanation for Remark 5.4 & How is the existence of the optimal design distribution ($\zeta$) guaranteed?
>
> **A4:** We clarify that the assumption removed in Remark 5.4 pertains to the expected Gram matrix of the random context distribution (i.e., we do not require the environment to stochastically provide diverse features). However, the existence of the G-optimal design distribution $\zeta$ depends solely on the geometry of the available action set $\mathcal{X}$, not the stochastic context distribution. As stated in Lemma B.5, we assume the fixed action set $\mathcal{X}$ spans $\mathbb{R}^d$. Under this condition, the G-optimal design $\zeta$ is guaranteed to exist. Our warm-up procedure then actively selects arms based on $\zeta$ to construct a well-conditioned history, thereby bypassing the need for the environment's natural distribution to be well-conditioned. If $\mathcal{X}$ does not span $\mathbb{R}^d$, the problem can be projected onto the subspace spanned by $\mathcal{X}$ without affecting the theoretical guarantees.

---

> ### Author Response · Authors · 2025-11-29
>
> ---
> ### **Q5:** What allows Neural-ES-despite dealing with a more general reward model-to avoid the complex warm-up required in the GLM-ES setting?
>
> **A5:** Neural-ES avoids the complex G-optimal design warm-up because it operates in the Neural Tangent Kernel (NTK) regime, where the network width is sufficiently large that the reward function is well-approximated by a linear model with fixed features corresponding to the gradients at initialization. In contrast, the GLM-ES analysis relies on the self-concordance property of the non-linear link function, where the Hessian depends on the unknown parameter $\theta$. The complex G-optimal design warm-up is strictly necessary in GLM-ES to ensure the initial parameter estimate is close enough to the true parameter, thereby bounding the local geometric distortion $\left(D_t\right)$ and guaranteeing that the perturbations are sufficient to induce optimism. Since Neural-ES essentially reduces to a high-dimensional linear bandit, a simple warm-up of pulling each arm once suffices to ensure the initial covariance matrix is well-defined.
>
> ---
> ### **Q6:** How does the MLE concentration result in Lemma A.3 differ from Lemma A.2 in Sawarni et al. (2024)?
>
> **A6:** Lemma A.2 in Sawarni et al. (2024) gives the upper bound of $|| \theta^{*} - \bar{\theta_t} ||_{H_t}$ at round $t$ as $\widetilde{O}(M^{2}d\lambda^{-1/2})$. The upper bound holds with probability at least $(1 - T^{-2})$, thus the union bound is $(1 - T^{-1})$ for this concentration result to hold for all rounds $t \in [T]$. When choosing $\delta = 1/T$, our Lemma A.3 gives the upper bound $\widetilde{O}(M^{4}d^{3}\lambda^{-5/2})$. When using regularization parameter as $\lambda = 1 \vee (2dM/S)$ (as in Theorem 5.5), we have the same order of concentration $\widetilde{O}(M^{3/2}d^{1/2})$ from both lemmas. Therefore, although these two concentration bounds are not identical, with our parameter setup in Theorem 5.5, they provide the same order of concentration result.
>
>
> ---
> ### **Q7:** Explanation for doubling-trick analyses
>
> **A7:** Our doubling-trick analysis is an application of the existing work (Besson & Kaufmann, 2018) in nonlinear bandit settings, but we added detailed analysis and experiments about how to set the parameters $T_{0}$ and $b$, as well as the empirical performance of doubling trick. We cited the original work (Besson & Kaufmann, 2018) in the original manuscript. Although our regret bound is based on the existing work, we discussed how to practically apply doubling-trick in the nonlinear bandit settings and whether the anytime version of ensemble sampling is practical compared to baseline algorithms.
>
>
> ---
> ### **Q8:** Detailed explanation for the sufficient condition for optimism in line 1244
>
> **A8:** This section of the proof is fully revised. We should use $\bar{H_t}$ in Section B.3 and we provided the corresponding concentration results for $|| \bar{\theta_t} - \theta^{*} ||_ {\bar{H_t}}$ and $||\widetilde{\theta}^j_t ||_ {\bar{H_t}}$.
>
> We proved that with the warm-up procedures and self-concordant assumption, these two norms are of the same order as $||\bar{\theta_t} - \theta^*||_ {H_t}$ and $||\tilde{\theta}^j_t||_ {H_t}$, the difference can be bounded by a constant factor $C^\star$ (Line 1326). Therefore, the order of the regret bound is not affected.
>
>
>
> ---
> We hope we have addressed all of your questions. If you have any further questions, we would be happy to answer them and if you don’t, would you kindly consider increasing your score?

---

### Official Review · Reviewer_V6D8 · 2025-10-31

**Soundness:** 3
**Presentation:** 3
**Contribution:** 2
**Rating:** 4
**Confidence:** 3

**Summary:**

This paper introduces Provable Anytime Ensemble Sampling, a unified framework for ensemble sampling in nonlinear contextual bandits with two concrete instantiations: Generalized Linear Model Ensemble Sampling (GLM-ES) and Neural Ensemble Sampling (Neural-ES), and extends both to the anytime setting via a doubling-trick schedule. The core mechanism is randomized exploration through reward perturbation while reusing past perturbations so that per-round computation remains constant. Under a fixed and finite action set with bounded features, the authors derive high-probability regret guarantees for GLM-ES (under M-self-concordance and a positive lower bound on the link derivative over the working domain) and for Neural-ES (under neural tangent kernel conditions for sufficiently wide networks). The anytime variants achieve the same asymptotic regret up to a constant factor, and experiments on linear, logistic, distance, and quadratic synthetic environments show competitive cumulative regret with noticeably reduced computational cost and no need to know the time horizon in advance.

**Strengths:**

The framework unifies randomized exploration across linear and nonlinear bandits with a single, conceptually clean recipe that reuses historical perturbations to keep updates inexpensive. The theory is clearly presented and provides high-probability guarantees, together with an anytime extension that is simple and practical. The GLM instantiation reduces the warm-up complexity relative to prior perturb-and-estimate methods, and the paper is generally well written and organized.

**Weaknesses:**

The empirical setup lists hyper-parameter choices and network sizes but gives little justification for why these values were selected, and it lacks systematic sensitivity/ablation results for ensemble size, regularization, perturbation variance, or the anytime schedule. Neural experiments are limited to a small multilayer perceptron, so it remains unclear whether the computational and statistical behavior persists for wider/deeper architectures used in practice.

Several modeling assumptions that are central to the guarantees (fixed finite action set, bounded features, GLM regularity, and wide-network/NTK-style training for Neural-ES) are not emphasized in the discussion of applicability. The paper would also benefit from clarifying the data-generation details in the synthetic environments (e.g., how the unknown parameter and action features are sampled) and from commenting on the ensemble size scaling with the number of actions, which may be large in real applications.

**Questions:**

1.Could the authors provide a more systematic sensitivity and ablation analysis covering ensemble size, regularization, perturbation variance, and the anytime schedule (e.g., base segment length and growth factor), including both regret and runtime/memory effects?
2.For Neural-ES, do the computational and theoretical advantages extend beyond small MLPs to wider/deeper networks commonly used in practice (e.g., larger MLPs or transformer backbones), and what concrete width/optimization regimes are required for the stated guarantees to remain valid?

The evaluation uses synthetic environments, which is standard for this area but leaves open how the approach behaves in more complex or real-world contexts (e.g., with changing action sets, larger feature spaces, or modern neural architectures).

---

> ### Author Response · Authors · 2025-11-29
>
> ## Response to Reviewer V6D8
>
> We thank the reviewer for your valuable time and effort in providing detailed feedback on our work. We hope our response will fully address all your questions.
>
> ---
> ### **Q1:** Additional experiments on sensitivity/ablation for ensemble size, regularization, perturbation variance, or the anytime schedule (regret and runtime/memory effects)
>
> **A1:** We test the following hyper-parameter settings on logistic bandit:
> Ensemble size $m$: 5, 50
> Regularization $\lambda$: 0, 0.1, 5.0
> Perturbation variance $\sigma_{R}$: 0.02, 0.2, 0.5
> The experiment results are plotted in Figure 2. Here we summarize the findings.
> (1) By increasing ensemble size $m$, we get marginal improvement of cumulative regret. Moderate ensemble size such as $m=10$ suffices to reach competitive performance for GLM-ES.
> (2) Using very small $\lambda$ causes instability in parameter estimation, while very large $\lambda$ causes estimated parameter to be too small. $\lambda \sim 1$ is a proper choice, the cumulative regret becomes higher when regularization is too small or too large.
> (3) Using very small $\sigma_{R}$ cannot provide enough exploration, while very large $\sigma_{R}$ introduces too much noise. $\sigma_{R} \sim 0.1$ is a proper choice to balance these two effects.
> We also added experiments with the following anytime schedule:
> $(T_{0}=100, b=2), (T_{0}=100, b=4)$
> The experiment results are plotted in Figure 3.
> In general, the regret bound becomes smaller as we choose higher $T_{0}$ and $b = (3 + \sqrt{5})/2 \approx 2.6$. Also, $T_{0}$ has minor effect on the performance of anytime ensemble sampling, while $b$ can significantly affect the performance.
> For practicality, we should keep $b$ at the optimal value $b = (3 + \sqrt{5})/2 \approx 2.6$ and set $T_{0}$ in the order of $10^{2}$.
> The results and discussions are added as Appendix F.
>
>
> ---
> ### **Q2:** Additional experiments on wider/deeper architectures
>
> **A2:** We test the following neural network architectures (width $N$, number of layers $L$):
> $(N=10, L=3), (N=20, L=4), (N=40, L=3)$.
> The results are plotted in Figure 4.
> In general, the performance of the model is not significantly improved with larger network structures beyond our chosen width and depth. Also, expanding the width of the neural network is generally more effective than adding more layers, which agrees with our theoretical guarantees.
> The results and discussions are added as Appendix F.
>
>
> ---
> ### **Q3:** Explanation for modeling assumptions and applicability
>
> **A3:** Regarding the modeling assumptions, the constraints of a fixed finite action set and bounded features $(\|X\|_2 \leq 1)$ are standard theoretical conditions to ensure that the reward function is realizable and gradients of loss functions remain bounded during optimization. In practical applications, these conditions are met by normalizing feature vectors and treating the action set as a candidate pool (e.g., in recommendation systems). Similarly, the wide-network assumption in Neural-ES ensures the Neural Tangent Kernel (NTK) regime holds for theoretical convergence. However, our empirical results show that standard network widths ($N=20$) are sufficient for strong performance.
>
> ---
> ### **Q4:** Explanation for data-generation details in the synthetic environments and ensemble size
>
> **A4:** To clarify the data generation, in our synthetic environments (Linear, Logistic, Distance, Quadratic) with $K=50$ and $d=20$, the unknown parameters $\theta^*$ and action feature vectors $X$ were sampled from a standard Gaussian distribution $\mathcal{N}(0, I_d)$ and normalized to unit $\ell_2$-norm to satisfy the boundedness assumption.
>
> Regarding ensemble size, while our theoretical analysis requires $m=\Omega(K \log T)$ to rigorously guarantee optimism via a union bound over $K$ arms, this is a conservative worst-case bound. Practically, we observe that ensemble sampling is efficient with much smaller sizes; our experiments achieved competitive regret using $m=10$ or a logarithmic schedule ($m \approx 2 \log \tau_i$) despite $K=50$, suggesting that $m$ does not need to scale linearly with $K$ in real-world deployments.
>
> ---
> ### **Q5:** Additional experiments on more complex or real-world contexts (e.g., with changing action sets, larger feature spaces, or modern neural architectures).
>
> **A5:** We added experiments at larger feature spaces with context dimension $d=50$ and number of arms $K=500$. We test the ensemble sampling algorithm along with baseline algorithms.
> The results are plotted in Figure 4.
> Our ensemble sampling algorithms remain competitive in larger feature spaces.
> The results and discussions are added as Appendix F.

---

> ### Author Response · Authors · 2025-11-29
>
> ---
> ### **Q6:** For Neural-ES, do the computational and theoretical advantages extend beyond small MLPs to wider/deeper networks commonly used in practice (e.g., larger MLPs or transformer backbones), and what concrete width/optimization regimes are required for the stated guarantees to remain valid?
>
> **A6:** Computationally, the advantage of Neural-ES extends directly to wider and deeper networks, including Transformer backbones. This benefit stems from the algorithm's use of incremental updates with persistent perturbations, which avoids the prohibitive cost of resampling the entire perturbation history and retraining required by methods like NPR [1]. This efficiency is critical for large-scale architectures where full retraining is computationally infeasible. Theoretically, our guarantees rely on the Neural Tangent Kernel (NTK) regime, where the network stays close to its linearization. To ensure this validity, the network width $N$ is required to scale polynomially with depth $L$ and horizon $T$, specifically adhering to a condition roughly proportional to $\tilde{\Omega}\left(T^7 L^{21}\right)$ to strictly control linearization error, alongside a small learning rate of $\eta=\mathcal{O}\left((N \lambda+N L T)^{-1}\right)$. However, while these wide-width conditions are necessary for the rigorous proof, our empirical results indicate the method is robust in standard, much smaller regimes (e.g., width $N=20$), suggesting practical applicability beyond the strict theoretical bounds [1,2,3,4].
>
>
> ---
> We hope we have addressed all of your questions. If you have any further questions, we would be happy to answer them and if you don’t, would you kindly consider increasing your score?
>
>
> ### References:
>
> [1] Jia, Yiling, et al. "Learning neural contextual bandits through perturbed rewards." arXiv preprint arXiv:2201.09910 (2022).
>
> [2] Zhou, Dongruo, Lihong Li, and Quanquan Gu. "Neural contextual bandits with ucb-based exploration." International conference on machine learning. PMLR, 2020.
>
> [3] Zhang, Weitong, et al. "Neural thompson sampling." arXiv preprint arXiv:2010.00827 (2020).
>
> [4] Nguyen-Tang, Thanh, et al. "Offline neural contextual bandits: Pessimism, optimization and generalization." arXiv preprint arXiv:2111.13807 (2021).

---

### Meta-Review · Area_Chair_ZDkC · 2026-01-06

**Summary:**

While most reviewers like the significance of this work, they also have concerns on the experiments, notations, and theoretical analysis. Some of the concerns have been addressed by the author rebuttal and the paper revision, but theoretical analysis remains a big concern.

More importantly, one reviewer found that this work contains multiple parts that are highly similar or nearly identical to portions of prior related work, raising serious concerns about potential plagiarism. In detail, there is a concerning amount of resemblance between this submission and two papers, Lee & Oh (2024) and Janz et al. (2024b). While the authors cited them in this paper, after carefully reading the submission and these two papers, I confirm some parts are very concerning where even identical notations are being used. Given the unaddressed concerns, I recommend reject.

Below I copy the concerns raised by the reviewer:

Appendix A.1, Line 730-789:
Appendices C.2 and C.3 of Janz et al. (2024b).
Other than notational differences, the order in which they introduce new concepts, facts, display equations, and a corollary from another work is very similar.

Appendix A.3 Line 885-971:
Proof of Theorem 2 in Appendix B of Lee & Oh (2024).
The order of the equations and their contents are very similar. Some sentences are nearly identical. While other sentences are not exactly the same, their meaning is the somewhat similar.

Appendix B.1, Line 996-1025, 1055-1063:
Appendix D.2 of Janz et al. (2024b).
Again, other than notational differences and paraphrasing, the two parts are very similar.

Appendix B.4, Line 1316-1347:
Section 5.2 of Lee & Oh (2024), specifically from the paragraph after Eq.(8) to Proposition 1.
The order and the contents of the equations are very similar, with some inline equations in Lee & Oh (2024) being display equations in this submission.

[1] Harin Lee and Min-hwan Oh. Improved regret of linear ensemble sampling. Advances in Neural Information Processing Systems, 37:92803–92831, 2024. https://arxiv.org/abs/2411.03932

[2] David Janz, Shuai Liu, Alex Ayoub, and Csaba Szepesvári. Exploration via linearly perturbed loss minimisation. In International Conference on Artificial Intelligence and Statistics, pp. 721–729. PMLR, 2024b. https://arxiv.org/abs/2311.07565

**Reviewer Concerns:**

Reviewers have concerns on the experiments, notations, and theoretical analysis. Some of the concerns have been addressed by the author rebuttal and the paper revision. However, the theoretical part remains a big concern. More importantly, one reviewer found that this work contains multiple parts that are highly similar or nearly identical to portions of prior related work, raising serious concerns about potential plagiarism.

**Reviewer Scores:**

Unfortunately, all reviewers didn't reply to the author rebuttal. Given the reviews, after carefully reading the author rebuttal, I find the reviewers were leaning towards a unanimous decision towards reject, so I don't think they could raise their scores if they had been able to fully engage in the author-reviewer discussion.

---

### Decision · Program_Chairs · 2026-01-26

Reject